# Conditional KRR: Injecting Unpenalized Features into Kernel Methods with Applications to Kernel Thresholding

**Rustem Takhanov** [*1 2]  **Zhenisbek Assylbekov** [3]

## Abstract

Conditionally positive definite (CPD) kernels are defined with respect to a function class $\mathcal{F}$. It is well known that such a kernel $K$ is associated with its native space (defined analogously to an RKHS), which in turn gives rise to a learning method — called conditional kernel ridge regression (conditional KRR) due to its analogy with KRR — where the estimated regression function is penalized by the square of its native space norm. This method is of interest because it can be viewed as classical linear regression, with features specified by $\mathcal{F}$, followed by the application of standard KRR to the residual (unexplained) component of the target variable. Methods of this type have recently attracted increasing attention.

We study the statistical properties of this method by reducing its behavior to that of KRR with another fixed kernel, called the residual kernel. Our main theoretical result shows that such a reduction is indeed possible, at the cost of an additional term in the expected test risk, bounded by $\mathcal{O}(1/\sqrt{N})$, where $N$ is the sample size and the hidden constant depends on the class $\mathcal{F}$ and the input distribution.

This reduction enables us to analyze conditional KRR in the case where $K$ is positive definite and $\mathcal{F}$ is given by the first $k$ principal eigenfunctions in the Mercer decomposition of $K$. We also consider the setting where $\mathcal{F}$ consists of $k$ random features from a random feature representation of $K$. It turns out that these two settings are closely related. Both our theoretical analysis and experiments confirm that conditional KRR outperforms

standard KRR in these cases whenever the $\mathcal{F}$-component of the regression function is more pronounced than the residual part.

## 1. Introduction

Kernel Ridge Regression (KRR) is a powerful supervised learning method that has found applications in the learning theory of neural networks (Jacot et al., 2018; 2020a), operator approximation (Köhne et al., 2025), and reinforcement learning (Novelli et al., 2025), and the study of the manifold hypothesis in machine learning (Takhanov, 2026), among others. To apply the method to a specific learning task, one must define a positive definite function $K(x, y)$ on pairs of inputs, called the kernel function. It has been observed that for KRR (and other kernel-based methods such as SVM or Kernel PCA), the requirement of positive definiteness can be relaxed to the more general property of conditional positive definiteness (Schölkopf, 2000; Chi et al., 2022). For a kernel that is conditionally positive definite (CPD) w.r.t. a class of functions $\mathcal{F}$, we only require that the quadratic form $\sum_{ij} K(x_i, x_j)\zeta_i\zeta_j$ is non-negative for any vector $[\zeta_i]$ orthogonal to the set $\{[f(x_i)] \mid f \in \mathcal{F}\}$. Classical techniques such as spline estimation and Gaussian process regression are parameterized by kernels of this type, where the class $\mathcal{F}$ is interpreted as a set of unpenalized features. This connection has made the study of CPD kernels an important theme in approximation theory over the past decades (Wahba, 1990; Poggio & Girosi, 1990; Schaback & Wendland, 2006).

The majority of work on CPD kernels focuses on the case where $\mathcal{F}$ is defined as the set of multivariate polynomials of degree at most $k$ (Duchon, 1977), or on variations of this definition, while the case of a general $\mathcal{F}$ has largely been neglected. One of the motivations of the present paper is that, even when the original kernel $K$ is simply positive definite, treating it as a CPD kernel w.r.t. a general class of functions $\mathcal{F}$ leads naturally to a broader framework, which we call conditional KRR. This extension allows us to develop a non-trivial statistical theory of learning within this setting, thereby deepening our understanding of standard KRR.

The organization of the paper is as follows. In Section 2, we

[1]Department of Mathematics, Nazarbayev University, Astana, Kazakhstan [2]Nazarbayev University Research Administration, Astana, Kazakhstan [3]Purdue University Fort Wayne, Indiana, USA. Correspondence to: Rustem Takhanov <rustem.takhanov@nu.edu.kz>.

*Proceedings of the 43$^{rd}$ International Conference on Machine Learning*, Seoul, South Korea. PMLR 306, 2026. Copyright 2026 by the author(s).

define CPD kernels and introduce the associated notion of the *residual kernel*, proving that the latter is positive definite (Theorem 2.1). Similar constructions are standard in the theory of native spaces induced by CPD kernels (e.g., see (Meinguet, 1979)), but our definition depends explicitly on the input data distribution, which makes it central to the subsequent development. In Section 3, after recalling the standard definition of a native space, we provide an alternative characterization in terms of the Reproducing Kernel Hilbert Space (RKHS) associated with the residual kernel (Theorem 3.1). The conditional KRR is then formulated analogously to the standard KRR, with the regularization term replaced by the squared native space semi-norm. The residual kernel further allows us to interpret this problem as a combination of linear regression and standard KRR applied to residual data (see Theorem 3.2; Diagram 1 provides a visualization of this result).

Section 4 develops the statistical theory of conditional KRR. In our framework, the regression function is decomposed into two components: the first belonging to $\mathcal{F}$ and the second to the RKHS of the residual kernel. We introduce the concept of an $\mathcal{F}$-conditional learner, which has full access to the $\mathcal{F}$-component the regression function and learns the second component from data using standard KRR with the residual kernel. To analyze the statistical properties of the estimator produced by conditional KRR, we compare it with the output of this learner. The distance between the two estimators is referred to as the *cost of conditioning*. This quantity measures the extent to which conditional KRR can be viewed as standard KRR with a modified kernel. Our main theoretical result, stated in Theorem 4.2, establishes that with probability at least $1 - \delta$, the cost of conditioning is bounded by $C \frac{\log k}{\sqrt{N}}$, where $N$ is the sample size, $k$ is the dimension of $\mathcal{F}$, and $C$ hides logarithmic factors in $k$ and $\delta$, as well as additional dependencies on the regression function, $K$, and $\mathcal{F}$.

In the next part of the paper (Section 5), we apply our theoretical results to the case where the initial kernel $K$ is already positive definite and, consequently, CPD w.r.t. any class $\mathcal{F}$. We study conditional KRR under three specific scenarios: (a) the *hard thresholding* case, i.e. where $\mathcal{F}$ is defined as the first $k$ principal eigenfunctions in the Mercer decomposition of $K$ (subsection 5.1); (b) the *soft thresholding* case, i.e. where $\mathcal{F}$ consists of $k$ random realizations of a Gaussian process with covariance function $K$ (subsection 5.2); (c) $\mathcal{F}$ consists of $k$ random features (or, equivalently, $k$ realizations of a random field) whose covariance function is $K$. Our theoretical analysis, corroborated by experimental evidence, demonstrates that the expected test risk of conditional KRR is strictly lower than that of standard KRR, provided that the $\mathcal{F}$-component of the signal is sufficiently strong.

**Related work**. The statistical properties of the KRR regression function estimator have been studied extensively, with particular focus on convergence rates (Caponnetto & De Vito, 2007; Marteau-Ferey et al., 2019; Cui et al., 2021), the distribution of expected risk under universality assumptions (Bordelon et al., 2020; Jacot et al., 2020a; Simon et al., 2023), and the double-descent phenomenon (Mei & Montanari, 2022; Nakkiran et al., 2021). Our results show that these existing estimates can be directly extended to conditional KRR, provided that one accounts for the cost of conditioning.

Conditional KRR belongs to a broader family of two-stage methods: first recovering the main component of the signal with a base neural network, and then learning from the residuals. As shown in (Yang et al., 2023), this strategy yields lower test risk than relying on the base network alone and additionally allows explicit memorization of the training labels. This line of research is related to the classical works on boosting (Freund & Schapire, 1997), where the strategy is to iteratively refine an ensemble by training each new weak learner on the residual errors left by the previous ones.

**Conflict of Interest Disclosure**. The authors declare that there are no financial conflicts of interest associated with this research.

## 2. Conditionally positive definite and residual kernels

Let $\mathcal{X} \subseteq \mathbb{R}^d$ be a compact set, $f_1, \ldots, f_k : \mathcal{X} \to \mathbb{R}$ be linearly independent continuous functions. Define $\mathcal{F} = \mathrm{span}\{f_1, \ldots, f_k\} \subseteq \mathbb{R}^{\mathcal{X}}$. We assume that $K : \mathcal{X} \times \mathcal{X} \to \mathbb{R}$ is a symmetric, continuous, conditionally positive definite (CPD) kernel with respect to $\mathcal{F}$, that is it satisfies: for any points $x_1, \ldots, x_n \in \mathcal{X}$ and any coefficients $\alpha_1, \ldots, \alpha_n \in \mathbb{R}$ such that

$$\sum_{i=1}^{n} \alpha_i f(x_i) = 0 \text{ for all } f \in \mathcal{F},$$

we have

$$\sum_{i=1}^{n} \sum_{j=1}^{n} \alpha_i \alpha_j K(x_i, x_j) \geq 0.$$

Note that if the inequality holds for all $\boldsymbol{\alpha} \in \mathbb{R}^n$ without additional constraints, then $K$ is said to be positive definite (PD).

Denote by $\mathcal{B}(\mathcal{X})$ the Borel $\sigma$-algebra on $\mathcal{X}$, and by $\mathcal{P}(\mathcal{X})$ the set of probability Borel measures on $\mathcal{X}$. For $P \in \mathcal{P}(\mathcal{X})$, the projection of $f$ onto $\mathcal{F}$, denoted $\Pi_P f$, is defined by $\Pi_P f(x) = \int_{\mathcal{X}} \Pi(x, y) f(y) dP(y)$, where $\Pi(x, y)$ is the kernel associated with the projection operator, given by

$$\Pi(x, y) = \sum_{i,j=1}^{k} (G^+)_{ij} f_i(x) f_j(y),$$

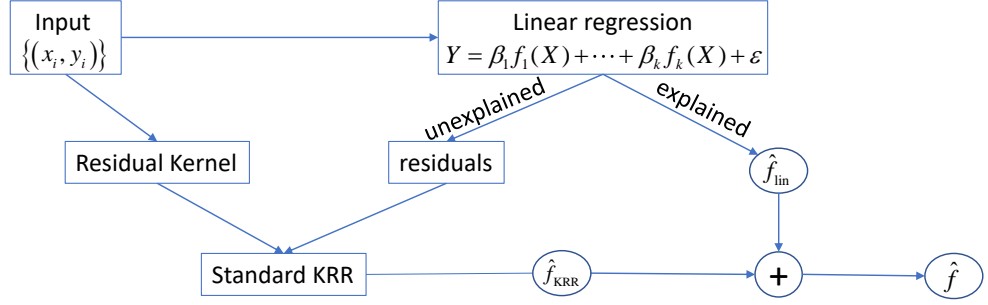

Figure 1. Structure of conditional KRR for $\mathcal{F} = \text{span}(\{f_1, \cdots, f_k\})$.

for $G = [\langle f_i, f_j \rangle_{L_2(\mathcal{X},P)}]_{i,j=1}^k$ and $G^+$ is the Moore-Penrose inverse of $G$. Given a function $f(x,\omega)$, the notation $\Pi_P f(\cdot, \omega)$ refers to the projection operator applied to the function $f$ for fixed $\omega$. The result is another function $\tilde{f}(x,\omega)$. If $G$ is invertible, then $\Pi_P f = f$ for any $f \in \mathcal{F}$. In this case, the distribution $P$ is said to be $\mathcal{F}$-nondegenerate. The following theorem extends the construction of the kernel given in equation (20) of (Meinguet, 1979).

**Theorem 2.1.** *Let $P \in \mathcal{P}(\mathcal{X})$ be $\mathcal{F}$-nondegenerate. Define the residual kernel*

$$K_P(x,y) = K(x,y) - \Pi_P[K(x,\cdot)](x,y) - $$
$$- \Pi_P[K(\cdot,y)](x,y) + \Pi_P[\Pi_P[K(x,\cdot)](\cdot,y)](x,y).$$

*Then, $K_P(x,y)$ is a positive definite kernel.*

Note that, using slightly more advanced notation, one can write $K_P = ((I - \Pi_P) \otimes (I - \Pi_P))[K]$, where $I$ denotes the identity operator on $L_2(\mathcal{X}, P)$, and $\otimes$ is the tensor product of operators on $L_2(\mathcal{X}, P)$, producing an operator on $L_2(\mathcal{X}, P) \otimes L_2(\mathcal{X}, P)$.

## 3. The native space and Ridge Regression with CPD kernels

The reduced native space of a CPD kernel $K$ w.r.t. $\mathcal{F}$, denoted $\widetilde{\mathcal{H}}_K^{\mathcal{F}}$, is defined as the completion of

$$\mathcal{L} = \big\{ f = \sum_{i=1}^n \alpha_i K(x_i, \cdot) \mid$$

$$\sum_{i=1}^n \alpha_i f_j(x_i) = 0 \text{ for all } j = 1, \ldots, k \big\},$$

equipped with the inner product

$$\langle \sum_{i=1}^n \alpha_i K(x_i, \cdot), \sum_{j=1}^n \beta_j K(x_j, \cdot) \rangle_{\mathcal{L}} = \sum_{i,j=1}^n \alpha_i \beta_j K(x_i, x_j).$$

The space $\widetilde{\mathcal{H}}_K^{\mathcal{F}}$ is a Hilbert space (Wendland, 2004).

Since $K$ is continuous, the reduced native space $\widetilde{\mathcal{H}}_K^{\mathcal{F}}$ embeds naturally into $C(\mathcal{X})$. Hence, w.l.o.g., we may regard $\widetilde{\mathcal{H}}_K^{\mathcal{F}}$ as a subspace of $C(\mathcal{X})$. The full native space is then defined as the direct sum

$$\mathcal{H}_K^{\mathcal{F}} = \widetilde{\mathcal{H}}_K^{\mathcal{F}} \oplus \mathcal{F},$$

equipped with the semi-norm

$$\|f\|_{\mathcal{H}_K^{\mathcal{F}}} = \sqrt{\langle f_\perp, f_\perp \rangle_{\widetilde{\mathcal{H}}_K^{\mathcal{F}}}}$$

where $f = f_\| + f_\perp$ is the unique decomposition with $f_\| \in \mathcal{F}$ and $f_\perp \in \widetilde{\mathcal{H}}_K^{\mathcal{F}}$. This semi-norm corresponds to the inner product $\langle \cdot, \cdot \rangle_{\widetilde{\mathcal{H}}_K^{\mathcal{F}}}$, turning $\mathcal{H}_K^{\mathcal{F}}$ into a semi-Hilbert space. The subspace $\mathcal{F}$ is referred to as the null space of $\mathcal{H}_K^{\mathcal{F}}$.

Let $\mathcal{H}_{K_P}$ denote the RKHS of the residual kernel $K_P$. Note that functions in $\mathcal{H}_{K_P}$ are all orthogonal to $\mathcal{F}$ in $L_2(\mathcal{X}, P)$. Then we define the semi-Hilbert space $\mathrm{H}_K^{\mathcal{F}}$ as the set of functions $\mathcal{H}_{K_P} \oplus \mathcal{F}$ with the inner product

$$\langle f_\| + f_\perp, g_\| + g_\perp \rangle_{\mathrm{H}_K^{\mathcal{F}}} = \langle f_\perp, g_\perp \rangle_{\mathcal{H}_{K_P}},$$

where $f_\| \in \mathcal{F}, f_\perp \in \mathcal{H}_{K_P}$. The following theorem claims that the latter two definitions are equivalent. It is a generalization of Theorem 4 from (Cucker & Smale, 2001) for PD kernels to the case of CPD kernels.

**Theorem 3.1.** *Let $P$ be a probabilistic Borel measure non-degenerate on $\mathcal{X}$. Then,*

$$\mathcal{H}_K^{\mathcal{F}} = \mathrm{H}_K^{\mathcal{F}}.$$

Now, suppose that we are given a dataset $\{(x_i, y_i)\}_{i=1}^N \subset \mathcal{X} \times \mathbb{R}$. We now introduce the conditional Kernel Ridge Regression problem, defined as the minimization of the functional

$$J(f) = \frac{1}{N} \sum_{i=1}^N (f(x_i) - y_i)^2 + \lambda \|f\|_{\mathcal{H}_K^{\mathcal{F}}}^2, \tag{1}$$

over all $f \in \mathcal{H}_K^{\mathcal{F}}$. The role of the empirical residual kernel is demonstrated by the following theorem, which establishes a connection between conditional KRR and standard KRR for PD kernels.

**Theorem 3.2.** *Let $P = \frac{1}{N}\sum_{i=1}^{N}\delta_{x_i}$ and let $\mathcal{H}_{K_P}$ be the RKHS of $K_P$. Assuming that $F = [f_i(x_j)]_{i=1}^{k}{}_{j=1}^{N} \in \mathbb{R}^{k \times N}$ is of rank $k$, we have*

$$\min_{f \in \mathcal{H}_K^{\mathcal{F}}} \frac{1}{N}\sum_{i=1}^{N}(f(x_i) - y_i)^2 + \lambda\|f\|_{\mathcal{H}_K^{\mathcal{F}}}^2 =$$

$$\min_{g \in \mathcal{H}_{K_P}} \frac{1}{N}\sum_{i=1}^{N}(g(x_i) - r_i)^2 + \lambda\|g\|_{\mathcal{H}_{K_P}}^2,$$

*where $r = (r_1, \ldots, r_N)^\top \in \mathbb{R}^N$ is a projection of $y = (y_1, \ldots, y_N)^\top \in \mathbb{R}^N$ onto the orthogonal complement of the row space of $F$.*

*If $f^*$ is an optimal function of the first task then $g = (I - \Pi_P)f^*$ is an optimal function for the second task. Reversely, if $g^*$ is an optimal function for the second task, then*

$$f = g^* + [f_1(x), \cdots, f_k(x)](FF^\top)^{-1}Fy,$$

*is an optimal function for the first task.*

*Remark* 3.3. The intuition behind this theorem is as follows. Suppose we are given a set of features $f_1, \cdots, f_k$. For a training set $\{(x_i, y_i)\}_{i=1}^{N} \subset \mathcal{X} \times \mathbb{R}$, we first solve a standard linear regression problem with the model

$$Y = \beta_1 f_1(X) + \ldots + \beta_k f_k(X) + \varepsilon,$$

which amounts to projecting the target vector $y$ onto the row space of $F$. The remaining unexplained component of $y$ is the residual vector $r$. These residuals can then be predicted using KRR with the kernel $K_P$. The theorem shows that this two-step procedure is exactly equivalent to performing conditional KRR with the kernel $K$, which is CPD w.r.t. $\mathcal{F}$. Diagram 1 visualizes this scheme.

## 4. The $\mathcal{F}$-conditional learning and the cost of conditioning

Suppose $P$ is a distribution whose support is $\mathcal{X}$. Throughout this section, we additionally assume that there exists $N_0 \in \mathbb{N}$ such that for all $N \geq N_0$, if $X_1, \ldots, X_N$ are i.i.d. samples from $P$, then $\text{rank}([f_i(X_j)]_{i=1}^{k}{}_{j=1}^{N}) = k$ almost surely. The residual kernel w.r.t. $\mathcal{F} = \text{span}(f_1, \cdots, f_k)$, $K_P$, has a Mercer-type representation,

$$K_P(x, y) = \sum_{i=1}^{\infty}\lambda_i\phi_i(x)\phi_i(y),$$

Each $\phi_i$ belongs to $L_2(\mathcal{X}, P) \cap \mathcal{F}^\perp$, where the orthogonality is taken w.r.t. the inner product in $L_2(\mathcal{X}, P)$. Let $f$ be a function from $\mathcal{H}_K^{\mathcal{F}}$ which, by Theorem 3.1, can be written as $f = f_\parallel + f_\perp$, where

$$f_\parallel = \sum_{i=1}^{k}u_i f_i, \quad f_\perp = \sum_{i=1}^{\infty}v_i\sqrt{\lambda_i}\phi_i.$$

By construction, $\|f\|_{\mathcal{H}_K^{\mathcal{F}}}^2 = \sum_{i=1}^{\infty}v_i^2$.

Let $P_{X,Y}$ be a distribution on $\mathcal{X} \times \mathbb{R}$ defined by

$$(X, Y) \sim P_{X,Y} \Leftrightarrow X \sim P, Y = f(X) + \tilde{\varepsilon},$$

where $\tilde{\varepsilon} \sim \mathcal{N}(0, \sigma^2)$ is independent of $X$. Pairs of the training set $\mathcal{T} = \{(X_1, Y_1), \cdots, (X_N, Y_N)\}$ are generated independently from $P_{X,Y}$, i.e. $Y_i = f(X_i) + \varepsilon_i$. To the latter training set one can relate another training set (called residual), $\mathcal{T}_{\text{res}} = \{(X_i, Y_i^\perp)\}_{i=1}^{N}$, where $Y_i^\perp = f_\perp(X_i) + \varepsilon_i$. The corresponding distribution over input–output pairs is denoted by $P_{X,Y}^\perp$. Note that the noise term is included as part of the residual training set.

We now outline the idea of $\mathcal{F}$-conditional learning. Suppose that, prior to learning the target mapping $f$ from the dataset $\mathcal{T}$, the learner has full access to the component $f_\parallel$. The learner can then construct the residual dataset $\mathcal{T}_{\text{res}}$ by defining $Y_i^\perp = Y_i - f_\parallel(X_i)$. Next, KRR with the residual kernel $K_P$ is applied to $\mathcal{T}_{\text{res}}$, yielding an estimator $h$ of the residual function $f_\perp$, i.e.

$$h = \arg\min_{g \in \mathcal{H}_{K_P}} \frac{1}{N}\sum_{i=1}^{N}(g(X_i) - Y_i^\perp)^2 + \lambda\|g\|_{\mathcal{H}_{K_P}}^2.$$

Suppose that $\hat{f} = \sum_{i=1}^{\infty}\hat{v}_i\sqrt{\lambda_i}\phi_i + \sum_{i=1}^{k}\hat{u}_i f_i$ is an argument at which (1) attains its minimum, i.e. $\hat{f} = \arg\min_{g \in \mathcal{H}_K^{\mathcal{F}}} \frac{1}{N}\sum_{i=1}^{N}(g(X_i) - Y_i)^2 + \lambda\|g\|_{\mathcal{H}_K^{\mathcal{F}}}^2$. For the trained function $\hat{f}$ one can define $\hat{f}_\perp = \sum_{i=1}^{\infty}\hat{v}_i\sqrt{\lambda_i}\phi_i$ and $\hat{f}_\parallel = \sum_{i=1}^{k}\hat{u}_i f_i$. Due to Theorem 3.2, it is natural to expect that $\hat{f}_\perp \approx h$ and $\hat{f}_\parallel \approx f_\parallel$. The discrepancy between $\hat{f}$, obtained without access to $f_\parallel$, and $f_\parallel + h$, produced by an $\mathcal{F}$-conditional learner, can be naturally interpreted as the cost of conditioning.

**Definition 4.1.** The difference

$$c_{\text{con}} = \mathbb{E}[(\hat{f}(X) - f_\parallel(X) - h(X))^2] =$$
$$\|\hat{f}_\perp - h\|_{L_2(\mathcal{X}, P)}^2 + \|\hat{f}_\parallel - f_\parallel\|_{L_2(\mathcal{X}, P)}^2 \tag{2}$$

is referred to as the cost of conditioning. Note that $c_{\text{con}}$ is a random variable, depending on $X_1, \cdots, X_N$ and the noise.

**Theorem 4.2.** *Suppose that $f_1, \cdots, f_k$ are orthogonal functions of unit norm in $L_2(\mathcal{X}, P)$, $k \geq 1$. With probability at least $1 - \delta$ over randomness in $X_1, \cdots, X_N$, we have*

$$\mathbb{E}_\varepsilon[c_{\text{con}}] \leq$$

$$c_1\|f\|_{\mathcal{H}_K^{\mathcal{F}}}^2 C_{K_P}^2 \max_{j:1 \leq j \leq k}\|f_j\|_{L_\infty(\mathcal{X})}^2 \frac{k\log^{1/2}(\frac{2k}{\delta})}{N^{1/2}} + \frac{c_2\sigma^2}{N},$$

*where*

$$C_{K_P} = \sqrt{\max_x K_P(x,x)},$$

$$c_1 = 32\sqrt{2}\big(2 + 3\lambda_1\big(\frac{7C_{K_P}}{\lambda} + \frac{343C_{K_P}^3}{\lambda^2}\big)^2\big),$$

$$c_2 = \frac{9\lambda_1 C_{K_P}^2}{\lambda^2}\big(\frac{C_{K_P}^2}{\lambda} + 1\big)^2 + 2k,$$

*and provided that* $N \geq \max\big(\big(\frac{28}{3}k \max_{j:1\leq j\leq k}\|f_j\|_{L_\infty(\mathcal{X})}^2 + \frac{4}{3}\big)\log(\frac{4k}{\delta}), k^2\log(\frac{2k}{\delta})\max_{j:1\leq j\leq k}\|f_j\|_{L_\infty(\mathcal{X})}^4\big)$ *and* $N \geq N_0$.

For fixed $\lambda \neq 0$, the second term behaves as $\mathcal{O}(\frac{\sigma^2(k+1)}{N})$, which matches the decay rate of the expected loss in linear regression with $k$ features. When the signal part of the output lies entirely in $\mathcal{F}$, i.e. $f_\perp = 0$ and $\|f\|_{\mathcal{H}_K^\mathcal{F}} = 0$, the first term in the inequality vanishes. In this case, conditional KRR yields $\hat{f}$ such that $\mathbb{E}_\varepsilon[\|\hat{f}_\perp - h\|_{L_2(\mathcal{X},P)}^2] = \mathcal{O}(\frac{\sigma^2(k+1)}{N})$, $\mathbb{E}_\varepsilon[\|\hat{f}_\| - f_\|\|_{L_2(\mathcal{X},P)}^2] = \mathcal{O}(\frac{\sigma^2(k+1)}{N})$. That is, $\hat{f}_\|$ recovers the signal $f$ with the accuracy of linear regression, while $\hat{f}_\perp$ is $\mathcal{O}(\frac{\sigma^2(k+1)}{N})$-close to $h$, the output of KRR with residual kernel $K_P$. In other words, noise can make a substantial contribution (beyond $\mathcal{O}(\frac{\sigma^2(k+1)}{N})$) only to the component orthogonal to $\mathcal{F}$, and hence orthogonal to the signal $f$. Unlike linear regression, however, $\mathcal{F}$-conditional learning is capable of memorizing the noise in the training set. This effect may be described as *weak benign overfitting*. Moreover, if the eigenvalues of the residual kernel $K_P$ decay as $\lambda_i \sim \frac{1}{i\log^\alpha i}$ with $\alpha > 1$, then $h \to 0$ as $N \to \infty$. In this regime, the learner exhibits partial memorization of the training set without degrading the error loss, a phenomenon known simply as *benign overfitting* (Mallinar et al., 2022).

Finally, toy experiments reported in Section 6 suggest that the cost of conditioning typically decays as $\sim \frac{1}{N}$, even when $f \notin \mathcal{F}$. In contrast, our theoretical bound permits an additional contribution arising from a nontrivial component $f_\perp$, which may decay only at the slower rate $\sim \frac{1}{\sqrt{N}}$. In our experiments, such slower behavior was observed only for artificial kernels, such as $K(x,y) = \sum_{i=0}^{300}\cos(i(x-y))$.

## 5. Applications of Theorem 4.2

### 5.1. $\mathcal{F}$-conditioning with $k$ principal eigenfunctions: hard thresholding

Suppose the initial kernel $K$ is positive definite, i.e.

$$K(x,y) = \sum_{i=1}^\infty \lambda_i\phi_i(x)\phi_i(y),$$

where $\{\lambda_i\}$ are strictly positive eigenvalues and $\{\phi_i\}$ are the corresponding eigenfunctions of the integral operator $\phi \to \int_\mathcal{X} K(\cdot,x)\phi(x)dP(x)$ acting on $L_2(\mathcal{X},P)$. Let us treat $K$

as a CPD kernel w.r.t. $\Phi_k = \text{span}(\{\phi_1,\cdots,\phi_k\})$ and study the task (1). Thus, the set of unpenalized features coincides with first $k$ eigenfunctions of $K$. Then, the residual kernel w.r.t. to $\Phi_k$ is simply the tail part of $K$, i.e.

$$K_P(x,y) = \sum_{i=k+1}^\infty \lambda_i\phi_i(x)\phi_i(y).$$

Following the formalism of the previous section, let us now assume that the regression function has the form:

$$f = \sum_{i=1}^k u_i\phi_i.$$

As shown in the previous section, with probability at least $1 - \delta$ over the randomness in the inputs, conditional KRR with a CPD kernel $K$ (w.r.t. $\Phi_k$) and a regularization parameter $\lambda > 0$ can be interpreted as standard KRR with the residual kernel $K_P$ applied to the residual dataset $\{(X_i,\varepsilon_i)\}_{i=1}^N$ (which now consists solely of noise, since $f \in \Phi_k$). The only difference is the presence of an additional conditioning cost, bounded by $\mathcal{O}(\frac{\sigma^2(k+1)}{N})$, which contributes to the test error (noting that, by construction, $\|f\|_{\mathcal{H}_K^\mathcal{F}} = 0$).

Let $\varkappa > 0$ be such that $\sum_{i=1}^\infty \frac{\lambda_i}{\lambda_i + \varkappa} + \frac{\lambda}{\varkappa} = N$, and let

$$\mathcal{E} = \mathcal{E}_{\text{noise}}\big(\sum_{i=1}^\infty (1 - \mathcal{L}_i)^2 u_i^2 + \sigma^2\big),$$

where $\mathcal{L}_i = \frac{\lambda_i}{\lambda_i + \varkappa}$ denotes the learnability of the mode $\phi_i$, and $\mathcal{E}_{\text{noise}} = \frac{N}{N - \sum_{i=1}^\infty \mathcal{L}_i^2}$ is the overfitting coefficient. According to (Simon et al., 2023), the expected error of KRR with the kernel $K$ approximately equals $\mathcal{E}$. Analogously, let $\varkappa' > 0$ be such that $\sum_{i=k+1}^\infty \frac{\lambda_i}{\lambda_i + \varkappa'} + \frac{\lambda}{\varkappa'} = N$ and $\mathcal{L}_i' = \frac{\lambda_i}{\lambda_i + \varkappa'}$, $\mathcal{E}_{\text{noise}}' = \frac{N}{N - \sum_{i=k+1}^\infty (\mathcal{L}_i')^2}$. Then, the output of KRR with the residual kernel $K_P$ has the expected error of approximately $\mathcal{E}' = \mathcal{E}_{\text{noise}}'\sigma^2$. To estimate the expected error of conditional KRR with the CPD kernel $K$ (w.r.t. $\Phi_k$), the loss $\mathcal{E}'$ must be augmented by the conditioning cost $\mathbb{E}[c_{\text{con}}] = \mathcal{O}(\frac{\sigma^2(k+1)}{N})$. Therefore, in order for the expected error of conditional KRR to be smaller than that of standard KRR (i.e., KRR with the PD kernel $K$ and no unpenalized features), the following condition must hold

$$0 > \mathbb{E}[c_{\text{con}}] + \mathcal{E}' - \mathcal{E} = \mathcal{E}_{\text{noise}}'\sigma^2 -$$
$$- \mathcal{E}_{\text{noise}}\big(\sum_{i=1}^k (1 - \mathcal{L}_i)^2 u_i^2 + \sigma^2\big) + \mathcal{O}(\frac{\sigma^2(k+1)}{N}),$$

or, equivalently,

$$\sum_{i=1}^k \frac{\varkappa^2 u_i^2}{(\lambda_i + \varkappa)^2} > \sigma^2\big(\frac{\mathcal{E}_{\text{noise}}'}{\mathcal{E}_{\text{noise}}} - 1\big) + \mathcal{O}(\frac{\sigma^2(k+1)}{N\mathcal{E}_{\text{noise}}}). \quad (3)$$

Note that the right-hand side of this inequality, as well as the coefficients $\frac{\varkappa^2}{(\lambda_i + \varkappa)^2}$ on the left-hand side, do not depend on the target function $f$. Hence, the inequality provides a sufficient condition on the coefficients of $f$ in the basis $\{\phi_i\}_{i=1}^k$ ensuring that conditional KRR outperforms standard KRR without unpenalized features (equivalently, that the expected test error is a U-shaped function of $k$). Our experiments confirm that the test error is often non-monotonic in $k$ (the number of unpenalized principal components) when the signal $f$ is sufficiently strong. In contrast, for pure-noise datasets ($f = 0$), the test MSE consistently increased with $k$ across all experiments. The corresponding experimental results are presented in Section 6.

### 5.2. $\mathcal{F}$-conditioning with $k$ random Gaussian features: soft thresholding

In what follows, we show that choosing $\mathcal{F} = \Phi_k = \mathrm{span}\{\phi_1, \ldots, \phi_k\}$ is closely related to defining $\mathcal{F}$ as $k$ random Gaussian features with the covariance function $K$. Recall that the kernel admits the Mercer decomposition $K(x, y) = \sum_{j=1}^\infty \lambda_j \phi_j(x)\phi_j(y)$ where $\{\phi_j\}_{j=1}^\infty \subset L_2(\mathcal{X}, P)$ forms an orthonormal system and the eigenvalues $\{\lambda_j\}$ are positive and decreasing. Let $\{f(\omega, x)\}_{x \in \mathcal{X}}$ denote a centered Gaussian random field with covariance function $K$. Using the Karhunen-Loéve representation of $f(\omega, x) \in L_2(\mathcal{X}, P)$, we have

$$f(\omega, x) = \sum_{j=1}^\infty \sqrt{\lambda_j} \xi_j(\omega)\phi_j(x),$$

where $\{\xi_j(\omega)\}_{j=1}^\infty \sim^{\mathrm{iid}} \mathcal{N}(0, 1)$.

Let us assume that $g_i(x) = f(\omega_i, x)$ for i.i.d. samples $\omega_1, \ldots, \omega_k$ and denote $\boldsymbol{\omega} = (\omega_1, \ldots, \omega_k)$. Thus, we have

$$g_i(x) = \sum_{j=1}^\infty \sqrt{\lambda_j} \xi_{ij}\phi_j(x),$$

where $\{\xi_{ij}\}_{i=1}^k{}_{j=1}^\infty \sim^{\mathrm{iid}} \mathcal{N}(0, 1)$. We now define $\mathcal{G}_k = \mathrm{span}(g_1, \cdots, g_k)$ and consider conditional KRR with the CPD kernel $K$ w.r.t. $\mathcal{G}_k$.

Let $K_P^{\boldsymbol{\omega}}$ be a residual kernel w.r.t. $\mathcal{G}_k$. Let us define

$$M_{\ell,m} = \sum_{i,j=1}^k \xi_{i\ell}(G^{-1})_{ij}\xi_{jm}.$$

where $G \in \mathbb{R}^{k \times k}$ is the Gram matrix with $G_{ij} = \langle g_i, g_j \rangle_{L_2(\mathcal{X}, P)} = \sum_{\ell=1}^\infty \lambda_\ell \xi_{i\ell}\xi_{j\ell}$. The kernel of the projection operator onto $\mathcal{G}_k$ in $L_2(\mathcal{X}, P)$ is

$$\Pi_k(x, y) =$$
$$\sum_{i,j=1}^k \Big(\sum_{\ell=1}^\infty \sqrt{\lambda_\ell}\xi_{i\ell}\phi_\ell(x)\Big)(G^{-1})_{ij}\Big(\sum_{m=1}^\infty \sqrt{\lambda_m}\xi_{jm}\phi_m(y)\Big).$$

After grouping terms we have

$$\Pi_k(x, y) =$$
$$\sum_{\ell,m=1}^\infty \sqrt{\lambda_\ell \lambda_m}\Big(\sum_{i,j=1}^k \xi_{i\ell}(G^{-1})_{ij}\xi_{jm}\Big)\phi_\ell(x)\phi_m(y) =$$
$$\sum_{\ell,m=1}^\infty \sqrt{\lambda_\ell \lambda_m}M_{\ell,m}\phi_\ell(x)\phi_m(y).$$

Since $K_P^{\boldsymbol{\omega}} = (I - \Pi_k) \otimes (I - \Pi_k)[K]$, the dependence of $K_P^{\boldsymbol{\omega}}$ on $\boldsymbol{\omega}$ is encapsulated in coefficients $M_{\ell,m}$. Diagonal elements of the random residual kernel can be given as

$$\langle \phi_i, \mathbb{E}_{Y \sim P} K_P^{\boldsymbol{\omega}}(\cdot, Y)\phi_i(Y)\rangle_{L_2(\mathcal{X}, P)} =$$
$$\lambda_i\Big(1 - 2\lambda_i M_{i,i} + \sum_{j=1}^\infty \lambda_j^2 M_{i,j}^2\Big).$$

*Remark* 5.1. In a slightly different context, the coefficients $M_{\ell,m}$ were analyzed in Appendix C.3.1 of (Jacot et al., 2020b) (see also Subsection I.7 of (Simon et al., 2023)), under the assumption that $k$ is large, corresponding to the so-called "thermodynamic limit". We have $G = \sum_{i=1}^\infty \lambda_i \xi_i \xi_i^\top$, where $\xi_i \sim \mathcal{N}(\mathbf{0}, I_k)$ are generated independently. Let $G_{-i} = \sum_{j:j \neq i} \lambda_j \xi_j \xi_j^\top$, that is $G = \lambda_i \xi_i \xi_i^\top + G_{-i}$. Then the Sherman-Morrison formula gives

$$G^{-1} = G_{-i}^{-1} - \frac{\lambda_i G_{-i}^{-1} \xi_i \xi_i^\top G_{-i}^{-1}}{1 + \lambda_i \xi_i^\top G_{-i}^{-1} \xi_i},$$

and, therefore,

$$M_{i,j} = \xi_i^\top G^{-1}\xi_j =$$
$$\xi_i^\top G_{-i}^{-1}\xi_j - \frac{\lambda_i \xi_i^\top G_{-i}^{-1}\xi_i \cdot \xi_i^\top G_{-i}^{-1}\xi_j}{1 + \lambda_i \xi_i^\top G_{-i}^{-1}\xi_i} = \frac{\xi_i^\top G_{-i}^{-1}\xi_j}{1 + \lambda_i \xi_i^\top G_{-i}^{-1}\xi_i}.$$

As shown in (Simon et al., 2023), the quantity $\xi_i^\top G_{-i}^{-1}\xi_i$ concentrates sharply around its mean as $k \to \infty$, and moreover $\mathbb{E}[\xi_i^\top G_{-i}^{-1}\xi_i] \approx \mathbb{E}[\xi_j^\top G_{-j}^{-1}\xi_j]$. The off-diagonal coefficients, i.e. $M_{i,j}, i \neq j$, concentrate sharply around zero. To analyze the effect of $k$ unpenalized random Gaussian features, it suffices to study the structure of $\mathbb{E}_{\boldsymbol{\omega}}[K_P^{\boldsymbol{\omega}}(x, y)]$, which is the subject of the next theorem.

**Theorem 5.2.** *The expectation of $K_P^{\boldsymbol{\omega}}$ over randomness in $\boldsymbol{\omega} = (\omega_1, \cdots, \omega_k)$, i.e. $\mathbb{E}_{\boldsymbol{\omega}}[K_P^{\boldsymbol{\omega}}(x, y)]$, is a Mercer kernel that is equal to*

$$\sum_{i=1}^\infty \mu_i \phi_i(x)\phi_i(y),$$

*where $\mu_i = \lambda_i(1 - 2\lambda_i \cdot \mathbb{E}[M_{i,i}] + \sum_{j=1}^\infty \lambda_j^2 \cdot \mathbb{E}[M_{i,j}^2])$.*

To analyze the behavior of $\frac{\mu_i}{\lambda_i}$, we need to estimate the quantity $1 - 2\lambda_i \mathbb{E}[M_{i,i}] + \sum_{j=1}^\infty \lambda_j^2 \mathbb{E}[M_{i,j}^2]$, which again,

turns out to be tractable in the thermodynamic limit. In Appendix F we provide a non-rigorous argument (supported by experiments) showing that, as $k \to \infty$, the following approximation holds:

$$\frac{\mu_i}{\lambda_i} \approx \frac{c\varkappa^2}{(\lambda_i + \varkappa)^2}, \tag{4}$$

where $\varkappa > 0$ satisfies $\sum_{i=1}^{\infty} \frac{\lambda_i}{\lambda_i + \varkappa} = k$. The interpretation of the latter estimate is straightforward: when $\lambda_i \gg \varkappa$, the $i$-th mode is strongly suppressed in the residual kernel, while for $\lambda_i \ll \varkappa$, the corresponding eigenvalue is amplified by some factor $c$. Our numerical experiments (see Figure 6 in Appendix F) confirm that this behavior persists even for finite $k$. This shows that defining $\mathcal{F}$ via $k$ random Gaussian features has a similar effect to choosing $\mathcal{F}$ as the top $k$ eigenfunctions: in both cases the residual kernel $K_P^{\omega}(x, y)$ resembles a truncated kernel, but with the suppression of large eigenvalues applied in a soft manner. For this reason, it is natural to refer to these two approaches as *soft thresholding* and *hard thresholding*, respectively.

We expect that the theoretical prediction of a U-shaped dependence of the expected error on $k$ should also hold for soft thresholding, just as it does for hard thresholding (under conditions analogous to formula (3)). Our experiments, reported in Section 6, confirm that the non-monotone dependence of the expected test error on $k$ commonly arises in this setting as well.

$\mathcal{F}$**-conditioning with $k$ random features**. Let us assume that $K$ is a Mercer kernel that is given through the random features mapping $f : \Omega \times \mathcal{X} \to \mathbb{R}$ and $(\Omega, \Sigma, \mathcal{P})$ is a probabilistic space, that is $K(x, y) = \mathbb{E}_{\omega \sim \mathcal{P}}[f(\omega, x)f(\omega, y)]$. Since $K$ is a positive definite kernel, it is in particular CPD w.r.t. any subspace $\mathcal{F} = \text{span}(f_1, \ldots, f_k)$. Given a dataset $\{(x_i, y_i)\}_{i=1}^N \subset \mathcal{X} \times \mathbb{R}$, we consider the conditional KRR problem w.r.t. $\mathcal{F}$, namely the optimization task (1).

We conducted experiments with $\mathcal{F} = \mathcal{R}_k$, where $\mathcal{R}_k = \text{span}\{g_1, \ldots, g_k\}$ and $g_i(x) = f(\omega_i, x)$ for i.i.d. samples $\omega_1, \ldots, \omega_k \sim \mathcal{P}$. When $\{f(\omega, x)\}_{x \in \mathcal{X}}$ is a Gaussian random field, this setup coincides with the soft thresholding framework. Hence, it can be seen as a generalization of soft thresholding to the case of non-Gaussian features. Prior work (Louart et al., 2018; Benigni & Péché, 2021) has shown that general random feature models behave similarly to the Gaussian case, and thus we expect the U-shaped dependence of the expected risk on $k$ to be a generic phenomenon here as well. This hypothesis is verified experimentally in the next section and Appendix H.3.

## 6. Experiments

**Hard thresholding (synthetic data case)**. To examine the cost of conditioning and the U-shaped dependence of the test

risk on the number of unpenalized principal eigenfunctions in the hard-thresholding case, predicted theoretically by inequality (3), we carried out a toy experiment. On the domain $\mathcal{X} = [0, 2\pi]$ with the uniform input distribution, we consider the kernel

$$K(x, y) = 1 + \sum_{i=1}^{\infty} i^{-2s} \big( \cos(ix)\cos(iy) + \sin(ix)\sin(iy) \big),$$

parameterized by a smoothness parameter $s > 0$. For a fixed parameter $k$, the set of unpenalized features $\mathcal{F}$ is defined as $\text{span}\big(\{\cos(ix)\}_{i=0}^k \cup \{\sin(ix)\}_{i=1}^k\big)$.

The dependence of $\hat{c}_{\text{con}}$ on the parameters $N, k$, and $\sigma^2$ for various target functions is shown in Figure 2. The plots for $k$ and $\sigma^2$ exhibit linear trends fully consistent with the predictions of Theorem 4.2. Across all experiments, we observed a decay rate of $\hat{c}_{\text{con}} \sim \frac{1}{N}$ as $N$ increases. In contrast, the upper bound of Theorem 4.2 scales as $\frac{1}{\sqrt{N}}$ whenever $\|f\|_{\mathcal{H}_K^{\mathcal{F}}} \neq 0$ or $f \notin \mathcal{F}$. Whether the faster $\frac{1}{N}$ decay is a general property of the hard thresholding setting, or merely a peculiarity of our experiments, remains an open theoretical question. For the regression function $f(x) = \sum_{n=0}^5 n\cos(nx)$ the resulting U-shaped behavior of the test error as a function of $k$ is illustrated in Figure 3. The last plot on Figure 3 shows test-MSE curves (with 95% confidence intervals) for varying regularization parameter $\lambda$ in two representative cases: $k = 0$ (standard KRR) and $k = 5$, where the regression function is $f(x) = \sum_{n=0}^5 n\cos(nx)$. Taken together, these results suggest that adjusting the number of unpenalized features within the hard-thresholding framework can be beneficial for essentially any value of $\lambda$, including the value optimally tuned for standard KRR.

**Hard thresholding (real world data case).** As an illustrative example, we used 12214 samples of the digits 7 and 9 from the MNIST training set. Each image was cropped to a $24 \times 24$ window by removing border pixels. We assigned the label $+1$ to digit 7 and $-1$ to digit 9. Since the eigenfunctions $\phi_i$ are not available in closed form for non-synthetic data, we estimated them from samples. We conducted numerical experiments using a certain estimator $\widehat{\phi}_i$ of $\phi_i$ in the hard-thresholding setup (whose description can be found in subsection H.2 of Appendix) on the 7-vs-9 MNIST dataset. We performed 10-fold cross-validation to evaluate the Conditional KRR model for a fixed parameter $\lambda = 0.01$ and a range of sizes $k$. In each fold, the data were re-standardized, Conditional KRR was fitted on the training subset and its test MSE was recorded for all $k$, after which the mean test MSE and its 95% confidence interval across folds were computed. Results for different kernels are shown on Figure 4.

As shown in the results, both the Gaussian and NNGP-erf kernels exhibit a U-shaped dependence of the test MSE on $k$.

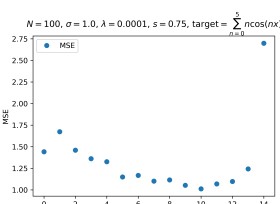

*Figure 2.* Dependence of the cost of conditioning on $N$, $k$ and $\sigma^2$ in the hard thresholding setting.

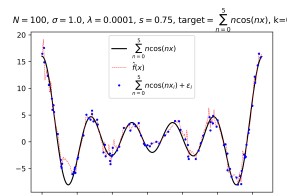
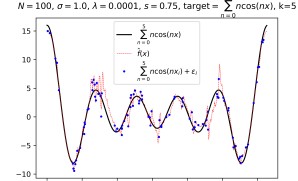
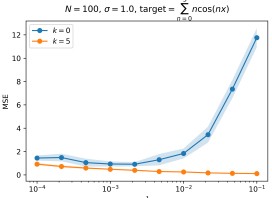

*Figure 3.* Comparison of test MSE for Conditional KRR with $k = 0$ (standard KRR) and $k = 5$. The last plot compares test MSEs across a range of regularization parameters $\lambda$.

In the case of the NNGP-erf kernel, however, overfitting is very mild and becomes noticeable only when $k$ approaches the size of the training set (approximately 11,000, the largest value for which $\widehat{\phi}_k(x)$ is defined). Interestingly, no overfitting is observed for the Laplace kernel. We attribute this behavior to the fact that, for the Laplace kernel, essentially all of the first 11,000 empirical eigenfunctions remain informative for prediction; detecting overfitting would require a substantially larger sample size to allow $\widehat{\phi}_k(x)$ to be defined for larger $k$.

**Experiments on soft thresholding with random features.** We also conducted experiments on $\mathcal{F}$-conditioning with $k$ random features (see the discussion in the end of subsection 5.2). In this setup, we worked directly with random feature representations rather than explicitly computing the kernel $K$. As shown in Appendix G, conditional KRR in this setting can be approximated by ridge regression with two types of random features: a large set of penalized features and $k$ unpenalized ones. We considered three activation functions: $\cos(x)$, $\mathrm{ReLU}(x)$, and $\tanh(x)$. In each case, a random field on $\mathcal{X} = \mathbb{S}^{d-1}$ with covariance $K$ was defined as follows:

(a) $f(x, [\omega, b]) = \cos(\omega^\top x + b)$ with $\omega \sim \mathcal{N}(\mathbf{0}, I_d)$ and $b \sim U([0, 2\pi])$;

(b) $f(x, [\omega, b]) = \mathrm{ReLU}(\omega^\top x + b)$ with $\omega \sim \mathcal{N}(\mathbf{0}, I_d)$ and $b \sim U([-1, 1])$;

(c) $f(x, [\omega, b]) = \tanh(\omega^\top x + b)$ with $\omega \sim \mathcal{N}(\mathbf{0}, I_d)$ and $b \sim U([-1, 1])$.

Note that in case (a), $K$ corresponds to the Gaussian kernel. The U-shaped dependence of the expected risk for all three cases for the target function $f(x_1, \cdots, x_d) = \sin(x_1) +$ $\frac{1}{2}\cos(x_2)$ is shown in Figure 5. Experiments on real world data can be found in subsection H.3 of Appendix.

Details of described experiments, together with additional experiments, are provided in Appendix H.

## 7. Conclusions and open problems

We have developed a statistical theory of learning with conditional KRR and applied it to both hard and soft thresholding settings. Attempting to study the memorization phenomenon in conditional KRR encounters an immediate difficulty: all of our bounds require the regularization parameter $\lambda \neq 0$, whereas perfect memorization of the training set is possible only when $\lambda = 0$. Extending our statistical analysis to cover this latter case remains an open direction for future research.

## Impact Statement

This paper develops a theoretical framework for conditional kernel ridge regression, a learning method that combines explicit feature representations with kernel-based learning. The main contribution is mathematical: we analyze how incorporating unpenalized features can improve prediction accuracy and alter the statistical behavior of kernel methods. The proposed framework may be useful for designing more efficient and interpretable machine learning systems, particularly in scientific computing, approximation theory, and high-dimensional data analysis.

The work is primarily theoretical and does not introduce a deployed system, dataset, or application targeting individuals or social groups. As such, we do not anticipate direct negative societal consequences arising specifically from this research.

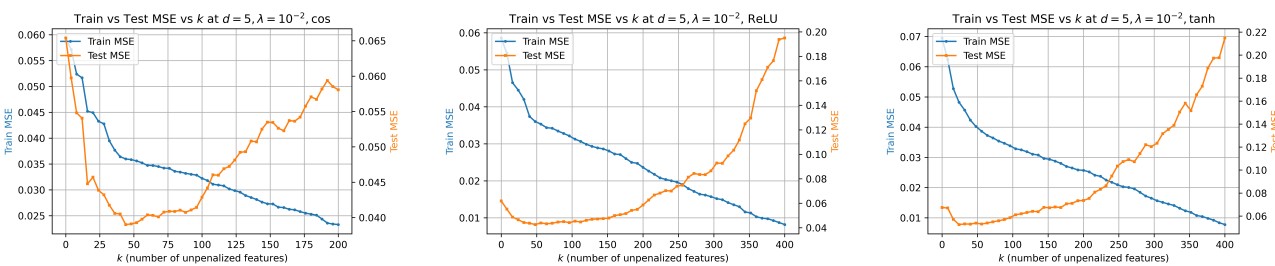

*Figure 4.* U-shaped Test MSE in the hard thresholding setup for the 7-vs-9 MNIST dataset (with standardization).

*Figure 5.* The effect of the soft thresholding for the cosine, ReLU and tanh activation functions and the regression function $f(x_1, \cdots, x_d) = \sin(x_1) + \frac{1}{2}\cos(x_2)$.

More broadly, this work contributes to the theoretical understanding of modern machine learning methods. Improved understanding of learning algorithms may support the development of more reliable, interpretable, and data-efficient AI systems.

## Acknowledgements

This research has been funded by the Science Committee of the Ministry of Science and Higher Education of the Republic of Kazakhstan (Grant No. AP27510283), PI R. Takhanov.

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

## A. Proof of Theorem 2.1

Let $\mathcal{M}(\mathcal{X})$ be the set of finite signed Borel measures on $\mathcal{X}$. The following characterization of CPD requires only standard argumentation.

**Lemma A.1.** *$K$ is CPD w.r.t. $\mathcal{F}$ if and only if for all finite signed Borel measures $\mu \in \mathcal{M}(\mathcal{X})$ satisfying $\int f(x)\,d\mu(x) = 0$ for all $f \in \mathcal{F}$, we have:*

$$\iint K(x, x')\,d\mu(x)\,d\mu(x') \geq 0.$$

Let us prove that $K_P(x, y)$ is a PD kernel, i.e. that

$$\iint K_P(x, y)\,d\mu(x)\,d\mu(y) \geq 0$$

for any $\mu \in \mathcal{M}(\mathcal{X})$. We define $\nu = (I - \Pi_P)^* \mu \in \mathcal{M}(\mathcal{X})$ as the unique signed measure satisfying

$$\int f(x)\,d\nu(x) = \int (I - \Pi_P) f(x)\,d\mu(x) \quad \text{for all } f \in C(\mathcal{X}).$$

From Riesz-Markov-Kakutani representation theorem we obtain that $\nu$ is a finite signed Borel measure. For every $f \in \mathcal{F}$, we have $(I - \Pi_P)f = 0$ due to nondegeneracy of $P$. Thus, we have

$$\int f(x)\,d\nu(x) = \int (I - \Pi_P) f(x)\,d\mu(x) = 0.$$

From the definition of $K_P$, we have $K_P = (I - \Pi_P)\left[(I - \Pi_P)\left[K(x, \cdot)\right](\cdot, y)\right]$, so for any $\mu \in \mathcal{M}(\mathcal{X})$,

$$\iint K_P(x, y)\,d\mu(x)\,d\mu(y) = \int \left[ \int (I - \Pi_P)\left[(I - \Pi_P)\left[K(x, \cdot)\right](\cdot, y)\right](x, y)\,d\mu(x) \right] d\mu(y) =$$

$$\int \left[ \int (I - \Pi_P)\left[K(x, \cdot)\right](x, y)\,d\nu(x) \right] d\mu(y) = \int \left[ \int (I - \Pi_P)\left[K(x, \cdot)\right](x, y)\,d\mu(y) \right] d\nu(x)$$

$$= \int \left[ \int K(x, y)\,d\nu(y) \right] d\nu(x) = \iint K(x, y)\,d\nu(x)d\nu(y).$$

The latter expression is non-negative due to conditional positive definiteness of $K$. Therefore, $K_P(x, y)$ is positive definite.

## B. Proof of Theorem 3.1

Define $\mathrm{O}_K^P : L_2(\mathcal{X}, P) \to L_2(\mathcal{X}, P)$ by

$$\mathrm{O}_K^P f(x) = \int_{\mathcal{X}} K_P(x, y) f(y) dP(y).$$

By Mercer's theorem (Takhanov, 2023), the kernel $K_P$ can be expanded as

$$K_P(x, y) = \sum_{i=1}^{\infty} \lambda_i \phi_i(x) \phi_i(y),$$

where $\{\lambda_i\}$ are non-zero eigenvalues of $\mathrm{O}_K^P$ and $\{\phi_i\}$ are corresponding orthogonal eigenfunctions of unit length. Note that $\phi_i \in L_2(\mathcal{X}, P) \cap \mathcal{F}^{\perp}$ due to $\mathrm{O}_K^P[L_2(\mathcal{X}, P) \cap \mathcal{F}^{\perp}] \subseteq L_2(\mathcal{X}, P) \cap \mathcal{F}^{\perp}$ and $\mathrm{O}_K^P[\mathcal{F}] = \{0\}$. It is well-known (see Theorem 4 from (Cucker & Smale, 2001)) that $\mathcal{H}_{K_P}$ is a Hilbert space with a set of functions $\sqrt{\mathrm{O}_K^P}[L_2(\mathcal{X}, P)]$, i.e. the set of functions of the form

$$\sum_{i=1}^{\infty} \sqrt{\lambda_i} x_i \phi_i$$

where $[x_i]_{i=1}^\infty \in l^2(\mathbb{N})$. For $f = \sum_{i=1}^\infty \sqrt{\lambda_i} x_i \phi_i$ and $g = \sum_{i=1}^\infty \sqrt{\lambda_i} y_i \phi_i$, the inner product on $\mathcal{H}_{K_P}$ equals

$$\langle f, g \rangle_{\mathcal{H}_{K_P}} = \sum_{i=1}^\infty x_i y_i.$$

First let us prove that $\mathcal{L} \subseteq \mathrm{H}_K^{\mathcal{F}}$ which will directly imply $\mathcal{L} \oplus \mathcal{F} \subseteq \mathrm{H}_K^{\mathcal{F}}$. Let $f \in \mathcal{L}$ and

$$f = \sum_{i=1}^n \alpha_i K(x_i, \cdot) \text{ such that } \sum_{i=1}^n \alpha_i f_j(x_i) = 0 \text{ for all } j = 1, \ldots, k.$$

Since $\sum_{i=1}^n \alpha_i f_j(x_i) = 0$, then $(I - \Pi_P)^* \sum_{i=1}^n \alpha_i \delta_{x_i} = \sum_{i=1}^n \alpha_i \delta_{x_i}$. Therefore, the function $f$ can be expressed as

$$f(x) = \int_{\mathcal{X}} K(y, x) d(\sum_{i=1}^n \alpha_i \delta_{x_i})(y) = \int_{\mathcal{X}} K(y, x) d((I - \Pi_P)^* \sum_{i=1}^n \alpha_i \delta_{x_i})(y) =$$

$$\int_{\mathcal{X}} (I - \Pi_P)[K(\cdot, z_2)](y, x) d(\sum_{i=1}^n \alpha_i \delta_{x_i})(y) = \sum_{i=1}^n \alpha_i (I - \Pi_P)[K(\cdot, z_2)](x_i, x) =$$

$$\sum_{i=1}^n \alpha_i (I - \Pi_P)[(I - \Pi_P)[K(\cdot, z_2)](z_1, \cdot)](x_i, x) + \alpha_i \Pi_P[(I - \Pi_P)[K(\cdot, z_2)](z_1, \cdot)](x_i, x).$$

Note that $\Pi_P[(I - \Pi_P)[K(\cdot, z_2)](z_1, \cdot)](x_i, x) \in \mathcal{F}$ and we obtained

$$f(x) = \sum_{i=1}^n \alpha_i K_P(x_i, x) + \tilde{f},$$

where $\tilde{f} \in \mathcal{F}$. Since

$$\sum_{i=1}^n \alpha_i K_P(x_i, x) = \sum_{i=1}^n \alpha_i \sum_{j=1}^\infty \lambda_j \phi_j(x_i) \phi_j(x) = \sum_{j=1}^\infty \sqrt{\lambda_j} (\sum_{i=1}^n \alpha_i \phi_j(x_i)) \sqrt{\lambda_j} \phi_j(x),$$

and

$$\|[\sqrt{\lambda_j}(\sum_{i=1}^n \alpha_i \phi_j(x_i))]_{j=1}^\infty\|_{l^2(\mathbb{N})}^2 = \sum_{j=1}^\infty \lambda_j \sum_{i,i'=1}^n \alpha_i \alpha_{i'} \phi_j(x_i) \phi_j(x_{i'}) = \sum \alpha_i \alpha_{i'} K(x_i, x_{i'}) < \infty$$

we conclude that $\sum_{i=1}^n \alpha_i K_P(x_i, x) \in \mathcal{H}_{K_P}$. Thus, we proved $f \in \mathrm{H}_K^{\mathcal{F}}$, and therefore, $\mathcal{L} \oplus \mathcal{F} \subseteq \mathrm{H}_K^{\mathcal{F}}$.

Let us now prove that for any $g \in \mathrm{H}_K^{\mathcal{F}}$ and the previous $f \in \mathcal{L}$ we have

$$\langle g, f \rangle_{\mathrm{H}_K^{\mathcal{F}}} = \sum_{i=1}^n \alpha_i g(x_i).$$

The latter property is an analog of the reproducing property of the kernel $K$ in the theory of RKHSs. By construction, we have $g = \sum_j r_j \sqrt{\lambda_j} \phi_j + \tilde{g}$ for $[r_i]_{i=1}^\infty \in l^2(\mathbb{N})$ and $\tilde{g} \in \mathcal{F}$. Thus,

$$\langle g, f \rangle_{\mathrm{H}_K^{\mathcal{F}}} = \sum_{j=1}^\infty r_j \sqrt{\lambda_j} (\sum_{i=1}^n \alpha_i \phi_j(x_i)) = \sum_{i=1}^n \alpha_i (g(x_i) - \tilde{g}(x_i)) = \sum_{i=1}^n \alpha_i g(x_i),$$

due to $\sum_{i=1}^n \alpha_i \tilde{g}(x_i) = 0$.

The inner product in $\mathcal{L}$ matches the inner product in $\mathrm{H}_K^{\mathcal{F}}$. Indeed, let $f, g \in \mathcal{L}$. In the previous analysis we established that

$$f = \sum_{j=1}^\infty p_j \sqrt{\lambda_j} \phi_j + \tilde{f}, g = \sum_{j=1}^\infty q_j \sqrt{\lambda_j} \phi_j + \tilde{g},$$

where $p_j = \sqrt{\lambda_j}(\sum_{i=1}^n \alpha_i \phi_j(x_i))$, $q_j = \sqrt{\lambda_j}(\sum_{i=1}^m \beta_i \phi_j(y_i))$ and $\tilde{f}, \tilde{g} \in \mathcal{F}$. Therefore,

$$\langle f, g \rangle_{\mathrm{H}_K^{\mathcal{F}}} = \sum_{j=1}^\infty p_j q_j = \sum_{j=1}^\infty \lambda_j (\sum_{i=1}^n \alpha_i \phi_j(x_i))(\sum_{i=1}^m \beta_i \phi_j(y_i)) = \sum_{i=1}^n \sum_{j=1}^m K(x_i, y_j) \alpha_i \beta_j.$$

So, we have $\langle f, g \rangle_{\mathcal{L}} = \langle f, g \rangle_{\mathrm{H}_K^{\mathcal{F}}}$. This implies to $\langle f, g \rangle_{\mathcal{H}_K^{\mathcal{F}}} = \langle f, g \rangle_{\mathrm{H}_K^{\mathcal{F}}}$ for any $f, g \in \mathcal{L} \oplus \mathcal{F}$.

To complete the proof we need to show that $\mathcal{L} \oplus \mathcal{F} \subseteq \mathrm{H}_K^{\mathcal{F}}$ is dense in $\mathrm{H}_K^{\mathcal{F}}$. The latter follows from the denseness of $(\mathcal{L} \oplus \mathcal{F}) \cap \mathcal{H}_{K_P}$ in $\mathcal{H}_{K_P}$. Indeed, let $f \in \mathcal{H}_{K_P}$ be orthogonal to all functions in $(\mathcal{L} \oplus \mathcal{F}) \cap \mathcal{H}_{K_P}$, then the previous analysis shows that it should be orthogonal to all functions from $\mathcal{L}$, i.e.

$$\langle f, \sum_{i=1}^n \alpha_i K(x_i, \cdot) \rangle_{\mathrm{H}_K^{\mathcal{F}}} = 0 \text{ whenever } \sum_{i=1}^n \alpha_i f_j(x_i) = 0 \text{ for all } j = 1, \ldots, k.$$

This implies

$$\sum_{i=1}^n \alpha_i f(x_i) = 0 \text{ whenever } \sum_{i=1}^n \alpha_i f_j(x_i) = 0 \text{ for all } j = 1, \ldots, k.$$

The latter implies $f \in \mathcal{F}$. Since $\mathcal{F} \cap \mathcal{H}_{K_P} = \{0\}$, we obtain $f = 0$. Theorem proved.

## C. Proof of Theorem 3.2

By the Representer Theorem (e.g. see Theorem 6.1 from (Auffray & Barbillon, 2009)), the solution $f^*$ of the initial task (1) has the form

$$f^*(x) = \sum_{i=1}^N \alpha_i K(x_i, x) + \sum_{j=1}^k \beta_j f_j(x),$$

where $\sum_i \alpha_i f_j(x_i) = 0$, $1 \leq j \leq k$, which leads us to the following optimization task

$$\min_{\alpha, \beta} \|\mathbf{K}\alpha + F^\top \beta - y\|^2 + \lambda \alpha^\top \mathbf{K}\alpha \quad \text{subject to} \quad F\alpha = 0,$$

where $y = (y_1, \ldots, y_N)^\top \in \mathbb{R}^N$, $\mathbf{K} = [K(x_i, x_j)]_{i,j=1}^N \in \mathbb{R}^{N \times N}$, $\alpha = (\alpha_1, \ldots, \alpha_N)^\top \in \mathbb{R}^N$ and $\beta = (\beta_1, \ldots, \beta_k)^\top \in \mathbb{R}^k$.

Since the matrix $FF^\top$ is invertible, the minimization over $\beta$ gives

$$\beta = -(FF^\top)^{-1}F(\mathbf{K}\alpha - y).$$

The matrix $\Pi = F^\top (FF^\top)^{-1}F$ corresponds to the projection operator onto the row space of $F$. Let us denote $r = (I_N - \Pi)y$, where $I_N = [\delta_{ij}]_{i,j=1}^N$. Note that $\alpha = (I_N - \Pi)\alpha$ due to $F\alpha = 0$. After we plug the expression for $\beta$ into the former objective, we obtain a new objective

$$\|(I_N - \Pi)(\mathbf{K}\alpha - y)\|^2 + \lambda \alpha^\top K\alpha =$$
$$\|(I_N - \Pi)\mathbf{K}(I_N - \Pi)\alpha - r\|^2 + \lambda \alpha^\top (I_N - \Pi)\mathbf{K}(I_N - \Pi)\alpha.$$

Let us denote $\tilde{\mathbf{K}} = (I_N - \Pi)\mathbf{K}(I_N - \Pi)$. We obtained the task

$$\min_\alpha \|\tilde{\mathbf{K}}\alpha - r\|^2 + \lambda \alpha^\top \tilde{\mathbf{K}}\alpha \quad \text{subject to} \quad F\alpha = 0.$$

For any $y \in \{x \in \mathbb{R}^N \mid Fx = 0\}^\perp = \mathrm{row}(F)$ we have $\tilde{\mathbf{K}}y = 0$. Therefore, the latter task is equivalent to solving the unconstrained

$$\min_{\alpha'} \|\tilde{\mathbf{K}}\alpha' - r\|^2 + \lambda \alpha'^\top \tilde{\mathbf{K}}\alpha',$$

and then setting $\alpha = (I_N - \Pi)\alpha'$. Further, let us denote solutions of the latter two tasks $\alpha$ and $\alpha'$ respectively. Note that $\tilde{\mathbf{K}}$ is the kernel matrix for the residual kernel function $K_P$. By Theorem 2.1, $K_P$ is positive semidefinite and the latter task leads to the KRR optimization task

$$\min_{g \in \mathcal{H}_{K_P}} \sum_{i=1}^N (g(x_i) - r_i)^2 + \lambda \|g\|_{\mathcal{H}_{K_P}}^2,$$

with the correspondence between solutions of the KRR and the previous one established by the rule

$$g^*(x) = \sum_{i=1}^{N} \alpha_i' K_P(x_i, \cdot).$$

Note that adding to $\alpha'$ any vector from $\text{row}(F)$ does not change $g^*$, therefore we can write $g^*(x) = \sum_{i=1}^{N} \alpha_i K_P(x_i, \cdot)$. Since $\alpha = (I_N - \Pi)\alpha$ and $[K_P(x_i, \cdot)]_{i=1}^{N} = (I_N - \Pi)[(I - \Pi_P)[K(x_i, \cdot)]]_{i=1}^{N}$, we obtain

$$g^*(x) = \sum_{i=1}^{N} \alpha_i (I - \Pi_P)[K(x, \cdot)](x_i, x)$$

That is, $g^* = (I - \Pi_P) f^*$.

Next, given $g^*$, let us recover $f^*$. We have

$$f^* = g^* + \Pi_P f^* = g^* + \Pi_P \left( \sum_{i=1}^{N} \alpha_i K(x_i, x) + \sum_{j=1}^{k} \beta_j f_j(x) \right) =$$

$$g^* + \Pi_P \left( \sum_{i=1}^{N} \alpha_i K(x_i, x) \right) + \beta^\top [f_1(x), \cdots, f_k(x)]^\top =$$

$$g^* + \alpha^\top \mathbf{K} F^\top (FF^\top)^{-1} [f_1(x), \cdots, f_k(x)]^\top + \beta^\top [f_1(x), \cdots, f_k(x)]^\top.$$

Using $\beta = -(FF^\top)^{-1} F(\mathbf{K}\alpha - y)$, we conclude

$$f^* = g^* + \Pi_P f^* = g^* + y^\top F^\top (FF^\top)^{-1} [f_1(x), \cdots, f_k(x)]^\top.$$

Theorem proved.

## D. Proof of Theorem 4.2

Let $l^2(\mathbb{N})$ denote the Hilbert space of sequences $[x_i]_{i=1}^{\infty}$ such that $\sum_i x_i^2 < \infty$ with the standard dot product of sequences. By $\mathcal{B}(A, B)$ we denote bounded linear operators between spaces $A$ and $B$. E.g., $\mathcal{B}(\mathbb{R}^N, l^2(\mathbb{N}))$ can be identifed with certain $\mathbb{N} \times N$ matrices.

### D.1. Expressions for transfer matrices

Following Section 4, let us introduce notations

$$u = [u_1, \cdots, u_k]^\top, v = [v_1, v_2, \cdots]^\top, \hat{u} = [\hat{u}_1, \cdots, \hat{u}_k]^\top, \hat{v} = [\hat{v}_1, \hat{v}_2, \cdots]^\top$$
$$y = [Y_1, \cdots, Y_N]^\top, \boldsymbol{\phi}_i = [\phi_i(X_1), \cdots, \phi_i(X_N)]^\top, \mathbf{f}_i = [f_i(X_1), \cdots, f_i(X_N)]^\top$$
$$\Phi = [\phi_i(X_j)]_{i=1,j=1}^{\infty,N}, F = [f_i(X_j)]_{i=1,j=1}^{k,N} \in \mathbb{R}^{k \times N}, \Lambda = [\lambda_i \delta_{ij}]_{i,j=1}^{\infty}.$$

Note that $y = \Phi^\top \Lambda^{1/2} v + F^\top u + \varepsilon$ where $\varepsilon \sim \mathcal{N}(\mathbf{0}, \sigma^2 I_N)$ is independent of $\mathcal{D}_N = (X_1, \cdots, X_N)$.

**Theorem D.1.** *Let $F$ be of rank $k$. For the $v$-part of the regression function we have $\hat{v} = T_\phi v + T_{\phi \varepsilon} \varepsilon$, with matrices $T_\phi$ and $T_{\phi \varepsilon}$ defined by*

$$T_\phi = \Lambda^{1/2} \Psi (\Psi^\top \Lambda \Psi + \lambda N I_N)^{-1} \Psi^\top \Lambda^{1/2},$$
$$T_{\phi \varepsilon} = \Lambda^{1/2} \Psi (\Psi^\top \Lambda \Psi + \lambda N I_N)^{-1} (I_N - F^\top (FF^\top)^{-1} F),$$

*where $\Psi = \Phi(I_N - F^\top (FF^\top)^{-1} F)$ and $I = [\delta_{ij}]_{i,j=1}^{\infty}$. For the $u$-part of the regression function we have $\hat{u} = u + T_f v + T_{f\varepsilon} \varepsilon$ where*

$$T_f = (FF^\top)^{-1} F \Phi^\top \Lambda^{1/2} (I - \Lambda^{1/2} \Psi (\Psi^\top \Lambda \Psi + \lambda N I_N)^{-1} \Psi^\top \Lambda^{1/2}),$$
$$T_{f\varepsilon} = (FF^\top)^{-1} F.$$

*Remark* D.2. The latter theorem claims that a linear relationships between coefficients of the regression function and the trained function can be described by the following diagram.

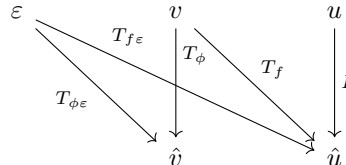

Note that $\hat{v}$ and $\hat{u} - u$ do not depend on $u$. This implies that for a fixed $u$, the distribution of $(\hat{f}(X) - f(X))^2$ for $X \sim P$ does not depend on $u$. Therefore, in a statistical analysis of this expression we may assume that $u = \mathbf{0}$.

The fact that $\hat{v}$ does not depend on $u$ follows from Theorem 3.2. Indeed, according to Remark 3.3 KRR with the CPD kernel can be understood as the two step process: the first step being the linear regression with features $f_1, \cdots, f_k$ and the second step being the KRR on residuals. The first step "erases" all correlations with $u$, i.e. the $\mathcal{F}$-part of the signal. That is why the part of the trained function that belongs to the RKHS of $K_P$ does not depend on $u$.

Further, given a kernel $\tilde{K}$, $\tilde{K}(x, \mathcal{D}_N)$ denotes the row $[\tilde{K}(x, X_1), \cdots, \tilde{K}(x, X_N)]$ and $\tilde{K}(\mathcal{D}_N, \mathcal{D}_N)$ denotes the matrix $[\tilde{K}(X_i, X_j)]_{i,j=1}^N$. We define $P_N = \frac{1}{N}\sum_{i=1}^N \delta_{X_i}$, i.e. the empirical measure. Also, in all lemmas below we assume that $F$ is of rank $k$, i.e. $P_N$ is $\mathcal{F}$-nondegenerate.

**Lemma D.3.** *We have* $\langle \phi_i, K_{P_N}(\cdot, \mathcal{D}_N)\rangle_{L_2(\mathcal{X}, P)} = \lambda_i \boldsymbol{\phi}_i^\top (I_N - F^\top (FF^\top)^{-1}F)$.

*Proof.* By construction, $\langle \phi_i, K_P(\cdot, y)\rangle_{L_2(\mathcal{X}, P)} = \lambda_i \phi_i(y)$. Let $\Pi(x, y) = \sum_{i,j=1}^k (G^{-1})_{ij} f_i(x) f_j(y)$ where $G = [G_{ij}]_{i,j=1}^k = [\langle f_i, f_j\rangle_{L_2(\mathcal{X}, P)}]_{i,j=1}^k$. The residual kernel equals

$$K_P(x, y) = K(x, y) - \mathbb{E}_{S \sim P}[K(x, S)\Pi(S, y)] - \mathbb{E}_{S \sim P}[\Pi(x, S)K(S, y)] + \mathbb{E}_{S, T \sim P}[\Pi(x, S)K(S, T)\Pi(T, y)].$$

Since $\langle \phi_i, f_j\rangle_{L_2(\mathcal{X}, P)} = 0$, we obtain

$$\langle \phi_i, K(\cdot, y) - \mathbb{E}_{S \sim P}[K(\cdot, S)\Pi(S, y)]\rangle_{L_2(\mathcal{X}, P)} = \lambda_i \phi_i(y).$$

For any $S$, $\Pi(S, y) \in \operatorname{span}(f_1, \cdots, f_k)$, therefore,

$$\langle \phi_i, K(\cdot, y)\rangle_{L_2(\mathcal{X}, P)} - \lambda_i \phi_i \in \operatorname{span}(f_1, \cdots, f_k).$$

The residual kernel w.r.t. $P_N$ equals

$$K_{P_N}(x, y) = K(x, y) - \mathbb{E}_{S \sim P_N}[K(x, S)\Pi_N(S, y)] - \mathbb{E}_{S \sim P_N}[\Pi_N(x, S)K(S, y)] +$$
$$\mathbb{E}_{S, T \sim P_N}[\Pi_N(x, S)K(S, T)\Pi_N(T, y)],$$

where $\Pi_N(x, y) = \sum_{i,j=1}^k (H^{-1})_{ij} f_i(x) f_j(y)$ and $H = [H_{ij}]_{i,j=1}^k = [\langle f_i, f_j\rangle_{L_2(\mathcal{X}, P_N)}]_{i,j=1}^k$. Therefore,

$$\langle \phi_i, K_{P_N}(\cdot, y)\rangle_{L_2(\mathcal{X}, P)} = \langle \phi_i, K(\cdot, y)\rangle_{L_2(\mathcal{X}, P)} - \mathbb{E}_{S \sim P_N}[\langle \phi_i, K(\cdot, S)\rangle_{L_2(\mathcal{X}, P)}\Pi_N(S, y)].$$

Since $\langle \phi_i, K(\cdot, y)\rangle_{L_2(\mathcal{X}, P)} - \lambda_i \phi_i \in \operatorname{span}(f_1, \cdots, f_k)$, we have

$$\langle \phi_i, K(\cdot, y)\rangle_{L_2(\mathcal{X}, P)} - \mathbb{E}_{S \sim P_N}[\langle \phi_i, K(\cdot, S)\rangle_{L_2(\mathcal{X}, P)}\Pi_N(S, y)] = \lambda_i \phi_i - \mathbb{E}_{S \sim P_N}[\lambda_i \phi_i(S)\Pi_N(S, y)].$$

Thus,

$$\langle \phi_i, K_{P_N}(\cdot, \mathcal{D}_N)\rangle_{L_2(\mathcal{X}, P)} = \lambda_i \boldsymbol{\phi}_i^\top - \frac{1}{N}\sum_{j=1}^N \lambda_i \phi_i(X_j)\Pi_N(X_j, \mathcal{D}_N) = \lambda_i \boldsymbol{\phi}_i^\top (I_N - F^\top(FF^\top)^{-1}F).$$

Lemma proved. □

**Lemma D.4.** *For any $\mathcal{F}$-nondegenerate distribution $Q$, we have*

$$(I_N - F^\top(FF^\top)^{-1}F)K(\mathcal{D}_N, \mathcal{D}_N)(I_N - F^\top(FF^\top)^{-1}F) =$$
$$(I_N - F^\top(FF^\top)^{-1}F)K_Q(\mathcal{D}_N, \mathcal{D}_N)(I_N - F^\top(FF^\top)^{-1}F).$$

*Proof.* The residual kernel equals

$$K_Q(x, y) = \int (\delta(x - s) - \Pi(x, s))(\delta(y - t) - \Pi(y, t))K(s, t)dQ(s)dQ(t),$$

where $\Pi(x, y) = \sum_{i,j=1}^{k}(G^{-1})_{ij}f_i(x)f_j(y)$ and $[G_{ij}]_{i,j=1}^{k} = [\langle f_i, f_j\rangle_{L_2(\mathcal{X}, Q)}]_{i,j=1}^{k}$. So,

$$K(x, y) = K_Q(x, y) + \int \Pi(x, s)K(s, y)dQ(s) +$$

$$\int \Pi(y, t)K(x, t)dQ(t) - \int \Pi(x, s)\Pi(y, t)K(s, t)dQ(s)QP(t).$$

Since $[\Pi(x_i, s)]_{i=1}^{N} \in \text{span}(\mathbf{f}_1, \cdots, \mathbf{f}_k)$ we have $(I_N - F^\top(FF^\top)^{-1}F)[\Pi(x_i, s)]_{i=1}^{N} = 0$. Analogously, $([\Pi(t, x_i)]_{i=1}^{N})^\top(I_N - F^\top(FF^\top)^{-1}F) = 0$. Therefore,

$$(I_N - F^\top(FF^\top)^{-1}F)K(\mathcal{D}_N, \mathcal{D}_N)(I_N - F^\top(FF^\top)^{-1}F) =$$
$$(I_N - F^\top(FF^\top)^{-1}F)K_Q(\mathcal{D}_N, \mathcal{D}_N)(I_N - F^\top(FF^\top)^{-1}F).$$

Lemma proved. $\square$

**Corollary D.5.** *We have*

$$(I_N - F^\top(FF^\top)^{-1}F)K(\mathcal{D}_N, \mathcal{D}_N)(I_N - F^\top(FF^\top)^{-1}F) =$$
$$(I_N - F^\top(FF^\top)^{-1}F)\Phi^\top\Lambda\Phi(I_N - F^\top(FF^\top)^{-1}F).$$

*Proof.* After setting $Q = P$ in the previous lemma and, from $K_P(\mathcal{D}_N, \mathcal{D}_N) = \Phi^\top\Lambda\Phi$, we obtain the needed statement. $\square$

*Proof of Theorem D.1.* The vector of residuals is given by

$$r = (I_N - F^\top(FF^\top)^{-1}F)y = (I_N - F^\top(FF^\top)^{-1}F)(\Phi^\top\Lambda^{1/2}v + F^\top u + \varepsilon) =$$
$$(I_N - F^\top(FF^\top)^{-1}F)(\Phi^\top\Lambda^{1/2}v + \varepsilon).$$

From Theorem 3.2, the trained mapping $\hat{f} = \sum_{i=1}^{\infty}\hat{v}_i\sqrt{\lambda_i}\phi_i + \sum_{i=1}^{k}\hat{u}_i f_i$ can be given as

$$K_{P_N}(x, \mathcal{D}_N)(K_{P_N}(\mathcal{D}_N, \mathcal{D}_N) + \lambda N I_N)^{-1}r + \sum_{i=1}^{k}\tilde{u}_i f_i,$$

which gives

$$\sqrt{\lambda_i}\hat{v}_i = \langle \phi_i, K_{P_N}(\cdot, \mathcal{D}_N)(K_{P_N}(\mathcal{D}_N, \mathcal{D}_N) + \lambda I_N)^{-1}r\rangle_{L_2(\mathcal{X}, P)}.$$

From Lemma D.3 we conclude $\langle \phi_i, K_{P_N}(\cdot, \mathcal{D}_N)\rangle_{L_2(\mathcal{X}, P)} = \lambda_i\phi_i^\top(I_N - F^\top(FF^\top)^{-1}F)$. Therefore,

$$\sqrt{\lambda_i}\hat{v}_i = \lambda_i\phi_i^\top(I_N - F^\top(FF^\top)^{-1}F)(K_{P_N}(\mathcal{D}_N, \mathcal{D}_N) + \lambda N I_N)^{-1}r =$$
$$\lambda_i\phi_i^\top(I_N - F^\top(FF^\top)^{-1}F)(K_{P_N}(\mathcal{D}_N, \mathcal{D}_N) + \lambda N I_N)^{+}(I_N - F^\top(FF^\top)^{-1}F)(\Phi^\top\Lambda^{1/2}v + \varepsilon) =$$
$$\lambda_i\phi_i^\top((I_N - F^\top(FF^\top)^{-1}F)K_{P_N}(\mathcal{D}_N, \mathcal{D}_N)(I_N - F^\top(FF^\top)^{-1}F) + \lambda N(I_N - F^\top(FF^\top)^{-1}F))^{+}(\Phi^\top\Lambda^{1/2}v + \varepsilon).$$

Above we used the property $(AB)^{+} = B^{+}A^{+}$ for commuting symmetric matrices $A, B$ and the fact that $A^{+} = A$ for a projection operator $A$. From Lemma D.4 and Corollary D.5 we conclude

$$(I_N - F^\top(FF^\top)^{-1}F)K_{P_N}(\mathcal{D}_N, \mathcal{D}_N)(I_N - F^\top(FF^\top)^{-1}F) =$$
$$(I_N - F^\top(FF^\top)^{-1}F)\Phi^\top\Lambda\Phi(I_N - F^\top(FF^\top)^{-1}F) =$$
$$(I_N - F^\top(FF^\top)^{-1}F)\Psi^\top\Lambda\Psi(I_N - F^\top(FF^\top)^{-1}F).$$

Thus,

$$\hat{v}_i = \sqrt{\lambda_i}\phi_i^\top (I_N - F^\top(FF^\top)^{-1}F)(\Psi^\top\Lambda\Psi + \lambda N I_N)^{-1}(I_N - F^\top(FF^\top)^{-1}F)(\Phi^\top\Lambda^{1/2}v + \varepsilon),$$

and, therefore, $T_\phi = \Lambda^{1/2}\Psi(\Psi^\top\Lambda\Psi + \lambda N I_N)^{-1}\Psi^\top\Lambda^{1/2}$. Also, $T_\varepsilon = \Lambda^{1/2}\Psi(\Psi^\top\Lambda\Psi + \lambda N I_N)^{-1}(I_N - F^\top(FF^\top)^{-1}F)$.

Let us now derive the formula for $\hat{u}$ as a function of $u, v$ and $\varepsilon$. First we will assume that $f_1, \cdots, f_k$ are orthogonal unit vectors in $L_2(\mathcal{X}, P_N)$. Using $\hat{f} = \sum_{i=1}^\infty \hat{v}_i\sqrt{\lambda_i}\phi_i + \sum_{i=1}^k \hat{u}_i f_i$, we have

$$\langle f_i, \hat{f}\rangle_{L_2(\mathcal{X}, P_N)} = \frac{1}{N}\sum_{j=1}^\infty \hat{v}_j\sqrt{\lambda_j}\mathbf{f}_i^\top\phi_j + \hat{u}_i = \frac{1}{N}\mathbf{f}_i^\top\Phi^\top\Lambda^{1/2}\hat{v} + \hat{u}_i.$$

From $\hat{f} = K_{P_N}(x, \mathcal{D}_N)(K_{P_N}(\mathcal{D}_N, \mathcal{D}_N) + \lambda N I_N)^{-1}r + \sum_{i=1}^k \tilde{u}_i f_i$ we also derive $\langle \hat{f}, f_i\rangle_{L_2(\mathcal{X}, P_N)} = \tilde{u}_i$. Thus,

$$\hat{u}_i = \tilde{u}_i - \frac{1}{N}\mathbf{f}_i^\top\Phi^\top\Lambda^{1/2}\hat{v}.$$

Since $\tilde{u} = (FF^\top)^{-1}Fy = \frac{1}{N}Fy$ and $y = \Phi^\top\Lambda^{1/2}v + F^\top u + \varepsilon$ we conclude

$$\hat{u} = u + \frac{1}{N}F\Phi^\top\Lambda^{1/2}v - \frac{1}{N}F\Phi^\top\Lambda^{1/2}\hat{v} + \frac{1}{N}F\varepsilon =$$

$$u + \frac{1}{N}F\Phi^\top\Lambda^{1/2}(v - \Lambda^{1/2}\Psi(\Psi^\top\Lambda\Psi + \lambda I_N)^{-1}\Psi^\top\Lambda^{1/2}v) + \frac{1}{N}F\varepsilon,$$

where $\Psi = \Phi(I_N - F^\top(FF^\top)^{-1}F) = \Phi(I_N - \frac{1}{N}F^\top F)$. Therefore,

$$F^\top(\hat{u} - u) = \frac{1}{N}F^\top F\Phi^\top\Lambda^{1/2}(v - \Lambda^{1/2}\Psi(\Psi^\top\Lambda\Psi + \lambda I_N)^{-1}\Psi^\top\Lambda^{1/2}v) + \frac{1}{N}F^\top F\varepsilon.$$

In the latter derivation we assumed that $f_1, \cdots, f_k$ are already orthogonalized in $L_2(\mathcal{X}, P_N)$. If we do not make such an assumption, the matrix of the projection operator onto $\mathcal{F}$ in $L_2(\mathcal{X}, P_N)$ is not $\frac{1}{N}F^\top F$, but $F^\top(FF^\top)^{-1}F$, which gives us

$$F^\top(\hat{u} - u) = F^\top(FF^\top)^{-1}F\left(\Phi^\top\Lambda^{1/2}(v - \Lambda^{1/2}\Psi(\Psi^\top\Lambda\Psi + \lambda N I_N)^{-1}\Psi^\top\Lambda^{1/2}v) + \varepsilon\right).$$

Thus,

$$\hat{u} = u + (FF^\top)^{-1}F\left(\Phi^\top\Lambda^{1/2}(v - \Lambda^{1/2}\Psi(\Psi^\top\Lambda\Psi + \lambda N I_N)^{-1}\Psi^\top\Lambda^{1/2}v) + \varepsilon\right).$$

where $\Psi = \Phi(I_N - F^\top(FF^\top)^{-1}F)$. From the latter the expression for $T_f$ is straightforward. $\qquad\square$

## D.2. Distance between $\hat{f}_\perp$ and $h_\perp$ ($\hat{f}_\|$ and $f_\|$)

Recall that $h = \arg\min_{g \in \mathcal{H}_{K_P}} \frac{1}{N}\sum_{i=1}^N (g(X_i) - Y_i^\perp)^2 + \lambda\|g\|_{\mathcal{H}_{K_P}}^2$. The following theorem bounds the difference between $h$ and $\hat{f}$ (or $\hat{f}_\perp$) in $\mathcal{H}_K$. Let $\hat{v}$ be such that $h = \sum_{i=1}^\infty \hat{v}_i\sqrt{\lambda_i}\phi_i$. From Theorem D.1 we conclude, $\hat{v} = \hat{\mathbf{T}}_\phi v + \hat{\mathbf{T}}_{\phi\varepsilon}\varepsilon$ where

$$\hat{\mathbf{T}}_\phi = \Lambda^{1/2}\Phi(\Phi^\top\Lambda\Phi + \lambda N I_N)^{-1}\Phi^\top\Lambda^{1/2}$$

and

$$\hat{\mathbf{T}}_{\phi\varepsilon} = \Lambda^{1/2}\Phi(\Phi^\top\Lambda\Phi + \lambda N I_N)^{-1}.$$

Let us introduce $t : \mathcal{B}(\mathbb{R}^N, l^2(\mathbb{N})) \to \mathcal{B}(l^2(\mathbb{N}), l^2(\mathbb{N}))$ by

$$t(A) = A(A^\top A + \lambda I_N)^{-1}A^\top \in \mathcal{B}(l^2(\mathbb{N}), l^2(\mathbb{N})).$$

Then, we have $\hat{\mathbf{T}}_\phi = t(\frac{1}{\sqrt{N}}\Lambda^{1/2}\Phi)$ and $T_\phi = t(\frac{1}{\sqrt{N}}\Lambda^{1/2}\Psi)$.

**Lemma D.6.** *We have*

$$\mathbb{E}_\varepsilon[\|h - \hat{f}\|^2_{\mathcal{H}_K^\mathcal{F}}] \le \|t(\frac{1}{\sqrt{N}}\Lambda^{1/2}\Phi) - t(\frac{1}{\sqrt{N}}\Lambda^{1/2}\Psi)\|^2_{\mathcal{B}(l^2(\mathbb{N}),l^2(\mathbb{N}))}\|f\|^2_{\mathcal{H}_K^\mathcal{F}} + \sigma^2\frac{9C_{K_P}^2}{N\lambda^2}\left(\frac{C_{K_P}^2}{\lambda} + 1\right)^2.$$

*Proof.* Using Theorem 3.1, the squared semi-norm $\|h - \hat{f}\|^2_{\mathcal{H}_K}$ equals

$$\|\sum_{i=1}^\infty \hat{\hat{v}}_i\sqrt{\lambda_i}\phi_i - \sum_{i=1}^\infty \hat{v}_i\sqrt{\lambda_i}\phi_i\|^2_{\mathcal{H}_{K_P}} = \|\hat{\hat{v}} - \hat{v}\|^2_{l^2(\mathbb{N})} = \|\hat{T}_\phi v + \hat{T}_{\phi\varepsilon}\varepsilon - T_\phi v - T_{\phi\varepsilon}\varepsilon\|^2_{l^2(\mathbb{N})} =$$

$$\|(\hat{T}_\phi - T_\phi)v\|^2_{l^2(\mathbb{N})} + \|(\hat{T}_{\phi\varepsilon} - T_{\phi\varepsilon})\varepsilon\|^2_{l^2(\mathbb{N})} + 2\langle(\hat{T}_\phi - T_\phi)v, (\hat{T}_{\phi\varepsilon} - T_{\phi\varepsilon})\varepsilon\rangle_{l^2(\mathbb{N})}.$$

Taking the expectation over $\varepsilon$ gives

$$\mathbb{E}_\varepsilon[\|h - \hat{f}\|^2_{\mathcal{H}_K^\mathcal{F}}] = \|(\hat{T}_\phi - T_\phi)v\|^2_{l^2(\mathbb{N})} + \sigma^2\text{Tr}((\hat{T}_{\phi\varepsilon} - T_{\phi\varepsilon})^\top(\hat{T}_{\phi\varepsilon} - T_{\phi\varepsilon})).$$

The $v$-dependent term can be bounded by

$$\|(\hat{T}_\phi - T_\phi)v\|^2_{l^2(\mathbb{N})} \le \|\hat{T}_\phi - T_\phi\|^2_{\mathcal{B}(l^2(\mathbb{N}),l^2(\mathbb{N}))}\|v\|^2_{l^2(\mathbb{N})} =$$

$$\|t(\frac{1}{\sqrt{N}}\Lambda^{1/2}\Phi) - t(\frac{1}{\sqrt{N}}\Lambda^{1/2}\Psi)\|^2_{\mathcal{B}(l^2(\mathbb{N}),l^2(\mathbb{N}))}\|f\|^2_{\mathcal{H}_K^\mathcal{F}}.$$

Let us denote $\Pi_F = F^\top(FF^\top)^{-1}F$. The coefficient $\text{Tr}((\hat{T}_{\phi\varepsilon} - T_{\phi\varepsilon})^\top(\hat{T}_{\phi\varepsilon} - T_{\phi\varepsilon})) = \|\hat{T}_{\phi\varepsilon} - T_{\phi\varepsilon}\|^2_F$ can be bounded by

$$\|\hat{T}_{\phi\varepsilon} - T_{\phi\varepsilon}\|_F \le \|\Lambda^{1/2}\Phi(\Psi^\top\Lambda\Psi + \lambda N I_N)^{-1} - \Lambda^{1/2}\Phi(\Phi^\top\Lambda\Phi + \lambda N I_N)^{-1}\|_F +$$

$$\|\Lambda^{1/2}\Phi\Pi_F(\Psi^\top\Lambda\Psi + \lambda N I_N)^{-1}\|_F + \|\Lambda^{1/2}\Phi(\Psi^\top\Lambda\Psi + \lambda N I_N)^{-1}\Pi_F\|_F + \|\Lambda^{1/2}\Phi\Pi_F(\Psi^\top\Lambda\Psi + \lambda N I_N)^{-1}\Pi_F\|_F.$$

The resolvent identity allows to bound the first term term by

$$\|\Lambda^{1/2}\Phi(\Psi^\top\Lambda\Psi + \lambda N I_N)^{-1} - \Lambda^{1/2}\Phi(\Phi^\top\Lambda\Phi + \lambda N I_N)^{-1}\|_F \le$$

$$\frac{1}{\sqrt{N}}\|\frac{1}{\sqrt{N}}\Lambda^{1/2}\Phi\|_F \cdot \|(\frac{1}{N}\Psi^\top\Lambda\Psi + \lambda I_N)^{-1} - (\frac{1}{N}\Phi^\top\Lambda\Phi + \lambda I_N)^{-1}\|_{\mathcal{B}(\mathbb{R}^N,\mathbb{R}^N)} \le$$

$$\frac{C_{K_P}}{\sqrt{N}}\|(\frac{1}{N}\Psi^\top\Lambda\Psi + \lambda I_N)^{-1} - (\frac{1}{N}\Phi^\top\Lambda\Phi + \lambda I_N)^{-1}\|_{\mathcal{B}(\mathbb{R}^N,\mathbb{R}^N)} \le$$

$$\frac{C_{K_P}}{\sqrt{N}\lambda^2}\|\frac{1}{N}\Psi^\top\Lambda\Psi - \frac{1}{N}\Phi^\top\Lambda\Phi\|_{\mathcal{B}(\mathbb{R}^N,\mathbb{R}^N)}.$$

Since

$$\|\frac{1}{N}\Psi^\top\Lambda\Psi - \frac{1}{N}\Phi^\top\Lambda\Phi\|_{\mathcal{B}(\mathbb{R}^N,\mathbb{R}^N)} \le \|\frac{1}{N}\Pi_F^\top\Phi^\top\Lambda\Phi\|_{\mathcal{B}(\mathbb{R}^N,\mathbb{R}^N)} + \|\frac{1}{N}\Phi^\top\Lambda\Phi\Pi_F\|_{\mathcal{B}(\mathbb{R}^N,\mathbb{R}^N)} +$$

$$\|\frac{1}{N}\Pi_F^\top\Phi^\top\Lambda\Phi\Pi_F\|_{\mathcal{B}(\mathbb{R}^N,\mathbb{R}^N)} \le 3\|\frac{1}{N}\Phi^\top\Lambda\Phi\|_{\mathcal{B}(\mathbb{R}^N,\mathbb{R}^N)} \le 3C_{K_P}^2,$$

the first term is bounded by $\frac{3C_{K_P}^3}{\sqrt{N}\lambda^2}$. The 2nd, 3rd and 4th terms are bounded by

$$\|\Lambda^{1/2}\Phi\Pi_F(\Psi^\top\Lambda\Psi + \lambda N I_N)^{-1}\|_F \le \frac{\|\Lambda^{1/2}\Phi\Pi_F\|_F}{\lambda N} \le \frac{\|\Lambda^{1/2}\Phi\|_F}{\lambda N} \le \frac{C_{K_P}}{\sqrt{N}\lambda},$$

$$\|\Lambda^{1/2}\Phi(\Psi^\top\Lambda\Psi + \lambda N I_N)^{-1}\Pi_F\|_F \le \|\Lambda^{1/2}\Phi(\Psi^\top\Lambda\Psi + \lambda N I_N)^{-1}\|_F \le \frac{\|\Lambda^{1/2}\Phi\|_F}{\lambda N} \le \frac{C_{K_P}}{\sqrt{N}\lambda},$$

$$\|\Lambda^{1/2}\Phi\Pi_F(\Psi^\top\Lambda\Psi + \lambda N I_N)^{-1}\Pi_F\|_F \le \frac{\|\Lambda^{1/2}\Phi\Pi_F\|_F}{\lambda N} \le \frac{\|\Lambda^{1/2}\Phi\|_F}{\lambda N} \le \frac{C_{K_P}}{\sqrt{N}\lambda}.$$

To conclude, we have

$$\|\hat{T}_{\phi\varepsilon} - T_{\phi\varepsilon}\|_F \le \frac{3C_{K_P}^3}{\sqrt{N}\lambda^2} + \frac{3C_{K_P}}{\sqrt{N}\lambda}.$$

Lemma proved. $\qquad\square$

The following theorem bounds the difference between $f_\parallel$ and $\hat{f}_\parallel$ in $L_2(\mathcal{X}, P)$.

**Lemma D.7.** *We have*

$$\mathbb{E}_\varepsilon[\|\hat{f}_\parallel - f_\parallel\|^2_{L_2(\mathcal{X},P)}] \leq \|T_f\|^2_{\mathcal{B}(l^2(\mathbb{N}),\mathbb{R}^k)}\|f\|^2_{\mathcal{H}^{\mathcal{F}}_K} + \frac{k}{N}\|(\frac{1}{N}FF^\top)^{-1}\|_{\mathcal{B}(\mathbb{R}^k,\mathbb{R}^k)}\sigma^2.$$

*Proof.* The squared norm $\|\hat{f}_\parallel - f_\parallel\|^2_{L_2(\mathcal{X},P)}$ equals

$$\mathbb{E}_\varepsilon[\|\sum_{i=1}^k \hat{u}_i f_i - \sum_{i=1}^k u_i f_i\|^2_{L_2(\mathcal{X},P)}] = \mathbb{E}_\varepsilon[\|\hat{u} - u\|^2] = \mathbb{E}_\varepsilon[\|T_f v + T_{f\varepsilon}\varepsilon\|^2] =$$

$$\|T_f v\|^2 + \sigma^2 \mathrm{Tr}(T_{f\varepsilon}^\top T_{f\varepsilon}) \leq \|T_f\|^2_{\mathcal{B}(l^2(\mathbb{N}),\mathbb{R}^k)}\|f\|^2_{\mathcal{H}^{\mathcal{F}}_K} + \sigma^2 \mathrm{Tr}(T_{f\varepsilon}^\top T_{f\varepsilon}).$$

The second term can be bounded by

$$\mathrm{Tr}(T_{f\varepsilon}^\top T_{f\varepsilon}) = \mathrm{Tr}(T_{f\varepsilon} T_{f\varepsilon}^\top) = \mathrm{Tr}((FF^\top)^{-1}FF^\top(FF^\top)^{-1}) = \mathrm{Tr}((FF^\top)^{-1}) \leq \frac{k}{N}\|(\frac{1}{N}FF^\top)^{-1}\|_{\mathcal{B}(\mathbb{R}^k,\mathbb{R}^k)}.$$

$\square$

Thus, to control $\mathbb{E}_\varepsilon[\|\hat{f}_\perp - h\|^2_{\mathcal{H}^{\mathcal{F}}_K}]$ and $\mathbb{E}_\varepsilon[\|\hat{f}_\parallel - f_\parallel\|^2_{L_2(\mathcal{X},P)}]$ we need to bound $\|t(\frac{1}{\sqrt{N}}\Lambda^{1/2}\Phi) - t(\frac{1}{\sqrt{N}}\Lambda^{1/2}\Psi)\|^2_{\mathcal{B}(l^2(\mathbb{N}),l^2(\mathbb{N}))}$, $\|T_f\|_{\mathcal{B}(l^2(\mathbb{N}),\mathbb{R}^k)}$ and $\|(\frac{1}{N}FF^\top)^{-1}\|_{\mathcal{B}(\mathbb{R}^k,\mathbb{R}^k)}$. Required bounds for the latter expressions are obtained in the next section.

### D.3. Concentration of transfer matrices

Let us introduce the notation:
$$e(x) = [\sqrt{\lambda_1}\phi_1(x), \sqrt{\lambda_2}\phi_2(x), \cdots]^\top \in l^2(\mathbb{N}).$$

Given $x_1, \cdots, x_N \in \mathcal{X}$, let
$$e(x_1, \cdots, x_N) = [e(x_1), \cdots, e(x_N)] \in \mathcal{B}(\mathbb{R}^N, l^2(\mathbb{N})),$$

Given continuous functions $f_j : \mathcal{X} \to \mathbb{R}$, $j = 1, \cdots, k$, let us define a vector-function $\mathbf{f}(x) = [f_1(x), \cdots, f_k(x)]^\top$ and denote
$$\mathbf{f}(x_1, \cdots, x_N) = [\mathbf{f}(x_1), \cdots, \mathbf{f}(x_N)] \in \mathbb{R}^{k \times N}.$$

Using the introduced notation, the matrix $\Lambda^{1/2}\Phi$ can be rewritten as $e(X_1, \cdots, X_N)$, the matrix $F$ as $\mathbf{f}(X_1, \cdots, X_N)$, and the matrix $\Lambda^{1/2}\Psi$ in the following form:

$$\Lambda^{1/2}\Psi = e(X_1, \cdots, X_N) \times (I_N - \mathbf{f}(X_1, \cdots, X_N)^\top(\mathbf{f}(X_1, \cdots, X_N)\mathbf{f}(X_1, \cdots, X_N)^\top)^{-1}\mathbf{f}(X_1, \cdots, X_N)).$$

Our first goal is to bound $\|\Lambda^{1/2}\Psi - \Lambda^{1/2}\Phi\|_{\mathcal{B}(\mathbb{R}^N,l^2(\mathbb{N}))}$, or to bound

$$\|e(X_1, \cdots, X_N)\mathbf{f}(X_1, \cdots, X_N)^\top(\mathbf{f}(X_1, \cdots, X_N)\mathbf{f}(X_1, \cdots, X_N)^\top)^{-1}\mathbf{f}(X_1, \cdots, X_N)\|_{\mathcal{B}(\mathbb{R}^N,l^2(\mathbb{N}))}.$$

The latter expression can be bounded by a product of

$$\|\frac{1}{N}e(X_1, \cdots, X_N)\mathbf{f}(X_1, \cdots, X_N)^\top\|_{\mathcal{B}(\mathbb{R}^k,l^2(\mathbb{N}))},$$

and

$$\|(\frac{1}{N}\mathbf{f}(X_1, \cdots, X_N)\mathbf{f}(X_1, \cdots, X_N)^\top)^{-1}\|_{\mathcal{B}(\mathbb{R}^k,\mathbb{R}^k)}\|\mathbf{f}(X_1, \cdots, X_N)\|_{\mathcal{B}(\mathbb{R}^N,\mathbb{R}^k)}.$$

The following lemma is dedicated to the first factor.

**Lemma D.8.** *Let $X_1, \ldots, X_N \sim^{iid} P$. For any $t > 0$, we have*

$$\|\frac{1}{N}e(X_1, \cdots, X_N)\mathbf{f}(X_1, \cdots, X_N)^\top\|_{\mathcal{B}(\mathbb{R}^k,l^2(\mathbb{N}))} \leq \sqrt{k\left(\frac{1}{N}C^2_{K_P}\max_{j:1\leq j\leq k}\|f_j\|^2_{L_\infty(\mathcal{X})} + t\right)},$$

*with probability at least $1 - ke^{-\frac{Nt^2}{8C^4_{K_P}\max_{j:1\leq j\leq k}\|f_j\|^4_{L_\infty(\mathcal{X})}}}$.*

To prove it we need to prepare a number of lemmas.

**Lemma D.9.** *Let $f \in \mathrm{span}(f_1, \cdots, f_k)$ and $X_1, \ldots, X_N \sim^{iid} P$. We have*

$$\mathbb{E}\left[\|f'_N\|^2_{\mathcal{H}_{K_P}}\right] = \frac{1}{N} E_{X \sim P}[f(X)^2 K_P(X, X)],$$

*where $f'_N(\cdot) = \frac{1}{N} \sum_{i=1}^{N} f(X_i) K_P(X_i, \cdot)$.*

*Proof.* By the reproducing property

$$\|f'_N\|^2_{\mathcal{H}_K} = \frac{1}{N^2} \sum_{i,j=1}^{N} f(X_i)f(X_j)\langle K_P(X_i, \cdot), K_P(X_j, \cdot)\rangle_{\mathcal{H}_{K_P}} = \frac{1}{N^2} \sum_{i,j=1}^{N} f(X_i)f(X_j)K_P(X_i, X_j).$$

Hence,

$$\mathbb{E}[\|f'_N\|^2_{\mathcal{H}_{K_P}}] = \frac{1}{N}\mathbb{E}_{X \sim P}[f(X)^2 K_P(X, X)],$$

due to $\int_{\mathcal{X}} K_P(x, y)f(y)dP(y) = 0$. Lemma proved. □

**Lemma D.10.** *Under the assumptions of the previous lemma, for any $t > 0$, we have*

$$\mathbb{P}[\|f'_N\|^2_{\mathcal{H}_{K_P}} > \frac{1}{N}C^2_{K_P}\|f\|^2_{L_\infty(\mathcal{X})} + t] \le e^{-\frac{Nt^2}{8C^4_{K_P}\|f\|^4_{L_\infty(\mathcal{X})}}}.$$

*Proof.* Let us define the function $\tilde{h}$ by $\tilde{h}(X_1, \cdots, X_N) = \|f'_N\|^2_{\mathcal{H}_{K_P}} - \frac{1}{N^2}\sum_{i=1}^{N} f(X_i)^2 K_P(X_i, X_i)$. The function satisfies

$$|\tilde{h}(x_1, \cdots, x_N) - \tilde{h}(x_1, \cdots, x_{i-1}, x'_i, x_{i+1}, \cdots, x_N)| \le$$

$$\frac{2}{N^2} \sum_{\substack{j:j\neq i}}^{N} |f(x_i)f(x_j)K_P(x_i, x_j) - f(x'_i)f(x_j)K_P(x'_i, x_j)| \le \frac{4C^2_{K_P}\|f\|^2_{L_\infty(\mathcal{X})}}{N}.$$

due to $|f(x)f(y)K_P(x, y)| \le C^2_{K_P}\|f\|^2_{L_\infty(\mathcal{X})}$.

Using McDiarmid's concentration inequality, we obtain

$$\mathbb{P}[\tilde{h}(X_1, \cdots, X_N) - \mathbb{E}[\tilde{h}(X_1, \cdots, X_N)] > t] \le e^{-\frac{Nt^2}{8C^4_{K_P}\|f\|^4_{L_\infty(\mathcal{X})}}}.$$

From $\int_{\mathcal{X}} K_P(x, y)f(y)dP(y) = 0$, we obtain $\mathbb{E}[\tilde{h}(X_1, \cdots, X_N)] = 0$. Thus,

$$\mathbb{P}[\tilde{h}(X_1, \cdots, X_N) > t] \le e^{-\frac{Nt^2}{8C^4_{K_P}\|f\|^4_{L_\infty(\mathcal{X})}}},$$

and

$$\mathbb{P}[[\|f'_N\|^2_{\mathcal{H}_{K_P}} > \frac{1}{N}C^2_{K_P}\|f\|^2_{L_\infty(\mathcal{X})} + t] \le e^{-\frac{Nt^2}{8C^4_{K_P}\|f\|^4_{L_\infty(\mathcal{X})}}}.$$

□

*Proof of Lemma D.8.* Using the notations of the previous lemma, we simply need to note that

$$\|\frac{1}{N} \sum_{i=1}^{N} e(X_i)f_j(X_i)\|^2_{l^2(\mathbb{N})} = \|(f_j)'_N\|^2_{\mathcal{H}_{K_P}}.$$

For each of functions $\{f_j\}$, using the previous lemma we obtain that one of the inequalities

$$\|(f_j)'_N\|^2_{\mathcal{H}_{K_P}} > \frac{1}{N}C^2_{K_P} \max_{j:1\le j\le k}\|f_j\|^2_{L_\infty(\mathcal{X})} + t, 1 \le j \le k,$$

can be violated with probability no more than $ke^{-\frac{Nt^2}{8C_{K_P}^4 \max_j \|f_j\|_{L_\infty(\mathcal{X})}^4}}$. Thus, with probability at least $1 - ke^{-\frac{Nt^2}{8C_{K_P}^4 \max_{j:1\leq j\leq k} \|f_j\|_{L_\infty(\mathcal{X})}^4}}$ we have

$$\max_{j:1\leq j\leq k} \|\frac{1}{N}\sum_{i=1}^N e(X_i)f_j(X_i)\|_{l^2(\mathbb{N})}^2 \leq \frac{1}{N}C_{K_P}^2 \max_{j:1\leq j\leq k} \|f_j\|_{L_\infty(\mathcal{X})}^2 + t.$$

From

$$\|\frac{1}{N}e(X_1,\cdots,X_N)\mathbf{f}(X_1,\cdots,X_N)^\top\|_{\mathcal{B}(\mathbb{R}^k,l^2(\mathbb{N}))} \leq \sqrt{k} \max_{j:1\leq j\leq k} \|\frac{1}{N}\sum_{i=1}^N e(X_i)f_j(X_i)\|_{l^2(\mathbb{N})},$$

we obtain the needed statement. $\square$

Let us now deal with the second factor, i.e.

$$\|(\frac{1}{N}\mathbf{f}(x_1,\cdots,x_N)\mathbf{f}(x_1,\cdots,x_N)^\top)^{-1}\|_{\mathcal{B}(\mathbb{R}^k,\mathbb{R}^k)}\|\mathbf{f}(x_1,\cdots,x_N)\|_{\mathcal{B}(\mathbb{R}^N,\mathbb{R}^k)}.$$

The matrix $\frac{1}{N}\mathbf{f}(x_1,\cdots,x_N)\mathbf{f}(x_1,\cdots,x_N)^\top$ is simply the empirical covariance matrix for features $f_1, ..., f_k$ and it concentrates around its mean w.r.t. the operator norm due to the standard Bernstein matrix inequality argument.

**Lemma D.11.** *Let $X_1,\ldots,X_N \sim^{iid} P$. We have*

$$\|(\frac{1}{N}\mathbf{f}(X_1,\cdots,X_N)\mathbf{f}(X_1,\cdots,X_N)^\top)^{-1}\|_{\mathcal{B}(\mathbb{R}^N,\mathbb{R}^N)} \leq 2,$$

$$\|(\frac{1}{N}\mathbf{f}(X_1,\cdots,X_N)\mathbf{f}(X_1,\cdots,X_N)^\top)^{-1}\|_{\mathcal{B}(\mathbb{R}^k,\mathbb{R}^k)}\|\mathbf{f}(X_1,\cdots,X_N)\|_{\mathcal{B}(\mathbb{R}^N,\mathbb{R}^k)} \leq \sqrt{6N},$$

*with probability at least $1 - 2k\exp\left(-\frac{N}{\frac{28}{3}k\max_{j:1\leq j\leq k}\|f_j\|_{L_\infty(\mathcal{X})}^2+\frac{4}{3}}\right)$.*

*Proof.* We have

$$\frac{1}{N}\mathbf{f}(X_1,\cdots,X_N)\mathbf{f}(X_1,\cdots,X_N)^\top - I_k = \frac{1}{N}\sum_{i=1}^N(\mathbf{f}(X_i)\mathbf{f}(X_i)^\top - I_k).$$

Matrices $\mathbf{f}(X_i)\mathbf{f}(X_i)^\top - I_k \in \mathbb{R}^{k\times k}$ are independent and

$$\|\mathbf{f}(X_i)\mathbf{f}(X_i)^\top - I_k\|_{\mathcal{B}(\mathbb{R}^k,\mathbb{R}^k)} \leq \|\mathbf{f}(X_i)\mathbf{f}(X_i)^\top\|_{\mathcal{B}(\mathbb{R}^k,\mathbb{R}^k)} + 1 \leq k\max_{j:1\leq j\leq k}\|f_j\|_{L_\infty(\mathcal{X})}^2 + 1.$$

Note that $\mathbb{E}[\mathbf{f}(X_i)\mathbf{f}(X_i)^\top - I_k] = 0$. For the second moment we have

$$\mathbb{E}[(\mathbf{f}(X_i)\mathbf{f}(X_i)^\top - I_k)^2] = \mathbb{E}[\|\mathbf{f}(X_i)\|^2\mathbf{f}(X_i)\mathbf{f}(X_i)^\top] - I_k \preceq$$
$$(k\max_{j:1\leq j\leq k}\|f_j\|_{L_\infty(\mathcal{X})}^2 - 1)I_k \preceq k\max_{j:1\leq j\leq k}\|f_j\|_{L_\infty(\mathcal{X})}^2 I_k.$$

After summing we obtain

$$\left\|\sum_{i=1}^N \mathbb{E}[(\mathbf{f}(X_i)\mathbf{f}(X_i)^\top - I_k)^2]\right\|_{\mathcal{B}(\mathbb{R}^k,\mathbb{R}^k)} \leq Nk\max_{j:1\leq j\leq k}\|f_j\|_{L_\infty(\mathcal{X})}^2.$$

Matrix Bernstein inequality (see Theorem 6.1.1 in (Tropp, 2015)) gives us

$$\mathbb{P}[\|\frac{1}{N}\sum_{i=1}^N(\mathbf{f}(X_i)\mathbf{f}(X_i)^\top - I_k)\|_{\mathcal{B}(\mathbb{R}^k,\mathbb{R}^k)} \geq t] \leq$$

$$2k\exp\left(-\frac{Nt^2/2}{k\max_{j:1\leq j\leq k}\|f_j\|_{L_\infty(\mathcal{X})}^2 + (k\max_{j:1\leq j\leq k}\|f_j\|_{L_\infty(\mathcal{X})}^2 + 1)t/3}\right).$$

Let us choose $t = \frac{1}{2}$ in the previous inequality. Then, we have

$$\|\frac{1}{N}\mathbf{f}(X_1, \cdots, X_N)\mathbf{f}(X_1, \cdots, X_N)^\top - I_k\|_{\mathcal{B}(\mathbb{R}^k, \mathbb{R}^k)} < \frac{1}{2}$$

with probability at least $1 - 2k \exp\left(-\dfrac{N/8}{k \max\limits_{j:1 \leq j \leq k} \|f_j\|_{L_\infty(\mathcal{X})}^2 + (k \max\limits_{j:1 \leq j \leq k} \|f_j\|_{L_\infty(\mathcal{X})}^2 + 1)/6}\right)$. In that case we have

$$\|(\frac{1}{N}\mathbf{f}(X_1, \cdots, X_N)\mathbf{f}(X_1, \cdots, X_N)^\top)^{-1}\|_{\mathcal{B}(\mathbb{R}^N, \mathbb{R}^N)} < \frac{1}{1 - 0.5} = 2,$$

and

$$\|\mathbf{f}(X_1, \cdots, X_N)\|_{\mathcal{B}(\mathbb{R}^N, \mathbb{R}^k)} = \sqrt{N}\sqrt{\|\frac{1}{N}\mathbf{f}(X_1, \cdots, X_N)\mathbf{f}(X_1, \cdots, X_N)^\top\|_{\mathcal{B}(\mathbb{R}^k, \mathbb{R}^k)}} \leq \sqrt{\frac{3N}{2}}.$$

Thus, we have

$$\|(\frac{1}{N}\mathbf{f}(X_1, \cdots, X_N)\mathbf{f}(X_1, \cdots, X_N)^\top)^{-1}\|_{\mathcal{B}(\mathbb{R}^N, \mathbb{R}^N)}\|\mathbf{f}(X_1, \cdots, X_N)\|_{\mathcal{B}(\mathbb{R}^N, \mathbb{R}^k)} \leq \sqrt{6N}.$$

Lemma proved. □

A combination of Lemma D.8 and Lemma D.11 gives us that for any $t > 0$,

$$\|\frac{1}{\sqrt{N}}\Lambda^{1/2}\Psi - \frac{1}{\sqrt{N}}\Lambda^{1/2}\Phi\|_{\mathcal{B}(\mathbb{R}^N, l^2(\mathbb{N}))} \leq \sqrt{6k\left(\frac{1}{N}C_{K_P}^2 \max\limits_{j:1 \leq j \leq k} \|f_j\|_{L_\infty(\mathcal{X})}^2 + t\right)}$$

with probability at least

$$q(t) = 1 - k \exp\left(-\frac{Nt^2}{8C_{K_P}^4 \max\limits_{j:1 \leq j \leq k} \|f_j\|_{L_\infty(\mathcal{X})}^4}\right) - 2k \exp\left(-\frac{N}{\frac{28}{3}k \max\limits_{j:1 \leq j \leq k} \|f_j\|_{L_\infty(\mathcal{X})}^2 + \frac{4}{3}}\right)$$

over randomness in inputs $X_1, \cdots, X_N$.

In the next step, we need to bound $\|t(\frac{1}{\sqrt{N}}\Lambda^{1/2}\Psi) - t(\frac{1}{\sqrt{N}}\Lambda^{1/2}\Phi)\|_{\mathcal{B}(l^2(\mathbb{N}), l^2(\mathbb{N}))}$. Note that

$$\|\frac{1}{\sqrt{N}}\Lambda^{1/2}\Phi\|_{\mathcal{B}(\mathbb{R}^N, l^2(\mathbb{N}))} \leq C_{K_P},$$

and therefore,

$$\|\frac{1}{\sqrt{N}}\Lambda^{1/2}\Psi\|_{\mathcal{B}(\mathbb{R}^N, l^2(\mathbb{N}))} \leq C_{K_P} + \sqrt{6k\left(\frac{1}{N}C_{K_P}^2 \max\limits_{j:1 \leq j \leq k} \|f_j\|_{L_\infty(\mathcal{X})}^2 + t\right)},$$

with probability at least $q(t)$. We now need a lemma that bounds $\|t(\frac{1}{\sqrt{N}}\Lambda^{1/2}\Psi) - t(\frac{1}{\sqrt{N}}\Lambda^{1/2}\Phi)\|_{\mathcal{B}(l^2(\mathbb{N}), l^2(\mathbb{N}))}$ in terms of $\|\frac{1}{\sqrt{N}}\Lambda^{1/2}\Psi\|_{\mathcal{B}(\mathbb{R}^N, l^2(\mathbb{N}))}, \|\frac{1}{\sqrt{N}}\Lambda^{1/2}\Phi\|_{\mathcal{B}(\mathbb{R}^N, l^2(\mathbb{N}))}$ and $\|\frac{1}{\sqrt{N}}\Lambda^{1/2}\Psi - \frac{1}{\sqrt{N}}\Lambda^{1/2}\Phi\|_{\mathcal{B}(\mathbb{R}^N, l^2(\mathbb{N}))}$.

**Lemma D.12.** *Let* $A, B \in \mathcal{B}(\mathbb{R}^N, l^2(\mathbb{N}))$ *be such that* $\|A\|_{\mathcal{B}(\mathbb{R}^N, l^2(\mathbb{N}))}, \|B\|_{\mathcal{B}(\mathbb{R}^N, l^2(\mathbb{N}))} \leq \alpha$. *Then,*

$$\|t(A) - t(B)\|_{\mathcal{B}(l^2(\mathbb{N}), l^2(\mathbb{N}))} \leq \left(\frac{2\alpha}{\lambda} + \frac{2\alpha^3}{\lambda^2}\right)\|A - B\|_{\mathcal{B}(\mathbb{R}^N, l^2(\mathbb{N}))}.$$

*Proof.* Let us denote $M_A = A^\top A + \lambda I_N$ and $M_B = B^\top B + \lambda I_N$. We have

$$t(A) - t(B) = AM_A^{-1}A^\top - BM_B^{-1}B^\top.$$

By adding and subtracting $AM_A^{-1}B^\top$ and $BM_A^{-1}B^\top$ we split

$$t(A) - t(B) =$$
$$(AM_A^{-1}A^\top - AM_A^{-1}B^\top) + (AM_A^{-1}B^\top - BM_A^{-1}B^\top) + (BM_A^{-1}B^\top - BM_B^{-1}B^\top) =$$
$$AM_A^{-1}(A-B)^\top + (A-B)M_A^{-1}B^\top + B(M_A^{-1} - M_B^{-1})B^\top.$$

The first term can be bounded by

$$\|AM_A^{-1}(A-B)^\top\|_{\mathcal{B}(l^2(\mathbb{N}),l^2(\mathbb{N}))} \leq$$
$$\|A\|_{\mathcal{B}(\mathbb{R}^N,l^2(\mathbb{N}))} \cdot \|M_A^{-1}\|_{\mathcal{B}(\mathbb{R}^N,\mathbb{R}^N)} \cdot \|A-B\|_{\mathcal{B}(\mathbb{R}^N,l^2(\mathbb{N}))} \leq \frac{\alpha}{\lambda}\|A-B\|_{\mathcal{B}(\mathbb{R}^N,l^2(\mathbb{N}))}.$$

The second term by

$$\|(A-B)M_A^{-1}B^\top\|_{\mathcal{B}(l^2(\mathbb{N}),l^2(\mathbb{N}))} \leq$$
$$\|A-B\|_{\mathcal{B}(\mathbb{R}^N,l^2(\mathbb{N}))} \cdot \|M_A^{-1}\|_{\mathcal{B}(\mathbb{R}^N,\mathbb{R}^N)} \cdot \|B\|_{\mathcal{B}(\mathbb{R}^N,l^2(\mathbb{N}))} \leq \frac{\alpha}{\lambda}\|A-B\|_{\mathcal{B}(\mathbb{R}^N,l^2(\mathbb{N}))}.$$

From the resolvent identity $M_A^{-1} - M_B^{-1} = M_A^{-1}(M_B - M_A)M_B^{-1}$, we obtain

$$\|M_A^{-1} - M_B^{-1}\|_{\mathcal{B}(\mathbb{R}^N,\mathbb{R}^N)} \leq \frac{1}{\lambda^2}\|M_A - M_B\|_{\mathcal{B}(\mathbb{R}^N,\mathbb{R}^N)} \leq$$
$$\frac{1}{\lambda^2}(\|A\|_{\mathcal{B}(\mathbb{R}^N,l^2(\mathbb{N}))} + \|B\|_{\mathcal{B}(\mathbb{R}^N,l^2(\mathbb{N}))})\|A-B\|_{\mathcal{B}(\mathbb{R}^N,l^2(\mathbb{N}))} = \frac{2\alpha}{\lambda^2}\|A-B\|_{\mathcal{B}(\mathbb{R}^N,l^2(\mathbb{N}))}$$

Using this, we bound the third term by

$$\|B(M_A^{-1} - M_B^{-1})B^\top\|_{\mathcal{B}(l^2(\mathbb{N}),l^2(\mathbb{N}))} \leq \|B\|^2_{\mathcal{B}(\mathbb{R}^N,l^2(\mathbb{N}))} \cdot \|M_A^{-1} - M_B^{-1}\|_{\mathcal{B}(\mathbb{R}^N,\mathbb{R}^N)} \leq$$
$$\alpha^2 \cdot \frac{2\alpha}{\lambda^2}\|A-B\|_{\mathcal{B}(\mathbb{R}^N,l^2(\mathbb{N}))} = \frac{2\alpha^3}{\lambda^2}\|A-B\|_{\mathcal{B}(\mathbb{R}^N,l^2(\mathbb{N}))}.$$

After collecting the bounds, we obtain

$$\|t(A) - t(B)\|_{\mathcal{B}(l^2(\mathbb{N}),l^2(\mathbb{N}))} \leq \frac{2\alpha}{\lambda}\|A-B\|_{\mathcal{B}(\mathbb{R}^N,l^2(\mathbb{N}))} + \frac{2\alpha^3}{\lambda^2}\|A-B\|_{\mathcal{B}(\mathbb{R}^N,l^2(\mathbb{N}))} =$$
$$\left(\frac{2\alpha}{\lambda} + \frac{2\alpha^3}{\lambda^2}\right)\|A-B\|_{\mathcal{B}(\mathbb{R}^N,l^2(\mathbb{N}))}.$$

Lemma proved. $\qquad\square$

**Lemma D.13.** *Let $t > 0$ and $\alpha = C_{K_P}\left(1 + \max\limits_{j:1\leq j\leq k}\|f_j\|_{L_\infty(\mathcal{X})}\sqrt{6k\left(\frac{1}{N} + t\right)}\right)$. Then,*

$$\|t(\frac{1}{\sqrt{N}}\Lambda^{1/2}\Psi) - t(\frac{1}{\sqrt{N}}\Lambda^{1/2}\Phi)\|_{\mathcal{B}(l^2(\mathbb{N}),l^2(\mathbb{N}))} \leq$$
$$\left(\frac{2\alpha}{\lambda} + \frac{2\alpha^3}{\lambda^2}\right)C_{K_P}\max\limits_{j:1\leq j\leq k}\|f_j\|_{L_\infty(\mathcal{X})}\sqrt{6k\left(\frac{1}{N} + t\right)},$$

*with probability at least $1 - k\exp\left(-\frac{Nt^2}{8}\right) - 2k\exp\left(-\frac{N}{\frac{28}{3}k\max\limits_{j:1\leq j\leq k}\|f_j\|^2_{L_\infty(\mathcal{X})}+\frac{4}{3}}\right)$.*

*Proof.* After we apply Lemma D.12 to $A = \frac{1}{\sqrt{N}}\Lambda^{1/2}\Psi$, $B = \frac{1}{\sqrt{N}}\Lambda^{1/2}\Phi$ we obtain the following statement: let $t > 0$ and $\alpha = C_{K_P} + \sqrt{6k\left(\frac{1}{N}C^2_{K_P}\max\limits_{j:1\leq j\leq k}\|f_j\|^2_{L_\infty(\mathcal{X})} + t\right)}$. Then,

$$\|t(\frac{1}{\sqrt{N}}\Lambda^{1/2}\Psi) - t(\frac{1}{\sqrt{N}}\Lambda^{1/2}\Phi)\|_{\mathcal{B}(l^2(\mathbb{N}),l^2(\mathbb{N}))} \leq$$
$$\left(\frac{2\alpha}{\lambda} + \frac{2\alpha^3}{\lambda^2}\right)\sqrt{6k\left(\frac{1}{N}C^2_{K_P}\max\limits_{j:1\leq j\leq k}\|f_j\|^2_{L_\infty(\mathcal{X})} + t\right)},$$

with probability at least $1 - k\exp\left(-\frac{Nt^2}{8C_{K_P}^4 \max\limits_{j:1\leq j\leq k}\|f_j\|_{L_\infty(\mathcal{X})}^4}\right) - 2k\exp\left(-\frac{N}{\frac{28}{3}k \max\limits_{j:1\leq j\leq k}\|f_j\|_{L_\infty(\mathcal{X})}^2 + \frac{4}{3}}\right)$. Rescaling $t = t'C_{K_P}^2 \max\limits_{j:1\leq j\leq k}\|f_j\|_{L_\infty(\mathcal{X})}^2$ gives the desired inequality. $\qquad\square$

Recall that $T_f = (FF^\top)^{-1}F\Phi^\top\Lambda^{1/2}(I - \Lambda^{1/2}\Psi(\Psi^\top\Lambda\Psi + \lambda N I_N)^{-1}\Psi^\top\Lambda^{1/2})$. We finally need to bound $\|T_f\|_{\mathcal{B}(l^2(\mathbb{N}),\mathbb{R}^k)}$ which is done in the next lemma.

**Lemma D.14.** *For any $t > 0$, we have*

$$\|T_f\|_{\mathcal{B}(l^2(\mathbb{N}),\mathbb{R}^k)} \leq 4C_{K_P} \max_{j:1\leq j\leq k}\|f_j\|_{L_\infty(\mathcal{X})}\sqrt{k\left(\frac{1}{N} + t\right)},$$

*with probability at least $1 - ke^{-\frac{Nt^2}{8}} - 2k\exp\left(-\frac{N}{\frac{28}{3}k \max\limits_{j:1\leq j\leq k}\|f_j\|_{L_\infty(\mathcal{X})}^2 + \frac{4}{3}}\right)$.*

*Proof.* From Lemma D.8 we obtain

$$\|\frac{1}{N}F\Phi^\top\Lambda^{1/2}\|_{\mathcal{B}(l^2(\mathbb{N}),\mathbb{R}^k)} = \|\frac{1}{N}\Lambda^{1/2}\Phi F^\top\|_{\mathcal{B}(\mathbb{R}^k,l^2(\mathbb{N}))} \leq \sqrt{k\left(\frac{1}{N}C_{K_P}^2 \max_{j:1\leq j\leq k}\|f_j\|_{L_\infty(\mathcal{X})}^2 + t\right)},$$

with probability at least $1 - ke^{-\frac{Nt^2}{8C_{K_P}^4 \max\limits_{j:1\leq j\leq k}\|f_j\|_{L_\infty(\mathcal{X})}^4}}$. Therefore, using Lemma D.11, we have

$$\|T_f\|_{\mathcal{B}(l^2(\mathbb{N}),\mathbb{R}^k)} \leq$$

$$\|(\frac{1}{N}FF^\top)^{-1}\|_{\mathcal{B}(\mathbb{R}^k,\mathbb{R}^k)}\|\frac{1}{N}F\Phi^\top\Lambda^{1/2}\|_{\mathcal{B}(l^2(\mathbb{N}),\mathbb{R}^k)} \cdot \|I - \Lambda^{1/2}\Psi(\Psi^\top\Lambda\Psi + \lambda I_N)^{-1}\Psi^\top\Lambda^{1/2}\|_{\mathcal{B}(l^2(\mathbb{N}),l^2(\mathbb{N}))} \leq$$

$$2\sqrt{k\left(\frac{1}{N}C_{K_P}^2 \max_{j:1\leq j\leq k}\|f_j\|_{L_\infty(\mathcal{X})}^2 + t\right)} \cdot \left(1 + \|\Lambda^{1/2}\Psi(\Psi^\top\Lambda\Psi + \lambda I_N)^{-1}\Psi^\top\Lambda^{1/2}\|_{\mathcal{B}(l^2(\mathbb{N}),l^2(\mathbb{N}))}\right) \leq$$

$$4\sqrt{k\left(\frac{1}{N}C_{K_P}^2 \max_{j:1\leq j\leq k}\|f_j\|_{L_\infty(\mathcal{X})}^2 + t\right)},$$

with probability at least $1 - ke^{-\frac{Nt^2}{8C_{K_P}^4 \max\limits_{j:1\leq j\leq k}\|f_j\|_{L_\infty(\mathcal{X})}^4}} - 2k\exp\left(-\frac{N}{\frac{28}{3}k \max\limits_{j:1\leq j\leq k}\|f_j\|_{L_\infty(\mathcal{X})}^2 + \frac{4}{3}}\right)$. By rescaling $t = t'C_{K_P}^2 \max\limits_{j:1\leq j\leq k}\|f_j\|_{L_\infty(\mathcal{X})}^2$ we obtain the needed inequality. $\qquad\square$

### D.4. Final steps of the proof

**Lemma D.15.** *Suppose that $f_1,\cdots,f_k$ are orthogonal functions of unit norm in $L_2(\mathcal{X},P)$. Let $t > 0$ and $\alpha = C_{K_P}\left(1 + \max\limits_{j:1\leq j\leq k}\|f_j\|_{L_\infty(\mathcal{X})}\sqrt{6k\left(\frac{1}{N} + t\right)}\right)$. With probability at least $1 - k\exp\left(-\frac{Nt^2}{8}\right) - 2k\exp\left(-\frac{N}{\frac{28}{3}k \max\limits_{j:1\leq j\leq k}\|f_j\|_{L_\infty(\mathcal{X})}^2 + \frac{4}{3}}\right)$ over randomness in $X_1,\cdots,X_N$, we have*

$$\mathbb{E}_\varepsilon[c_{\mathrm{con}}] \leq \|f\|_{\mathcal{H}_K}^2 C_{K_P}^2 \max_{j:1\leq j\leq k}\|f_j\|_{L_\infty(\mathcal{X})}^2 k\left(\frac{1}{N} + t\right)\left(16 + 6\lambda_1\left(\frac{2\alpha}{\lambda} + \frac{2\alpha^3}{\lambda^2}\right)^2\right) +$$

$$\frac{\sigma^2}{N}\left(\frac{9\lambda_1 C_{K_P}^2}{\lambda^2}\left(\frac{C_{K_P}^2}{\lambda} + 1\right)^2 + 2k\right).$$

*Proof of Lemma D.15.* From Theorem 3.1 we conclude that

$$\|\hat{f}_\perp - h\|_{L_2(\mathcal{X},P)} \leq \sqrt{\lambda_1}\|\hat{f}_\perp - h\|_{\mathcal{H}_{K_P}} = \sqrt{\lambda_1}\|\hat{f}_\perp - h\|_{\mathcal{H}_K^\mathcal{F}}.$$

Therefore,

$$c_{\mathrm{con}} \le \lambda_1 \|\hat{f}_\perp - h\|_{\mathcal{H}_K^{\mathcal{F}}}^2 + \|\hat{f}_\| - f_\|\|_{L_2(\mathcal{X},P)}^2.$$

From Lemmas D.6 and D.7 we obtain

$$\mathbb{E}_\varepsilon[c_{\mathrm{con}}] \le \lambda_1 \|t(\frac{1}{\sqrt{N}}\Lambda^{1/2}\Phi) - t(\frac{1}{\sqrt{N}}\Lambda^{1/2}\Psi)\|_{\mathcal{B}(l^2(\mathbb{N}),l^2(\mathbb{N}))}^2 \|f\|_{\mathcal{H}_K^{\mathcal{F}}}^2 +$$

$$\lambda_1\sigma^2 \frac{9C_{K_P}^2}{N\lambda^2}\left(\frac{C_{K_P}^2}{\lambda}+1\right)^2 + \|T_f\|_{\mathcal{B}(l^2(\mathbb{N}),\mathbb{R}^k)}^2 \|f\|_{\mathcal{H}_K^{\mathcal{F}}}^2 + \frac{k}{N}\|(\frac{1}{N}FF^\top)^{-1}\|_{\mathcal{B}(\mathbb{R}^k,\mathbb{R}^k)}\sigma^2.$$

Using Lemma D.13, the first term is bounded by

$$\lambda_1\|f\|_{\mathcal{H}_K^{\mathcal{F}}}^2 \left(\frac{2\alpha}{\lambda}+\frac{2\alpha^3}{\lambda^2}\right)^2 C_{K_P}^2 \max_{j:1\le j\le k}\|f_j\|_{L_\infty(\mathcal{X})}^2 6k\left(\frac{1}{N}+t\right),$$

with probability at least $1 - k\exp\left(-\frac{Nt^2}{8}\right) - 2k\exp\left(-\frac{N}{\frac{28}{3}k\max_{j:1\le j\le k}\|f_j\|_{L_\infty(\mathcal{X})}^2+\frac{4}{3}}\right)$, where $t > 0$ and $\alpha = C_{K_P}\left(1 + \max_{j:1\le j\le k}\|f_j\|_{L_\infty(\mathcal{X})}\sqrt{6k\left(\frac{1}{N}+t\right)}\right)$.

Using Lemma D.14, the third term is bounded by

$$16\|f\|_{\mathcal{H}_K^{\mathcal{F}}}^2 C_{K_P}^2 \max_{j:1\le j\le k}\|f_j\|_{L_\infty(\mathcal{X})}^2 k\left(\frac{1}{N}+t\right),$$

with probability at least $1 - k\exp\left(-\frac{Nt^2}{8}\right) - 2k\exp\left(-\frac{N}{\frac{28}{3}k\max_{j:1\le j\le k}\|f_j\|_{L_\infty(\mathcal{X})}^2+\frac{4}{3}}\right)$.

Using Lemma D.11, the last term is bounded by

$$\frac{2k\sigma^2}{N},$$

with probability at least $1 - 2k\exp\left(-\frac{N}{\frac{28}{3}k\max_{j:1\le j\le k}\|f_j\|_{L_\infty(\mathcal{X})}^2+\frac{4}{3}}\right)$.

Thus, with probability at least $1 - 2k\exp\left(-\frac{Nt^2}{8}\right) - 6k\exp\left(-\frac{N}{\frac{28}{3}k\max_{j:1\le j\le k}\|f_j\|_{L_\infty(\mathcal{X})}^2+\frac{4}{3}}\right)$, we have

$$\mathbb{E}_\varepsilon[c_{\mathrm{con}}] \le \|f\|_{\mathcal{H}_K^{\mathcal{F}}}^2 C_{K_P}^2 \max_{j:1\le j\le k}\|f_j\|_{L_\infty(\mathcal{X})}^2 k\left(\frac{1}{N}+t\right)\left(16+6\left(\frac{2\alpha}{\lambda}+\frac{2\alpha^3}{\lambda^2}\right)^2\lambda_1\right)+$$

$$\lambda_1\sigma^2 \frac{9C_{K_P}^2}{N\lambda^2}\left(\frac{C_{K_P}^2}{\lambda}+1\right)^2 + \frac{2k\sigma^2}{N}.$$

The latter almost coincides with the statement of lemma, though constants in front of $k$ in the expression for the probability are different. Note that $k\exp\left(-\frac{Nt^2}{8}\right)$ is the probability of the violation of the inequality of Lemma D.8 and $2k\exp\left(-\frac{N}{\frac{28}{3}k\max_{j:1\le j\le k}\|f_j\|_{L_\infty(\mathcal{X})}^2+\frac{4}{3}}\right)$ is the probability of the violation of the inequality of Lemma D.11. In the latter sequence of arguments we counted the first probabiliity twice and the second one three times. More accurate reasoning gives us that the last inequality is true with probability at least $1 - k\exp\left(-\frac{Nt^2}{8}\right) - 2k\exp\left(-\frac{N}{\frac{28}{3}k\max_{j:1\le j\le k}\|f_j\|_{L_\infty(\mathcal{X})}^2+\frac{4}{3}}\right)$. □

*Proof of Theorem 4.2.* Let $t > 0$ be such that $k \exp\left(-\frac{Nt^2}{8}\right) = \frac{\delta}{2}$, i.e. $t = \sqrt{\frac{8\log(\frac{2k}{\delta})}{N}}$. Our assumption that $N$ satisfies

$N \geq \left(\frac{28}{3}k \max_{j:1\leq j\leq k} \|f_j\|_{L_\infty(\mathcal{X})}^2 + \frac{4}{3}\right)\log(\frac{4k}{\delta})$ is equivalent to $2k \exp\left(-\frac{N}{\frac{28}{3}k \max_{j:1\leq j\leq k}\|f_j\|_{L_\infty(\mathcal{X})}^2 + \frac{4}{3}}\right) \leq \frac{\delta}{2}$. In Lemma D.15

we have $\alpha = C_{K_P}\left(1 + \max_{j:1\leq j\leq k}\|f_j\|_{L_\infty(\mathcal{X})}\sqrt{6k\left(\frac{1}{N}+t\right)}\right)$. So, using $\sqrt{\frac{8\log(\frac{2k}{\delta})}{N}} \geq \frac{1}{N}$, we have

$$\alpha \leq C_{K_P}\left(1 + \max_{j:1\leq j\leq k}\|f_j\|_{L_\infty(\mathcal{X})}\sqrt{12k\sqrt{\frac{8\log(\frac{2k}{\delta})}{N}}}\right) \leq$$

$$C_{K_P}\left(1 + \max_{j:1\leq j\leq k}\|f_j\|_{L_\infty(\mathcal{X})}\frac{6k^{1/2}\log^{1/4}(\frac{2k}{\delta})}{N^{1/4}}\right) \leq 7C_{K_P}.$$

provided that $N \geq k^2\log(\frac{2k}{\delta})\max_{j:1\leq j\leq k}\|f_j\|_{L_\infty(\mathcal{X})}^4$.

From Lemma D.15 we obtain that with probability at least $1 - \delta$ we have

$$\mathbb{E}_\varepsilon[c_{\mathrm{con}}] \leq \|f\|_{\mathcal{H}_K^{\mathcal{F}}}^2 C_{K_P}^2 \max_{j:1\leq j\leq k}\|f_j\|_{L_\infty(\mathcal{X})}^2 2k\sqrt{\frac{8\log(\frac{2k}{\delta})}{N}}\left(16 + 6\lambda_1\left(\frac{14C_{K_P}}{\lambda} + \frac{686C_{K_P}^3}{\lambda^2}\right)^2\right) +$$

$$\frac{\sigma^2}{N}\left(\frac{9\lambda_1 C_{K_P}^2}{\lambda^2}\left(\frac{C_{K_P}^2}{\lambda}+1\right)^2 + 2k\right).$$

Theorem proved. □

## E. Proof of Theorem 5.2

Since

$$\mathbb{E}[\Pi_k(x,y)] = \sum_{\ell,m=1}^\infty \sqrt{\lambda_\ell\lambda_m} \cdot \mathbb{E}[M_{\ell,m}] \cdot \phi_\ell(x)\phi_m(y),$$

our task reduces to computing

$$\mathbb{E}[M_{\ell,m}] = \mathbb{E}\left[\sum_{i,j=1}^k \xi_{i\ell}(G^{-1})_{ij}\xi_{jm}\right].$$

**Lemma E.1.** *The off-diagonal elements of the projection coefficient matrix $[M_{\ell,m}]_{\ell,m=1}^\infty$ satisfy*

$$\mathbb{E}[M_{\ell,m}] = 0 \quad \text{for } \ell \neq m.$$

*Proof.* Define $v_\ell = (\xi_{1\ell}, \ldots, \xi_{k\ell})^\top \in \mathbb{R}^k$. Note that $G = \sum_{j=1}^\infty \lambda_j v_j v_j^\top$ and $M_{\ell,m} = v_\ell^\top G^{-1} v_m$. Consider flipping the sign of all components of $v_\ell$, i.e., define

$$\tilde{v}_j = \begin{cases} -v_\ell & \text{if } j = \ell, \\ v_j & \text{if } j \neq \ell. \end{cases}$$

For $\tilde{G} = \sum_{j=1}^\infty \lambda_j \tilde{v}_j \tilde{v}_j^\top$, and $\tilde{M}_{\ell,m} = \tilde{v}_\ell^\top \tilde{G}^{-1}\tilde{v}_m$, observe $\tilde{G} = G$ and

$$\tilde{M}_{\ell,m} = (-v_\ell)^\top G^{-1} v_m = -v_\ell^\top G^{-1} v_m = -M_{\ell,m}.$$

Since Gaussians are symmetric, the joint distribution of all $\xi_{ij} \sim \mathcal{N}(0,1)$ is invariant under sign flips of any single coordinate vector $v_\ell$. Therefore, $\mathbb{E}[M_{\ell,m}] = 0$. □

Recall that $\Pi_P$ denotes the projection operator onto the $\mathrm{span}(g_1, \cdots, g_k)$. From Lemma E.1 we conclude

$$\mathbb{E}[\Pi_k(x, y)] = \sum_{\ell=1}^{\infty} \lambda_\ell \cdot \mathbb{E}[M_{\ell,\ell}] \cdot \phi_\ell(x)\phi_\ell(y).$$

and, therefore,

$$\mathbb{E}[\Pi_P[K(\cdot, z_2)](x, y)] = \langle \mathbb{E}[\Pi_n(x, \cdot)], K(\cdot, y) \rangle_{L_2(\mathcal{X}, P)} = \sum_{\ell=1}^{\infty} \lambda_\ell^2 \cdot \mathbb{E}[M_{\ell,\ell}] \cdot \phi_\ell(x)\phi_\ell(y).$$

Let us now compute

$$\mathbb{E}[\Pi_P[\Pi_P[K(\cdot, z_2)](z_1, \cdot)](x, y)] = \mathbb{E}\left[ \iint \Pi_k(x, u) K(u, v) \Pi_k(v, y) \, dP(u) dP(v) \right].$$

Recall that $K(u, v) = \sum_{\ell=1}^{\infty} \lambda_\ell \phi_\ell(u) \phi_\ell(v)$. Now plug into the expression for $\Pi_k(x, y)$ and obtain that the latter equals

$$\iint (\sum_{i,j} \sqrt{\lambda_i \lambda_j} M_{i,j} \phi_i(x) \phi_j(u))(\sum_{\ell} \lambda_\ell \phi_\ell(u) \phi_\ell(v))(\sum_{m,n} \sqrt{\lambda_m \lambda_n} M_{m,n} \phi_m(v) \phi_n(y)) dP(u) dP(v)$$

$$= \sum_{i,j,n} \lambda_j \sqrt{\lambda_i \lambda_j \lambda_j \lambda_n} \cdot M_{i,j} M_{j,n} \cdot \phi_i(x) \phi_n(y) = \sum_{i,j,n} \lambda_j^2 \sqrt{\lambda_i \lambda_n} \cdot M_{i,j} M_{j,n} \cdot \phi_i(x) \phi_n(y).$$

Thus,

$$\mathbb{E}[\Pi_P[\Pi_P[K(\cdot, z_2)](z_1, \cdot)](x, y)] = \sum_{i,n} \phi_i(x) \phi_n(y) \cdot \sqrt{\lambda_i \lambda_n} \cdot \left( \sum_j \lambda_j^2 \cdot \mathbb{E}[M_{i,j} M_{j,n}] \right)$$

Let us define

$$C_{i,n} = \sum_{j=1}^{\infty} \lambda_j^2 \cdot M_{i,j} M_{j,n},$$

so that

$$\mathbb{E}[\Pi_P[\Pi_P[K(\cdot, z_2)](z_1, \cdot)](x, y)] = \sum_{i,n} \sqrt{\lambda_i \lambda_n} \cdot \mathbb{E}[C_{i,n}] \cdot \phi_i(x) \phi_n(y).$$

**Lemma E.2.** *The off-diagonal elements $C_{i,n}$ satisfy*

$$\mathbb{E}[C_{i,n}] = 0 \quad \text{for } i \neq n.$$

*Proof.* Fix $i \neq n$. Using the notations of the previous lemma we have

$$C_{i,n} = \sum_j \lambda_j^2 \cdot v_i^\top G^{-1} v_j \cdot v_j^\top G^{-1} v_n.$$

Flipping the sign of $v_i \to -v_i$, and leaving all other $v_j$ unchanged, leads to the change in the sign of $C_{i,n}$. Therefore, $\mathbb{E}[C_{i,n}] = 0$. $\qquad\square$

From Lemma E.2 we obtain

$$\mathbb{E}[\Pi_P[\Pi_P[K(\cdot, z_2)](z_1, \cdot)](x, y)] = \sum_i \lambda_i \cdot \mathbb{E}[C_{i,i}] \cdot \phi_i(x) \phi_i(y),$$

with

$$\mathbb{E}[C_{i,i}] = \sum_j \lambda_j^2 \cdot \mathbb{E}[M_{ij}^2].$$

So, we proved

$$\mathbb{E}_{\boldsymbol{\omega}}[K_P^{\boldsymbol{\omega}}(x, y)] = \sum_{i=1}^{\infty} \lambda_i (1 - 2\lambda_i \cdot \mathbb{E}[M_{i,i}] + \sum_j \lambda_j^2 \cdot \mathbb{E}[M_{i,j}^2]) \cdot \phi_i(x) \phi_i(y).$$

# F. The behaviour of $\mu_i/\lambda_i$ in the thermodynamic limit

Let us qualitively analyze how $\{\mu_i\}$ are related to $\{\lambda_i\}$. This type of non-rigorous analysis has been applied to a similar expression in (Simon et al., 2023), though it should be considered as a way to derive the formula (4), rather than a mathematically precise statement. So, as pointed out in Remark 5.1, we may substitute $(\xi_i^\top G_{-i}^{-1}\xi_i)^{-1}$ with a constant $\varkappa$ around which this expression concentrates as $k \to +\infty$. That is, $\mathbb{E}[M_{i,i}] \approx M_{i,i} \approx \frac{1/\varkappa}{1+\lambda_i/\varkappa} = \frac{1}{\lambda_i+\varkappa}$. Since $\sum_{i=1}^{\infty} \lambda_i \xi_i^\top G^{-1}\xi_i = \mathrm{Tr}(G^{-1}G) = k$, the constant $\varkappa > 0$ can be calculated from the condition $\sum_{i=1}^{\infty} \frac{\lambda_i}{\lambda_i+\varkappa} = k$.

The expression $\sum_j \lambda_j^2 \mathbb{E}[M_{i,j}^2]$ in Theorem 5.2 decomposes to $\lambda_i^2 \mathbb{E}[M_{i,i}^2] + \sum_{j\neq i} \lambda_j^2 \mathbb{E}[M_{i,j}^2]$, where $\lambda_i^2 \mathbb{E}[M_{i,i}^2] \approx \frac{\lambda_i^2}{(\lambda_i+\varkappa)^2}$. The remaining part without the expectation equals

$$\sum_{j:j\neq i} \lambda_j^2 M_{i,j}^2 = \frac{\xi_i^\top G_{-i}^{-1} \sum_{j:j\neq i} \lambda_j^2 \xi_j \xi_j^\top G_{-i}^{-1} \xi_i}{\left(1 + \lambda_i \xi_i^\top G_{-i}^{-1} \xi_i\right)^2}.$$

If we neglect the remaining term, using Theorem 5.2, we would obtain $\frac{\mu_i}{\lambda_i} \approx 1 - \frac{2\lambda_i}{\lambda_i+\varkappa} + \frac{\lambda_i^2}{(\lambda_i+\varkappa)^2} = \frac{\varkappa^2}{(\lambda_i+\varkappa)^2}$. As our experiments show (see Figures 6 and 7), this term cannot be neglected although it contributes proportionally to $\frac{\varkappa^2}{(\lambda_i+\varkappa)^2}$. So, we conjecture

$$\frac{\mu_i}{\lambda_i} \approx \frac{c\varkappa^2}{(\lambda_i + \varkappa)^2}.$$

For $\lambda_i = \frac{1}{i^a}$ we observe $c \approx a$.

# G. Random Feature approximation to the conditional KRR

The goal of this section is to establish existence of the random feature approximation of conditional KRR, similar to the approximation of standard KRR (Rahimi & Recht, 2007) (see also (Takhanov, 2024)). Recall that $K(x,y) = \mathbb{E}_{\omega\sim\mathcal{P}}[f(\omega,x)f(\omega,y)]$ where $(\Omega, \Sigma, \mathcal{P})$ is a probabilistic space. Let us now introduce new features $f_i'(x) = \frac{1}{\sqrt{m}} f(\omega_i', x)$ for i.i.d. samples $\omega_1', \ldots, \omega_m' \sim \mathcal{P}$ and define feature vectors as $\phi(x) = [f_1(x), \ldots, f_k(x)]^\top$ and $\psi(x) = [f_1'(x), \ldots, f_m'(x)]^\top$. Consider the following random feature ridge regression (RFRR) problem

$$\min_{u\in\mathbb{R}^k, w\in\mathbb{R}^m} \frac{1}{N} \sum_{i=1}^{N} (u^\top\phi(x_i) + w^\top\psi(x_i) - y_i)^2 + \lambda\|w\|_2^2. \tag{5}$$

The meaning of this task is to give a budget on weights of features $f_i'$ while having a complete freedom in selection of weights for the features $f_i, i = 1, \cdots, k$.

**Theorem G.1.** *Let* $\mathrm{rank}([f_i(x_j)]_{i=1}^{k}{}_{j=1}^{N}) = k$ *and* $u \in \mathbb{R}^k, w \in \mathbb{R}^m$ *be the solution of the task* (5). *Then, as* $m \to +\infty$,

$$u^\top\phi(x) + w^\top\psi(x) \to f(x) \text{ with probability 1,}$$

*where $f$ is the solution of the task* (1).

*Proof.* The goal is to minimize
$$\|A^\top u + B^\top w - y\|^2 + \lambda N w^\top w,$$
where $y = (y_1, \ldots, y_N)^\top \in \mathbb{R}^N$ and $A = [f_i(x_j)]_{i=1}^{k}{}_{j=1}^{N} \in \mathbb{R}^{k\times N}$ and $B = [f_i'(x_j)]_{i=1}^{m}{}_{j=1}^{N} \in \mathbb{R}^{m\times N}$. Gradients w.r.t. $u$ and $w$ are equal to $0$ if an only if
$$AA^\top u = -A(B^\top w - y),$$
$$(BB^\top + \lambda N I_m)w = -B(A^\top u - y).$$
So, the trained function satisfies
$$u^\top\phi(x) + w^\top\psi(x) = -(w^\top B - y^\top)A^\top(AA^\top)^{-1}\phi(x) + w^\top\psi(x) =$$
$$y^\top A^\top(AA^\top)^{-1}\phi(x) + w^\top(\psi(x) - BA^\top(AA^\top)^{-1}\phi(x)).$$

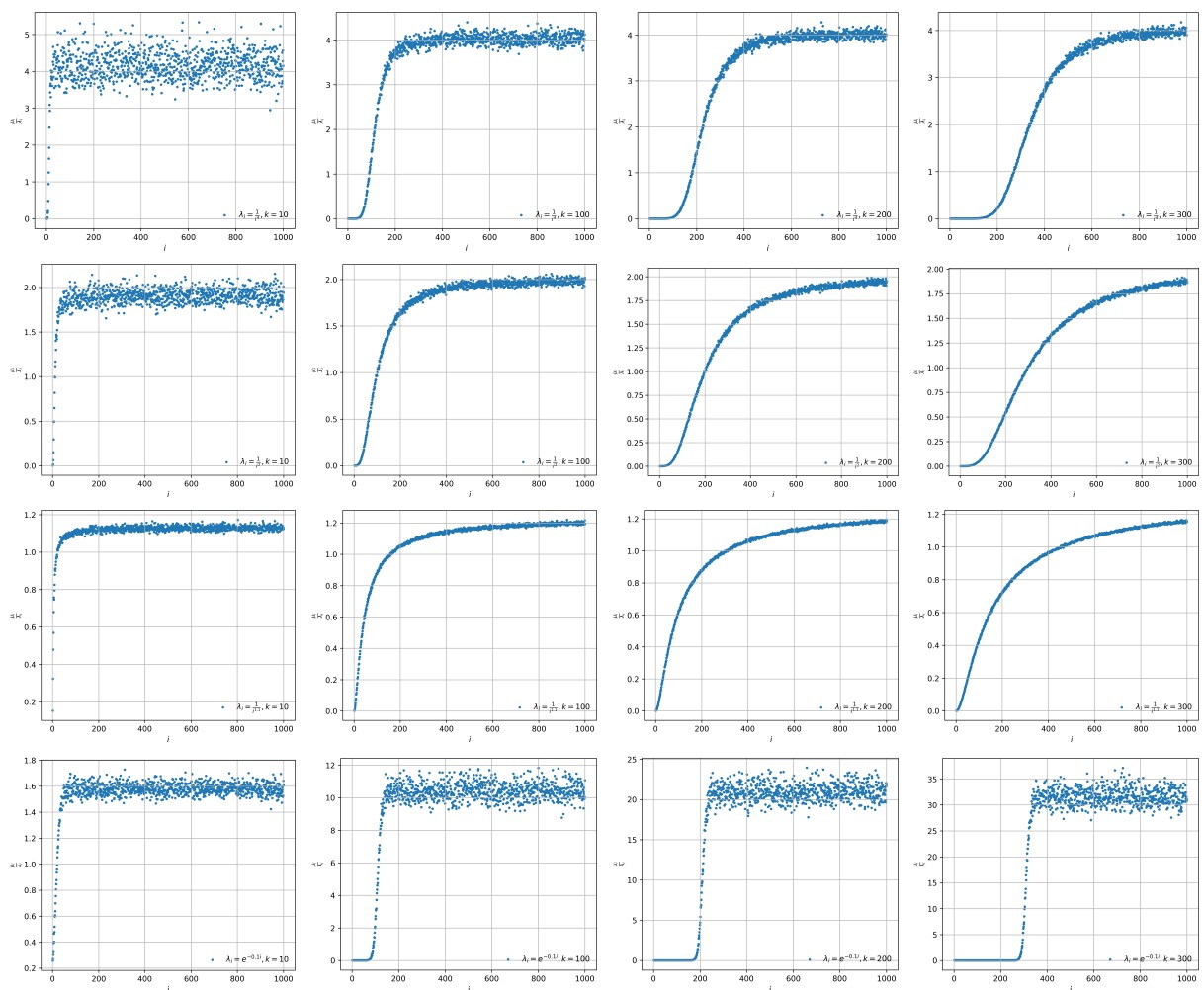

*Figure 6.* The behaviour of $1 - 2\lambda_i \cdot \mathbb{E}[M_{i,i}] + \sum_{j=1}^{\infty} \lambda_j^2 \cdot \mathbb{E}[M_{i,j}^2]$ computed by 50 times Monte-Carlo sampling for $k = 10, 100, 200, 300$ (columns) and eigenvalues (a) $\lambda_i = \frac{1}{i^4}$, (b) $\lambda_i = \frac{1}{i^2}$, (c) $\lambda_i = \frac{1}{i^{1.1}}$, (d) $\lambda_i = e^{-0.1i}$ (rows).

Note that $y^\top A^\top (AA^\top)^{-1}\phi(x)$ is the output mapping of the linear regression with the feature vector $\phi(x)$ applied to the training data.

Recall that $\Pi_{P_N}$ denotes the projection operator onto $\mathrm{span}(f_1, \cdots, f_k)$ in $L_2(\mathcal{X}, P_N)$. Let $\tilde{f}_i = (I - \Pi_{P_N})[f'_i]$,

$$\tilde{\psi}(x) = [\tilde{f}_1(x), \cdots, \tilde{f}_m(x)]^\top = (I - \Pi_{P_N})[\psi]$$

and $\tilde{B} = [\tilde{f}_i(x_j)]_{i=1}^m {}_{j=1}^N \in \mathbb{R}^{m \times N}$. By construction, $BA^\top (AA^\top)^{-1}\phi(x) = \Pi_{P_N}[\psi](x) = \psi(x) - \tilde{\psi}(x)$. Thus, we have

$$u^\top \phi(x) + w^\top \psi(x) = w^\top \tilde{\psi}(x) + y^\top A^\top (AA^\top)^{-1}\phi(x).$$

The matrix $\Pi = A^\top (AA^\top)^{-1} A$ is the projection matrix onto the row space of $A$. So, we have

$$(BB^\top + \lambda N I_m)w = -B(-\Pi B^\top w + \Pi y - y) \Rightarrow$$
$$w = (B(I - \Pi)B^\top + \lambda N I_m)^{-1} B(I - \Pi)y.$$

The vector $r = (I - \Pi)y$ is exactly the vector of residuals. By construction, $\tilde{B} = B(I - \Pi)$. Then, $\tilde{B}\tilde{B}^\top = B(I - \Pi)B^\top$ and $Br = \tilde{B}r$. Therefore,

$$w = (\tilde{B}\tilde{B}^\top + \lambda N I_m)^{-1} \tilde{B}r.$$

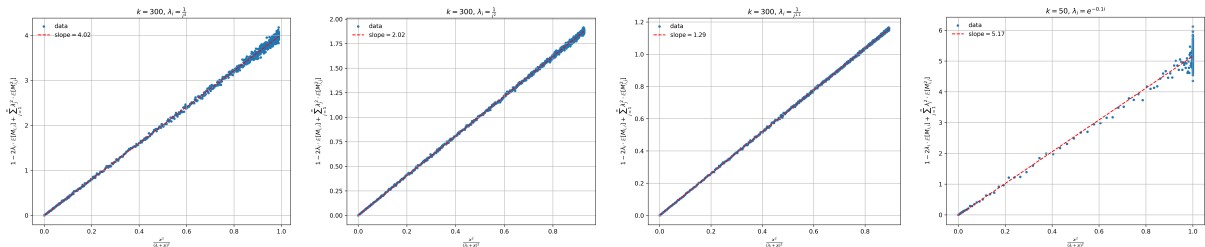

*Figure 7.* Scatter plot for $\frac{\varkappa^2}{(\lambda_i+\varkappa)^2}$ vs. $1 - 2\lambda_i \cdot \mathbb{E}[M_{i,i}] + \sum_{j=1}^{\infty} \lambda_j^2 \cdot \mathbb{E}[M_{i,j}^2]$ for (a) $\lambda_i = \frac{1}{i^4}$, (b) $\lambda_i = \frac{1}{i^2}$, (c) $\lambda_i = \frac{1}{i^{1.1}}$, (d) $\lambda_i = e^{-0.1i}$.

The Woodbury matrix identity gives $(\tilde{B}\tilde{B}^\top + \lambda N I_m)^{-1} = \frac{1}{\lambda N} I_m - \frac{1}{\lambda N} \tilde{B}(\lambda N I_N + \tilde{B}^\top \tilde{B})^{-1} \tilde{B}^\top$, and we obtain the standard kernel trick identity

$$w = \tilde{B}(\lambda N I_N + \tilde{B}^\top \tilde{B})^{-1} r.$$

Let us denote $\tilde{K} = [\tilde{\psi}(x_i)^\top \tilde{\psi}(x_j)]_{i,j=1}^N = \tilde{B}^\top \tilde{B}$. So, $w = \sum_{j=1}^N a_i \tilde{\psi}(x_i)$ where $a = [a_i]_{i=1}^N = (\tilde{K} + \lambda N I_N)^{-1} r$. Thus, the first term of the trained function $\tilde{f}(x) = w^\top \tilde{\psi}(x)$ is

$$\tilde{f}(x) = \sum_{i=1}^N a_i \tilde{\psi}(x_i)^\top \tilde{\psi}(x) = r^\top (\tilde{K} + \lambda N I_N)^{-1} [\tilde{\psi}(x_1)^\top \tilde{\psi}(x), \cdots, \tilde{\psi}(x_N)^\top \tilde{\psi}(x)]^\top.$$

Let us analyze the behaviour of that function under $m \to +\infty$. By the law of large numbers

$$\tilde{K} \to [K_{P_N}(x_i, x_j)]_{i,j=1}^N,$$

and

$$\tilde{\psi}(x_i)^\top \tilde{\psi}(x) \to K_{P_N}(x_i, x),$$

as $m \to +\infty$. That is

$$\tilde{f}(x) \to r^\top ([K_{P_N}(x_i, x_j)] + \lambda N I_N)^{-1} [K_{P_N}(x_1, x), \cdots, K_{P_N}(x_N, x)]^\top.$$

The latter is exactly the solution of

$$\min_{g \in \mathcal{H}_{K_{P_N}}} \frac{1}{N} \sum_{i=1}^N (g(x_i) - \tilde{y}_i)^2 + \lambda \|g\|_{\mathcal{H}_{K_{P_N}}}^2,$$

Using Theorem 3.2 we conclude that

$$u^\top \phi(x) + w^\top \psi(x) \to f(x) \text{ with probability } 1.$$

$\square$

Thus, RFRR method can be considered as an approximation of the conditional KRR.

## H. Details of experiments and additional experiments

### H.1. Details of experiments on hard thresholding with synthetic data: dependence of the cost of conditioning on $N, k, \sigma^2$

To verify the decay rates of the conditioning cost $c_{\text{con}}$ predicted by Theorem 4.2, we used the experimental setup described in Section 6 (the hard thresholding with synthetic data case). For fixed parameters $N$ (sample size), $k$ (chosen such that $\dim(\mathcal{F}) = 2k + 1$), $\sigma^2$ (noise variance), and a selected regression function, we repeated the following procedure 20 or 50 times: (a) sample training and test sets; (b) train both the conditional KRR and the $\mathcal{F}$-conditional learner; (c) estimate $c_{\text{con}}$ as the mean squared distance between the two resulting estimators on the test set. Finally, we averaged $c_{\text{con}}$ across repetitions and denote this empirical estimate by $\hat{c}_{\text{con}}$.

## H.2. Details of experiments on the hard thresholding with real world data

For real (non-synthetic) data, the eigenfunctions $\phi_i$ of the integral operator $\phi \mapsto \int_{\mathcal{X}} K(\cdot, x)\phi(x)dP(x)$ are not available in closed form and must be estimated from samples. Given a Mercer kernel $K$ and training points $X_1, \ldots, X_N$, let $G = [K(X_i, X_j)] \in \mathbb{R}^{N \times N}$ be the Gram matrix, and let $(\hat{\lambda}_i, \alpha_i)$ denote the eigenpairs of the Hermitian matrix $\frac{1}{N}G$, ordered so that $\hat{\lambda}_1 \geq \hat{\lambda}_2 \geq \cdots$ and normalized by $\|\alpha_i\|_2 = 1$. We write $P_N = \frac{1}{N}\sum_{j=1}^{N}\delta_{X_j}$ for the empirical distribution.

We estimate the eigenfunction $\phi_i$ by its empirical extension

$$\widehat{\phi}_i(x) = \frac{1}{\sqrt{N}\hat{\lambda}_i}\sum_{j=1}^{N}\alpha_i(j)K(x, X_j), \qquad i = 1, \ldots, \operatorname{rank}(G).$$

This normalization is chosen so that $\widehat{\phi}_i(X_\ell) = \sqrt{N}\alpha_i(\ell)$, making the family $\widehat{\phi}_i$ orthonormal in $L_2(\mathcal{X}, P_N)$. Moreover, each $\widehat{\phi}_i$ is an eigenfunction of the empirical integral operator

$$O_N f(x) = \frac{1}{N}\sum_{j=1}^{N}K(x, X_j)f(X_j),$$

satisfying $O_N\widehat{\phi}_i = \hat{\lambda}_i\widehat{\phi}_i$.

In all our experiments, we therefore substitute $\widehat{\phi}_i$ for the true eigenfunctions $\phi_i$ and define $\mathcal{F}_k = \{\widehat{\phi}_1, \ldots, \widehat{\phi}_k\}$. Conditional KRR with respect to this $\mathcal{F}_k$ can be seen as a practical approximation of the hard-thresholding setting described in Subsection 5.1.

| Kernel | KRR | C-KRR |
|---|---|---|
| Gaussian (RBF) | $0.0671 \pm 0.0015$ | $0.0732 \pm 0.0023$ |
| Laplace | $0.0750 \pm 0.0029$ | $0.0744 \pm 0.0023$ |
| Matern ($\nu = 1.5$) | $0.1102 \pm 0.0038$ | $0.1087 \pm 0.0029$ |
| NNGP (erf) | $0.0574 \pm 0.0020$ | $0.0572 \pm 0.0022$ |

*Table 1.* Test MSE comparison of KRR ($k = 0$) with $\lambda$ optimized on the validation set and the hard thresholding setup ($\lambda = 0.01$) with $k$ optimized on the validation set for different kernels (on the 7-vs-9 MNIST).

In Section 6 we report results of experiments with the 7-vs-9 MNIST dataset. We also compared the test MSE of Conditional KRR—using a fixed regularization parameter $\lambda$ and selecting $k$ via validation—with the test MSE of standard KRR equipped with an optimally tuned $\lambda$. As shown in Table 1, the resulting test errors are extremely close, to the point where their difference is statistically insignificant (at least for the kernels considered and for the MNIST dataset).

We also investigated the effect of varying $k$ while keeping $\lambda$ fixed at the value optimally tuned for standard KRR. In this setting, the test MSE for small $k$ is nearly identical to its value at $k = 0$, and it increases only for sufficiently large $k$. These empirical findings suggest that when $\lambda$ is already optimized for KRR, adjusting $k$ provides essentially no additional benefit. Naturally, this conclusion applies only to the MNIST dataset and the set of kernels examined here.

## H.3. Soft thresholding with random features: additional experiments

According to Theorem G.1, the larger $m$ (the number of penalized random features), the closer RFRR approximates conditional KRR. The plots reported in Section 6 were obtained with $m = 2000$. We also tested the method on several non-synthetic datasets and consistently observed the same U-shaped behavior. Figure 8 shows the RFRR train/test MSE as a function of $k$ (the number of unpenalized random features) for $m = 10000$ and 7-vs-9 MNIST dataset (rescaled so that pixel intensities fall within $[0, 1]$, and mean-centered). For the cosine activation, adding unpenalized features consistently worsens the test MSE due to catastrophic overfitting, a well-known issue in ridgeless Gaussian KRR (equivalent to using unpenalized random features with cosine activation).

*Remark* H.1 (Beyond hard and soft thresholding setups). Suppose that $\lambda$ is optimally tuned for standard KRR. Although the results above suggest that using the eigenfunctions of the operator $\phi \longmapsto \int_{\mathcal{X}} K(\cdot, x)\phi(x)dP(x)$ as unpenalized features does not improve the test MSE, this does not imply that Conditional KRR cannot outperform standard KRR when supplied with a different choice of unpenalized features.

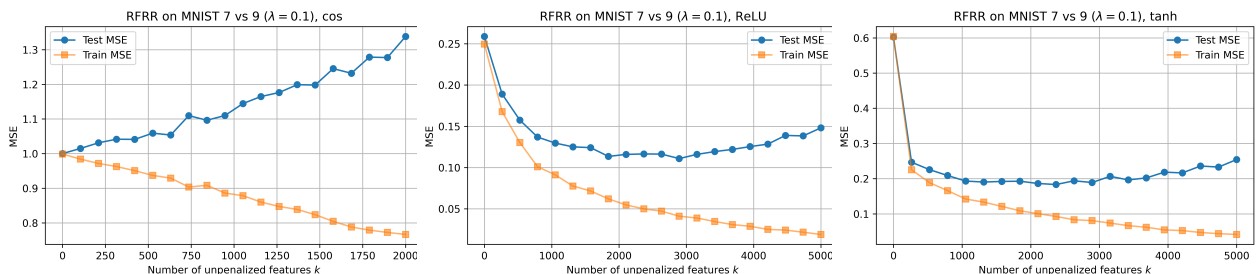

*Figure 8.* The effect of the soft thresholding for the cosine, ReLU and tanh activation functions on the 7-vs-9 MNIST dataset.

To demonstrate that Conditional KRR can exhibit a clear U-shaped dependence of $\mathrm{MSE}(k)$ on $k$ for the 7-vs-9 MNIST dataset, we conducted the following experiment. We first trained a two-layer neural network with ReLU activation and 20 hidden units, i.e., the model $\mathrm{NN}_\theta(x) = \sum_{i=1}^{20} a_i \mathrm{ReLU}(w_i^\top x + b_i) + c$, using $L_2$-regularization and the 7-vs-9 MNIST training set. Next, we trained a random-feature approximation of Conditional KRR based on the corresponding ReLU kernel

$$K(x,y) = \mathbb{E}_{w \sim \mathcal{N}(0, I_d/d), b \sim U[-1,1]} \left[ \mathrm{ReLU}(w^\top x + b) \mathrm{ReLU}(w^\top y + b) \right],$$

using $m = 10{,}000$ random features. We set $k = 20$ and defined the unpenalized subspace $\mathcal{F}_k = \{\mathrm{ReLU}(w_i^\top x + b_i) \mid 1 \le i \le k\}$, i.e. the features extracted from the trained neural network. The resulting test MSE curves as functions of $\lambda$ are shown in Figure 9.

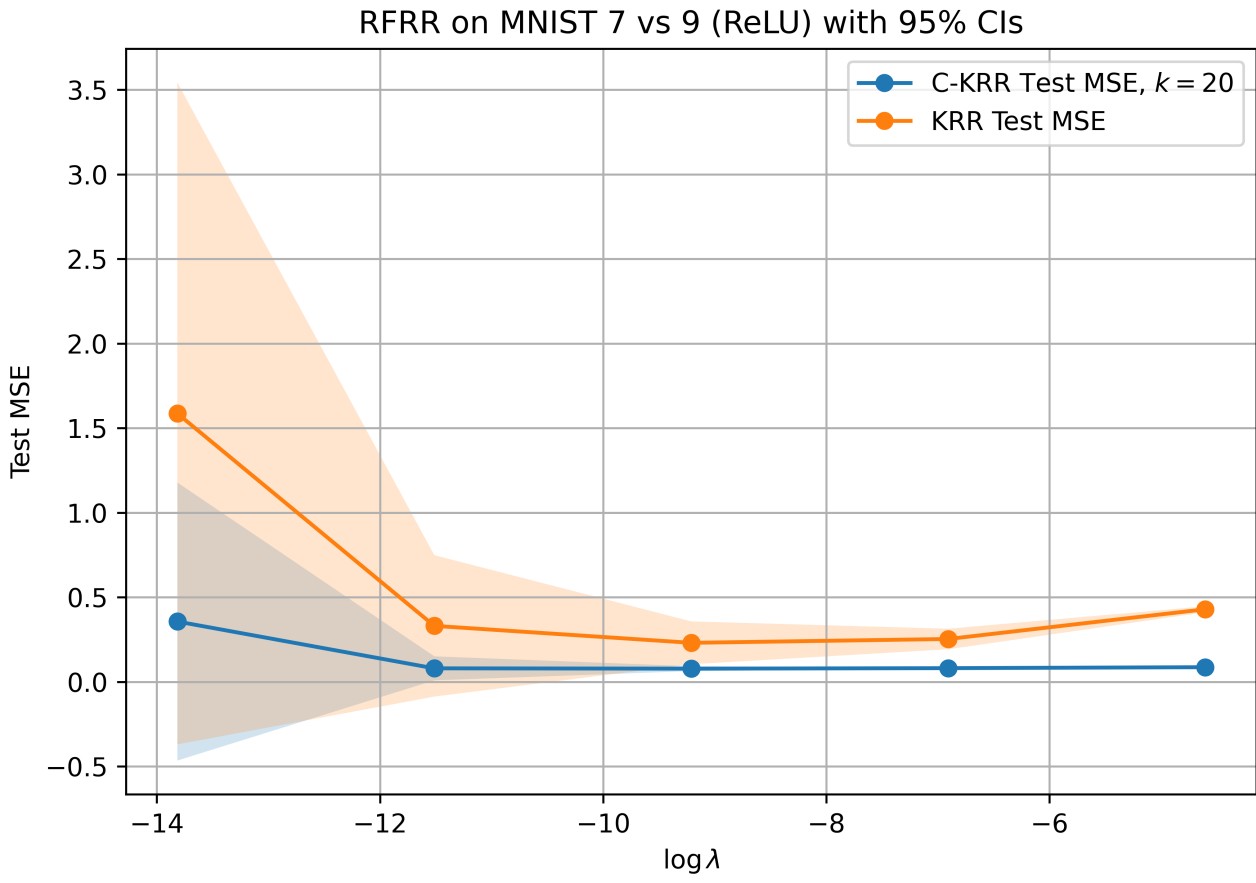

*Figure 9.* Conditional KRR with trained unpenalized features vs standard KRR.

The code can be accessed on GitHub, allowing for easy reproduction of our results.

