# OpenReview forum: "Conditional KRR: Injecting Unpenalized Features into Kernel Methods with Applications to Kernel Thresholding"
_ICML.cc/2026/Conference — ICML 2026 regular_

### Official Review · Reviewer_F9qm · 2026-03-02

**Soundness:** 3
**Presentation:** 2
**Significance:** 3
**Originality:** 3
**Overall Recommendation:** 4
**Confidence:** 3

**Summary:**

The paper studies conditional kernel ridge regression (conditional KRR) for kernels that are conditionally positive definite with respect to a function class $F$. The key observation is that conditional KRR can be equivalently viewed as ordinary KRR with a data-dependent "residual kernel" plus an additive error term interpreted as a "cost of conditioning". The authors give a bound for this cost, analyze several choices of unpenalized features (top eigenfunctions, random Gaussian features, random features), and provide experimental evidence suggesting a U-shaped test-error curve as a function of the number of unpenalized features.

**Compliance With Llm Reviewing Policy:**

Affirmed.

**Key Questions For Authors:**

First, the paper should give a more interpretable characterization of when conditional KRR yields lower expected risk than standard KRR. Inequality (3) is too abstract, and no concrete regimes or examples are provided.

Second, it remains unclear why the theoretically derived rate is only $O( k / \sqrt{N} )$ when numerically the decay appears closer to $1/N$. The authors should explain whether this gap reflects limitations of the proof technique or whether they believe a sharper bound is fundamentally impossible.

Third, it is important to understand how sensitive the data-dependent residual kernel is to sampling fluctuations, since this object is key to the method’s behavior.

Fourth, the empirical section lacks comparisons beyond standard KRR; methods such as truncated eigen-KRR, additive models with unpenalized components, or random-feature ridge with low-rank augmentation could highlight whether conditional KRR brings any distinctive advantage.

**Limitations:**

yes

**Strengths And Weaknesses:**

Overall, this study addresses a fundamental issue in understanding how unpenalized feature injection interacts with kernel ridge regression. The authors seek to present an important concept, reducing conditional KRR to ordinary KRR via a residual kernel, and provide a mathematically structured argument.

Soundness: The mathematical formulation is broadly correct and the reduction to a residual kernel is valid. However, the main statistical guarantee is not very sharp. Although the bound on the conditioning cost is technically involved, its final rate is quite coarse and behaves essentially as $k/\sqrt{N}$, which is worse than what is achievable in modern kernel regression analyses that exploit eigenvalue decay more effectively. Many of the assumptions, including orthonormality of the $F$-features, nondegeneracy of the distribution, and certain Mercer-type expansions, are strong and do not receive comparison with established results. The theoretical development thus feels more structural than statistical, because the paper does not clarify under which realistic conditions conditional KRR provides provable gains over standard KRR beyond an abstract inequality.

The presentation is mathematically dense and the flow of ideas is difficult to follow. The main conceptual message, namely, that conditional KRR reduces to standard KRR with a data-dependent kernel, is somewhat buried beneath heavy notation. Several classical ingredients from spline-based CPD kernels and native-space constructions are reintroduced with extensive formality, but the paper provides limited intuition as to why these elements lead to improved generalization or when the conditioning mechanism becomes beneficial in practice. A more direct comparison with known methods such as spectral truncation, additive models with unpenalized components, or spline null-space formulations would significantly clarify the contribution.

Significance: The conceptual connection to KRR and its variants is appealing, but it is unclear how impactful the results are for the broader machine-learning community. The statistical rate is relatively weak, the assumptions may not hold in most applied scenarios, and the practical value relies on having prior knowledge of the correct unpenalized component in the regression function. Moreover, although the paper parallels recent developments on eigen-learning, benign overfitting, and random-feature models, it does not produce new theoretical insight on these topics. Consequently, the contribution feels incremental and mainly interpretive rather than advancing the state of the art.

The originality of the work is moderate. While the specific formulation of the "cost of conditioning" seems new, much of the theoretical machinery is classical and well established in the literature on CPD kernels, spline theory, and the theory of native spaces. The links drawn between hard thresholding, soft thresholding, and random-feature models resemble existing arguments in kernel eigen-analysis and random matrix theory. Hence the work does not substantially extend these lines of research.

---

> ### Author Rebuttal · Authors · 2026-03-31
>
> We are grateful for the reviewer’s constructive questions.
>
> F9qm: "... why the theoretically derived rate is only $O( k / \sqrt{N} )$ when numerically the decay appears closer to $1/N$..."
>
> The observed rate $1/N$ was due to some regularity of the kernel we used in experiments. We did not have a goal to demonstrate the worst-case scenario of $1 / \sqrt{N}$, but now we have found the case where square root is observed in practice. Let $\mathcal{X}=[0,2\pi]$ with the uniform distribution on it, the kernel be $K(x,y)=\sum_{i=0}^{300}\cos(i(x-y))$, $\mathcal{F}=\\{\cos(ix),\sin(ix)\mid 0\leq i\leq 5\\}$, and the target is $f(x)=\cos(6x)+\sin(6x)$ given with noise of unit variance. In that case, experiment shows $1 / \sqrt{N}$ decay rate of the cost of conditioning. The code and the plot can be found in the github repository of the paper.
>
> So, to summarize, the proof technique is adequate, though with additional restrictions on the kernel (e.g. on the eigenvalues decay rate or smoothness) it is very likely that a better bound can be proved.
>
> F9qm: ".. how sensitive the data-dependent residual kernel is to sampling fluctuations..."
>
> We did not address that question specifically in the paper, but in fact, some estimations on that can be inferred from the proof of Theorem 4.2. There are two types of residual kernels that we deal with in paper: a residual kernel induced by the fixed (usually unknown) data distribution $P$, $K_P$ (this is the fixed kernel we deal with in our bound on the cost of conditioning in Section 4). Also, a residual kernel can be defined by the empirical distribution $P_N = \frac{1}{N}\sum_{i}\delta_{X_i}$ (this one we deal with in Theorem 3.2 and in an implementation of C-KRR). To distinguish these two notions, let us denote the second kernel by $K_{P_N}$. The kernel $K_{P_N}$ is the one that you refer to as the data-dependent residual kernel.
>
> The data-dependent residual kernel’s empirical matrix is $\tilde{K} = [K_{P_N}(X_i,X_j)]_{i,j=1}^N$, and it is the matrix we compute in a practical implementation of Conditional KRR. Fluctuations of this matrix can be estimated from our proof of Theorem 4.2.
>
> Note that the data-dependent residual kernel matrix $\tilde K$ can be approximated by the empirical kernel matrix
> $[K_P(X_i, X_j)]_{i,j=1}^N$
>
> , of the residual kernel $K_P$ defined by the actual data distribution $P$. So, if we believe that approximation is accurate, then the question is reduced to the study of fluctuations of the empirical kernel matrix $[K_P(X_i,X_j)]_{i,j=1}^N$
>
> , which is a subject of many works. Basically, many properties of the random matrix $[K_P(X_i,X_j)]_{i,j=1}^N$ can be understood from the spectrum of the integral operator $\phi\to \int K_P(\cdot, x)\phi(x)dP(x)$.
>
> Therefore, the next question to address in that context is how close is the matrix $\tilde{K}$ to the empirical kernel matrix $[K_P(X_i,X_j)]_{i,j=1}^N$
>
> . It is natural to measure the deviation by the operator norm of their difference, $\frac{1}{N}\tilde{K}-\frac{1}{N}[K_P(X_i,X_j)]_{i,j=1}^N$
>
> . In an argument very similar in style to Lemma D.12, the latter operator norm can be bounded by some factor of the operator norm of $\frac{1}{\sqrt{N}}\Lambda^{1/2}(\Psi-\Phi)$ (this is due to the fact that $\tilde{K} = \Psi^\top \Lambda\Psi$ and $[K_P(X_i,X_j)] = \Phi^\top \Lambda\Phi$), which itself can be bounded as in line 1179 of paper. So, this argumentation leads to a bound on the operator norm of $\frac{1}{N}\tilde{K}-\frac{1}{N}[K_P(X_i,X_j)]_{i,j=1}^N$ of order $O(\frac{k^{1/2}\log^{1/4}(2k/\delta)}{N^{1/4}})$.
>
> In summary, if $\frac{k\log^{1/2}(2k/\delta)}{\sqrt{N}}$ is small, then fluctuations of $\tilde{K}$ can be substituted by fluctuations of the random matrix $[K_P(X_i,X_j)]_{i,j=1}^N$. We can add the detailed proof of this statement in the final version of the paper.
>
>
> F9qm: "Fourth, the empirical section lacks comparisons beyond standard KRR; methods such as truncated eigen-KRR…"
>
> We will include these baselines in the revised empirical section.
>
> F9qm: "First, ..."
>
> Inequality (3) can be interpreted as a signal-versus-noise condition. In the hard-thresholding case $\mathcal{F}=\{\phi_1,...,\phi_k\}$ conditional KRR improves over standard KRR when the target has sufficiently large energy in the first $k$ eigenfunctions of the kernel. In that regime, standard KRR over-shrinks the very coordinates where the signal is strongest, while conditional KRR removes that shrinkage. By contrast, when the $\mathcal{F}$-component is weak or absent (e.g. pure-noise data), the additional conditioning cost and variance dominate, so increasing $k$ worsens performance. Thus one expects three concrete regimes: improvement for strongly aligned signals, deterioration for noise-dominated signals, and a flat dependence in the intermediate regime. Our synthetic and real world data hard-thresholding experiments and random-feature experiments are consistent with exactly this picture.

---

> > ### Author Rebuttal · Reviewer_F9qm · 2026-04-05
> >
> > I appreciate the authors' thorough clarifications on the technical aspects. The theoretical development is solid and provides new results. I maintain my score because the contribution of this work is entirely theoretical and lacks practical applicability.

---

### Official Review · Reviewer_fVuJ · 2026-03-11

**Soundness:** 3
**Presentation:** 2
**Significance:** 3
**Originality:** 3
**Overall Recommendation:** 4
**Confidence:** 4

**Summary:**

This paper investigates conditional kernel ridge regression. Building on its theoretical development, the authors propose a two-stage learning procedure in which the target function is first approximated using a finite set of $k$ features (or basis functions), after which ridge regression with a residual kernel is applied to the remaining residual component. The authors theoretically characterize the approximation gap, referred to as the cost of conditioning, between the proposed predictors and oracle counterparts. In addition, the paper discusses concrete scenarios in which conditional kernel ridge regression is expected to outperform standard kernel ridge regression. These theoretical insights are supported by simulation studies, and the paper further illustrates the practical relevance of the proposed method through an application to real data.

**Compliance With Llm Reviewing Policy:**

Affirmed.

**Final Justification:**

The paper has several notable strengths, including a well-motivated training procedure, strong theoretical development, and empirical results that support the theory. Overall, I find the paper technically interesting and theoretically meaningful, but it still has some practical limitations whose implications for real-world application have not yet been fully resolved.

**Key Questions For Authors:**

1. [Related to Weaknesses 1 and 2] Could the authors further clarify the practical motivation for conditional KRR? In particular, under what types of real-world data characteristics or problem settings do the authors expect conditional KRR to provide a meaningful advantage over standard KRR beyond the specific scenarios analyzed in the paper?

2. [Related to Weakness 3] Could the authors clarify how the number of unpenalized features, $k$, should be chosen in practice? In particular, is there any theoretical or empirical guideline for selecting $k$ in real applications? A validation-based strategy may not be practically appealing, because changing $k$ also changes the residual kernel regression problem and may necessitate re-tuning the regularization parameter $\lambda$ for each candidate $k$.

**Limitations:**

Yes

**Strengths And Weaknesses:**

[Strengths]
1. The paper proposes a model training procedure that is strongly motivated and supported by its theoretical development.
2. The theoretical analysis provides a characterization of the approximation error and offers concrete insight into scenarios where the proposed method may outperform the standard model.
3. The theoretical findings are well supported by both simulation experiments and real-data analysis.

[Weaknesses]
1. Although the theoretical analysis is carefully developed, the paper provides somewhat limited motivation for why conditional KRR is useful from a practical application perspective. In particular, the scenarios in which the proposed method may outperform standard KRR appear to be relatively specific, and it is not yet fully convincing how broadly such advantages would arise in real data analysis.
2. Related to Weakness 1, the scope of the empirical evaluation is limited. A broader assessment across more diverse datasets, together with comparisons not only to standard KRR but also to other relevant nonlinear baselines, would have allowed a clearer evaluation of the practical value.
3. The proposed method inherently relies on a finite number of unpenalized features. For widely used kernels such as the Gaussian kernel, these features must in practice be constructed through empirical extensions or Nyström-type approximations. However, such procedures introduce additional approximation error, and the choice of which features to include remains an important practical issue in itself. Moreover, the selection of the number of unpenalized features, $k$, involves an inherent trade-off. Increasing $k$ may improve the quality of the feature approximation, but, as suggested by the theoretical results, it may also increase the cost of conditioning and thereby lead to a larger expected error. On the other hand, if $k$ is chosen too small, the method becomes closer to standard KRR, potentially weakening the advantage of conditional KRR. As a result, the practical performance of the method may depend substantially on the quality of the feature approximation and on the choice of $k$, yet these practical selection issues and potential limitations are not discussed sufficiently in the paper.

---

> ### Author Rebuttal · Authors · 2026-03-31
>
> Thank you for raising these important points.
>
> fVuJ: "Could the authors further clarify the practical motivation for conditional KRR? In particular, under what types of real-world data characteristics or problem settings do the authors expect conditional KRR to provide a meaningful advantage over standard KRR beyond the specific scenarios analyzed in the paper?"
>
> C-KRR is useful in settings where part of the signal lies in a low-dimensional, interpretable feature space $\mathcal{F}$, while the remaining structure is nonlinear. In such cases, standard KRR shrinks all directions uniformly and may over-regularize the informative components. C-KRR avoids this by keeping $\mathcal{F}$ unpenalized and modeling only the residual with a kernel, leading to improved performance when the signal has sufficient energy in $\mathcal{F}$. This situation is common in practice (e.g., geometric features in vision, seasonal components in time series, or engineered features in tabular data). Some additional experiments (on handcrafted unpenalized features and $1/\sqrt{N}$-decay rate of the cost of conditioning) can be found in the paper's repository. We will strengthen the empirical section by adding comparisons to truncated KRR, additive models, and augmented random-feature methods, and by including experiments with handcrafted features and additional datasets.
>
> fVuJ: "Could the authors clarify how the number of unpenalized features, k, should be chosen in practice?"
>
> In many applications, the feature space $\mathcal{F}$ is chosen based on prior knowledge and is very interpretable. In such cases, k is naturally small and fixed, and no tuning is required.
> Although the residual kernel depends on k, we find empirically that the optimal lambda is relatively stable across k. A practical recommendation is to increase k progressively (e.g., ordered by feature importance or eigenvalues) and stop when performance ceases to improve. Our theory further suggests including only directions with sufficiently strong signal, leading to a non-monotone dependence on k observed in practice.

---

> > ### Author Rebuttal · Reviewer_fVuJ · 2026-04-01
> >
> > I thank the authors for their response. However, I will maintain my score, as my concern regarding the practical applicability of the method, especially in terms of how to determine the number of unpenalized features and which features should be selected in real-world settings, has not been sufficiently addressed.

---

### Official Review · Reviewer_REZN · 2026-03-12

**Soundness:** 3
**Presentation:** 3
**Significance:** 3
**Originality:** 3
**Overall Recommendation:** 5
**Confidence:** 3

**Summary:**

This paper studies kernel ridge regression in the setting of conditionally positive definite (CPD) kernels. In this framework, a kernel is associated with a feature space that contains a set of functions that are not penalized by the regularization term. These functions form a finite-dimensional space that plays the role of an unpenalized component of the model. Such kernels arise naturally in settings such as spline estimation and Gaussian process regression.

The paper introduces the concept of a residual kernel obtained by removing the contribution of the unpenalized feature space from the original kernel. The authors show that this residual kernel is positive definite and that conditional kernel ridge regression can be interpreted in a simple way: first perform linear regression on the chosen feature space, then apply standard kernel ridge regression to the remaining residual signal. This provides a clear interpretation of the method as combining explicit features with kernel learning.

The main theoretical contribution analyzes how closely this procedure behaves compared to an idealized learner that knows the part of the signal lying in the unpenalized feature space. The paper introduces the notion of a “cost of conditioning,” which measures the difference between the estimator produced by conditional kernel ridge regression and the oracle estimator. The authors show that this cost decreases as the sample size increases, providing guarantees that the method behaves similarly to kernel ridge regression with a modified kernel.

The theory is further studied in two specific cases: when the unpenalized features correspond to the leading eigenfunctions of the kernel, and when they are generated using random features. In both settings the analysis predicts that the test error should depend non-monotonically on the number of unpenalized features. Experiments on synthetic data and a small MNIST example illustrate this predicted U-shaped behavior and show that the proposed approach can outperform standard kernel ridge regression when the signal has a strong component in the chosen feature space.

**Compliance With Llm Reviewing Policy:**

Affirmed.

**Final Justification:**

I think the rebuttal addressed most of the concerns and if the additional discussion and experiments are added to the revised version, I would recommend accepting the paper.

**Key Questions For Authors:**

please check weaknesses above

**Limitations:**

yes

**Strengths And Weaknesses:**

Strengths
----
1. The paper provides a clear theoretical framework for kernel ridge regression with conditionally positive definite kernels. The equivalence between conditional kernel ridge regression and the two-step procedure (linear regression on the unpenalized features followed by KRR on the residuals) provides a useful structural interpretation of the method.

2. The theoretical analysis is carefully developed. In particular, the paper introduces the notion of the “cost of conditioning,” which measures the gap between conditional KRR and an oracle learner that has direct access to the unpenalized component of the signal. The main bound shows that this cost decreases at a rate proportional to the square root of the feature dimension divided by the sample size, up to logarithmic factors.

3. The application of the framework to both eigenfunction-based features and random feature constructions is well motivated. Treating a positive definite kernel as conditionally positive definite with respect to its leading eigenfunctions provides an interpretable connection between conditional KRR and spectral truncation methods.

4. The theoretical conditions derived for when conditional KRR should outperform standard KRR are interpretable and provide useful intuition about how the signal structure and feature space dimension influence performance.


Weaknesses
---

1. The main theoretical result controls the cost of conditioning, i.e. the gap between conditional KRR and an oracle learner that knows the component of the signal in the unpenalized feature space. However, the paper does not state a single end-to-end excess-risk theorem for the final conditional KRR estimator itself. It would be helpful if the authors could briefly clarify how the cost-of-conditioning result should be combined with standard KRR guarantees for the residual problem to obtain a direct test-risk interpretation.

2. In the hard-thresholding theory, the unpenalized feature space is defined using the leading population eigenfunctions of the kernel. In the real-data MNIST experiment, however, these eigenfunctions are not available and are instead estimated from samples. It would be helpful if the authors could comment on how this additional approximation step affects the interpretation of the theory and whether they expect the guarantees to extend to the empirical-eigenfunction setting.

3. The paper gives useful intuition for when conditional KRR can outperform standard KRR, but it would be helpful to clarify more explicitly in what practical regimes this is expected to offer an advantage beyond ordinary tuning of the regularization parameter in standard KRR.

---

> ### Author Rebuttal · Authors · 2026-03-31
>
> We appreciate the reviewer’s insightful feedback.
>
> REZN: "The main theoretical result controls the cost of conditioning, i.e. the gap between conditional KRR and an oracle learner that knows the component of the signal in the unpenalized feature space. However, the paper does not state a single end-to-end excess-risk theorem for the final conditional KRR estimator itself. It would be helpful if the authors could briefly clarify how the cost-of-conditioning result should be combined with standard KRR guarantees for the residual problem to obtain a direct test-risk interpretation."
>
> One of the ways to have a bound on the expected test risk of C-KRR would be the following.
> Let $\hat f$ denote the C-KRR estimator, and let $f = f_{\parallel} + f_{\perp}$ be the decomposition of the target. Let $h$ denote the estimator obtained by performing KRR with the residual kernel $K_P$ and targets without $\mathcal{F}$-part. By definition, the cost of conditioning is $c_{{\rm con}} = \|\hat f - (f_{\parallel} + h)\|_{L_2(\mathcal{X}, P)}^2$.
>
> Using the decomposition
>
> $\hat f - f = \big(\hat f - (f_{\parallel} + h)\big) + (h - f_{\perp})$,
>
> and applying the inequality $\|a+b\|^2 \le 2\|a\|^2 + 2\|b\|^2$, we obtain
>
> $\lVert \hat f - f \rVert_{L_2(\mathcal{X}, P)}^2 \le 2 c_{con} + 2 \lVert h - f_{\perp} \rVert_{L_2(\mathcal{X}, P)}^2$
>
> The second term corresponds exactly to the test error of standard KRR applied to the residual problem with kernel $K_P$. E.g. it can be estimated using a kernel alignment risk estimator.
>
> There is also another approach, based on the triangle inequality, that does not have the factor 2 in front of standard KRR’s risk, but has other disadvantages.
>
> REZN: "In the hard-thresholding theory…"
>
> The same question was asked by XqBe and answered before.
>
> REZN: "… in what practical regimes this is expected to offer an advantage beyond ordinary tuning of the regularization parameter in standard KRR."
>
> The last question of XqBe is very similar and was answered before. Also, there is an overlap with the first question of fVuJ.

---

> > ### Author Rebuttal · Reviewer_REZN · 2026-04-02
> >
> > Thanks for your response for the questions and providing the additional experiments. I will increase my score.

---

### Official Review · Reviewer_XqBe · 2026-03-13

**Soundness:** 3
**Presentation:** 3
**Significance:** 3
**Originality:** 3
**Overall Recommendation:** 4
**Confidence:** 3

**Summary:**

This paper introduces conditional Kernel Ridge Regression (C-KRR), a framework that integrates a class of unpenalized features into standard KRR using conditionally positive definite kernels. The authors demonstrate that C-KRR is mathematically equivalent to performing unpenalized regression on these selected features, followed by standard KRR on the residuals using a strictly positive definite residual kernel. The primary theoretical contribution is establishing an $\mathcal{O}(1/\sqrt{N})$ upper bound for the cost of conditioning, quantifying the distance between the C-KRR estimator and an ideal learner. Applications to hard thresholding (principal eigenfunctions) and soft thresholding (random features) theoretically explain and empirically verify a U-shaped test error curve, showing performance gains when the unpenalized signal is dominant.

**Compliance With Llm Reviewing Policy:**

Affirmed.

**Key Questions For Authors:**

1. In Appendix H.3, the features are learned by a neural network on the same training set. Theorem 4.2 assumes $\mathcal{F}$ is fixed or independent of the current data. If $\mathcal{F}$ is data-dependent, does the current theoretical bound still hold?
2. If we redefine the baseline and directly compare the practical estimators $|\hat{f}\_{\text{C-KRR}} - \hat{f}\_{\text{Standard-KRR}}|$ (instead of introducing the idealized $\mathcal{F}$-conditional learner), can the proof framework of Theorem 4.2 be adapted to derive a theoretical bound for their error gap?
3. Can the authors provide a real-world experiment (without relying on pre-trained neural network features) where C-KRR clearly outperforms standard KRR with an optimally tuned $\lambda$?

**Limitations:**

The paper currently lacks any discussion of broader societal impacts. On the other hand, they touched upon some technical limitations.

**Strengths And Weaknesses:**

Strengths:
1. Theorem 3.2 establishes elegant theoretical equivalence, which is not only mathematically rigorous but also highly interpretable.
2. This paper establishes solid statistical learning bounds for conditional KRR.

Weaknesses:
1. While the theoretical framework is elegantly constructed, its practical utility on real-world datasets is questionable. In Appendix H.2 Table 1, when the regularization parameter $\lambda$ of standard KRR is optimally tuned, its test error (e.g., 0.0671 for the Gaussian kernel) is actually lower than that of Conditional KRR (0.0732). This indicates that the added algorithmic complexity of selecting the optimal number of unpenalized features $k$ and computing the residual kernel fails to yield enough performance gains over the optimized-regularized KRR in non-synthetic scenarios.
2. The cost of conditioning is defined as the discrepancy between the C-KRR estimator and an omniscient $\mathcal{F}$-conditional learner. However, this ideal baseline assumes prior, exact knowledge of the target function's component $f_\parallel$ within the specific feature space $\mathcal{F}$. From the perspective of practical algorithmic evaluation, directly quantifying the generalization error gap between C-KRR and standard KRR would serve as a much more meaningful and informative metric.
3. All theoretical bounds in this paper (e.g., Theorem 4.2) strictly require $\lambda > 0$. Given that the first stage of the equivalent C-KRR procedure is completely unregularized, exploring the framework's behavior as $\lambda \to 0$ is crucial for a complete theoretical picture.
4. Real-World experiment demonstrating a clear U-shaped performance curve relies on a pre-trained neural network to extract features, which weakens the argument for this framework as an independent improvement for kernel methods.

---

> ### Author Rebuttal · Authors · 2026-03-31
>
> We thank the reviewer for helpful and thoughtful comments.
>
> XqBe: “If $\mathcal{F}$ is data-dependent, does the current theoretical bound still hold?” + REZN: “In the hard-thresholding theory, the unpenalized feature space is defined using the leading population eigenfunctions of the kernel. In the real-data MNIST experiment, however, these eigenfunctions are not available and are instead estimated from samples. It would be helpful if the authors could comment on how this additional approximation step affects the interpretation of the theory and whether they expect the guarantees to extend to the empirical-eigenfunction setting.”
>
> Yes, the bound was obtained under the assumption that the unpenalized features are data-independent. Therefore, it does not strictly hold when $\mathcal{F}$ is data-dependent. Among the experiments reported in the paper, only two involve data-dependent unpenalized features.
>
> The first case is Appendix H.2, where eigenfunctions are estimated from data. When $k$ is moderate, this estimation is quite accurate (analogous to estimating the spectrum of an integral operator via empirical eigenvalues, which is reliable for lower-order components). Hence, the bound should remain reasonably accurate for smaller values of $k$ in this setting.
>
> The second case is Appendix H.3, where pretrained features are used as unpenalized features. This experiment was included only to demonstrate that injecting informative features can improve KRR. To address the reviewer’s concern (and Question 3), we additionally conducted experiments on the same dataset (MNIST 7 vs 9) using a simple set of handcrafted, non-pretrained features. The outcome is similar: injecting good features improves performance.
>
> XqBe: "If we redefine the baseline and directly compare the practical estimators
> $|\hat f_{\text{C-KRR}} - \hat f_{\text{Standard-KRR}}|$ (instead of introducing the idealized $\mathcal{F}$-conditional learner), can the proof framework of Theorem 4.2 be adapted to derive a theoretical bound for their error gap?"
>
> Instead of adapting the proof framework, we would reduce the estimation of $|\hat f_{\text{C-KRR}} - \hat f_{\text{Standard-KRR}}|$ to the estimation of the cost of conditioning and the estimation of $|\hat f_{\text{F-cond-learner}} - \hat f_{\text{Standard-KRR}}|$. In our opinion, that would be the most natural approach.
>
> Then, estimation of $|\hat f_{\text{F-cond-learner}} - \hat f_{\text{Standard-KRR}}|$ is easier, because both estimators deal with fixed kernels. Paradoxically, your question gives an argument that the $\mathcal{F}$-conditional learner is an inevitable proxy object to study C-KRR.
>
> XqBe: “Can the authors provide a real-world experiment (without relying on pre-trained neural network features) where C-KRR clearly outperforms standard KRR with an optimally tuned $\lambda$?” + REZN:” … in what practical regimes this is expected to offer an advantage beyond ordinary tuning of the regularization parameter in standard KRR.”
>
>
> Yes, consider an experiment with the same dataset, MNIST 7 vs 9, and the set $\mathcal{F}$ defined as the following handcrafted 11 features: 1) total ink, 2) center of mass x, 3) center of mass y, 4) second central moment xx, 5) second central moment yy, 6) second central moment xy, 7) upper-half ink, 8) lower-half ink, 9) left-half ink, 10) right-half ink, 11) central 8x8 ink. All the rest is the same as in the experiment from Remark H.1 of Appendix H.3.
>
> So, features are not pre-trained, but the results are quite similar to the previous experiments:
>
> lambda=1e-02 | KRR=0.4268 | C-KRR handcrafted=0.3767
>
> lambda=1e-03 | KRR=0.2443 | C-KRR handcrafted=0.2299
>
> lambda=1e-04 | KRR=0.2359 | C-KRR handcrafted=0.1936
>
> lambda=1e-05 | KRR=0.4441 | C-KRR handcrafted=0.4141
>
> lambda=1e-06 | KRR=0.7704 | C-KRR handcrafted=0.9827
>
> So, that is an expected outcome: when good features are injected, C-KRR improves KRR.
>
>
>
> XqBe: “All theoretical bounds in this paper (e.g., Theorem 4.2) strictly require $\lambda>0$. Given that the first stage of the equivalent C-KRR procedure is completely unregularized, exploring the framework's behavior as $\lambda\to 0$ is crucial for a complete theoretical picture.”
>
> This is the task for future work. The main difficulty in the analysis of C-KRR in the regime $\lambda\to 0$ is that the residual kernel $K_P$ is ill-conditioned by construction (because we eliminate the $\mathcal{F}$-component of its image space). Though still the $\mathcal{F}$-conditional learner has to deal with “full-rank noise”. So, the cost of conditioning cannot be easily controlled in that regime and C-KRR has to be studied directly.

---

> > ### Author Rebuttal · Reviewer_XqBe · 2026-04-03
> >
> > Thanks for the rebuttal. The authors address most of my concerns, and I remain with a positive score.

---

### Decision · Program_Chairs · 2026-04-30

**Decision:**

Accept (regular)

**Comment:**

The submission provides a theoretical analysis of learning a linear predictor with some unregularized features and many regularized features, or equivalently a kernel.  The reviewers all appreciated the contribution and found it valuable.